# Nonequilibrium polysome dynamics promote chromosome segregation and its coupling to cell growth in *Escherichia coli*

**Alexandros Papagiannakis[1,2,3], Qiwei Yu[4], Sander K Govers[1†], Wei-Hsiang Lin[1,2‡], Ned S Wingreen[4,5], Christine Jacobs-Wagner[1,2,3,6]\***

[1]Howard Hughes Medical Institute, Stanford University, Stanford, United States; [2]Sarafan Chemistry, Engineering, and Medicine for Human Health Institute, Stanford University, Stanford, United States; [3]Department of Biology, Stanford University, Stanford, United States; [4]Lewis-Sigler Institute for Integrative Genomics, Princeton University, Princeton, United States; [5]Department of Molecular Biology, Princeton University, Princeton, United States; [6]Department of Microbiology and Immunology, School of Medicine, Stanford University, Stanford, United States

**\*For correspondence:**
jacobs-wagner@stanford.edu

**Present address:** †Department of Biology, KU Leuven, Leuven, Belgium; ‡Institute of Molecular Biology, Academia Sinica, Taipei, Taiwan

**Competing interest:** The authors declare that no competing interests exist.

## eLife Assessment

This **important** study presents **compelling** observational data supporting a role for transcription and polysome accumulation in the separation of newly replicated bacterial chromosomes. Through a comprehensive and rigorous comparative analysis of the spatiotemporal dynamics of ribosomal accumulation, nucleoid segregation, and cell division, the authors develop a model that nucleoid segregation rates are determined at least in part by the accumulation of ribosomes in the center of the cell, exerting a steric force to drive nucleoid segregation prior to cell division. This model circumvents the need to invoke as yet unidentified active mechanisms (e.g. an equivalent to a eukaryotic spindle) as drivers of bacterial chromosome segregation and intrinsically couples this vital step in the cell cycle to cell growth.

**Abstract** Chromosome segregation is essential for cellular proliferation. Unlike eukaryotes, bacteria lack cytoskeleton-based machinery to segregate their chromosomal DNA (nucleoid). The bacterial ParABS system segregates the duplicated chromosomal regions near the origin of replication. However, this function does not explain how bacterial cells partition the rest (bulk) of the chromosomal material. Furthermore, some bacteria, including *Escherichia coli*, lack a ParABS system. Yet, *E. coli* faithfully segregates nucleoids across various growth rates. Here, we provide theoretical and experimental evidence that polysome production during chromosomal gene expression helps compact, split, segregate, and position nucleoids in *E. coli* through nonequilibrium dynamics that depend on polysome synthesis, degradation (through mRNA decay), and exclusion from the DNA meshwork. These dynamics inherently couple chromosome segregation to biomass growth across nutritional conditions. Halting chromosomal gene expression and thus polysome production immediately stops sister nucleoid migration, while ensuing polysome depletion gradually reverses nucleoid segregation. Redirecting gene expression away from the chromosome and toward plasmids causes ectopic polysome accumulations that are sufficient to drive aberrant nucleoid dynamics. Cell width enlargement experiments suggest that limiting the exchange of polysomes across DNA-free regions ensures nucleoid segregation along the cell length. Our findings suggest a self-organizing mechanism for coupling nucleoid compaction and segregation to cell growth without the apparent requirement of regulatory molecules.

## Introduction

All cells must segregate their replicated chromosomes to propagate genetic information to daughter cells at division. Remarkably, this essential process is far less understood in bacteria than in eukaryotes. Eukaryotic cells use the mitotic spindle, a sophisticated cytoskeleton-based machine, for chromosome segregation. No equivalent structure has been identified in bacteria, which fold their chromosomal material into a membrane-less organelle called the nucleoid. Many bacterial species use a ParABS system to segregate a specific DNA region proximal to the chromosomal origin of replication (*oriC*) (*Figge et al., 2003*; *Ireton et al., 1994*; *Jalal and Le, 2020*; *Lim et al., 2014*; *Livny et al., 2007*; *Mohl and Gober, 1997*). However, this initial step does not explain how the terminal regions of the chromosome separate (*Harju et al., 2024*) or how sister nucleoids move away from each other. Moreover, the chromosomally encoded *parA* and/or *parB* genes are often not essential for cell viability in various bacteria (*Bartosik et al., 2009*; *Charaka and Misra, 2012*; *Donczew et al., 2016*; *Donovan et al., 2010*; *Du et al., 2016*; *Ireton et al., 1994*; *Jakimowicz et al., 2007a*; *Jakimowicz et al., 2007b*; *Jecz et al., 2015*; *Kadoya et al., 2011*; *Kawalek et al., 2020*; *Kim et al., 2000*; *Lagage et al., 2016*; *Lee and Grossman, 2006*; *Lewis et al., 2002*; *Li, 2019*; *Li et al., 2015*; *Minnen et al., 2011*; *Santi and McKinney, 2015*; *Takacs et al., 2022*; *Yamaichi et al., 2007*; *Yu et al., 2010*). In fact, some bacteria do not even encode a ParABS system for *oriC* partitioning (*Livny et al., 2007*), with *Escherichia coli* being a prime example. Yet, *E. coli* segregates its duplicated nucleoids faithfully.

Several non-dedicated cellular processes have been proposed to play a role. For example, DNA replication may facilitate chromosome partitioning through a DNA polymerase-dependent extrusion mechanism (*Lemon and Grossman, 2000*). Simulations of coarse-grained polymer models have also suggested that the conformational entropy of confined circular DNA chains could provide an unmixing force that separates two mixed DNA polymers (*Jun and Mulder, 2006*; *Jun and Wright, 2010*; *Pande et al., 2023*). However, recent modeling work has argued that entropic forces generated from excluded volume interactions between partially replicated chromosomes inhibit (rather than promote) their segregation (*Harju et al., 2024*). Mechanical stress between overlapping sister nucleoids (*Fisher et al., 2013*) and DNA loop extrusion (*Harju et al., 2024*) have also been proposed to contribute to chromosome segregation. *E. coli* mutants in which the *mukB* gene required for DNA loop extrusion is deleted exhibit moderate nucleoid segregation defects linked to the organization and condensation of the nucleoid (*Danilova et al., 2007*; *Mäkelä et al., 2021*; *Sawitzke and Austin, 2000*). However, even in these mutants, most cells successfully segregate and partition their nucleoids between the daughter cells. Furthermore, topoisomerase I mutations or *seqA* deletion suppress the chromosome segregation defects associated with the *mukB* deletion (*Danilova et al., 2007*; *Sawitzke and Austin, 2000*; *Weitao et al., 1999*), suggesting that other mechanisms exist.

Furthermore, while these various proposed mechanisms likely contribute to the splitting of the nucleoid into two objects, they cannot explain how separated sister nucleoids move away from each other. In addition, the dominant mechanism underlying the nucleoid diffusional bias must somehow be coordinated with the growth rate. In the presence of a poor-quality carbon source, *E. coli* initiates nucleoid segregation late in the cell division cycle. As the quality of the carbon source improves and the growth rate increases, nucleoid segregation occurs at an increasingly earlier stage of the cell division cycle until it initiates in the preceding cycle to accommodate faster division times (*Bates and Kleckner, 2005*; *Govers et al., 2024*; *Tiruvadi-Krishnan et al., 2022*). *Jacob et al., 1963* offered an elegant solution to couple chromosome segregation to cell growth by hypothesizing that duplicated chromosomes migrate apart through membrane attachment and localized growth of the membrane between the attachment points. Drug experiments have indeed suggested that the chromosomal DNA is attached to the cytoplasmic membrane along the radial axis through a process known as transertion (*Bakshi et al., 2015*; *Binenbaum et al., 1999*; *Kaval et al., 2023*; *Kim et al., 2024*; *Matsumoto et al., 2015*; *Spahn et al., 2023*; *Woldringh, 2002*), which occurs when transcription, translation, and insertion of membrane proteins occur at the same time. However, for the nucleoids to migrate directionally, the 1963 model requires localized growth of the cytoplasmic membrane between the splitting nucleoids. To our knowledge, there is no evidence of such localized expansion of the inner membrane. Even if the connection between the DNA and cytoplasmic membrane extends to the peptidoglycan cell wall, *E. coli* cells elongate via a dispersed mode of peptidoglycan growth along the cell length up to the final stage of the division cycle (*Cooper and Hsieh, 1988*; *Navarro et al., 2022*; *Wientjes and Nanninga, 1989*; *Woldringh et al., 1987*). In addition, experiments with

*B. subtilis* cells stripped of their cell walls indicate that nucleoid segregation does not require cell wall attachment (*Wu et al., 2020*). Finally, the segregation of chromosomal loci is faster than the rate of cell elongation, ruling out cell elongation and nucleoid tethering to the membrane or cell wall as a major driver of chromosome segregation (*Kuwada et al., 2013*).

We hypothesized that the mechanism(s) driving nucleoid segregation and its coupling to growth rate may be linked to the bacterial cellular organization. In contrast to eukaryotes, bacteria do not insulate their DNA in membrane-bound compartments. As a result, the chromosomal meshwork can freely interact with the macromolecules present in the cytoplasm, which exhibits a high level of polydispersity and crowding (*McGuffee and Elcock, 2010*; *Zimmerman and Trach, 1991*). Experiments and modeling have suggested that macromolecular crowders exert compaction forces on the compressible nucleoids through steric (i.e. excluded-volume) interactions (*Mondal et al., 2011*; *Pelletier et al., 2012*; *Wu et al., 2019a*; *Yang et al., 2020*; *Zhang et al., 2009*). Monosomes and polysomes (hereafter collectively referred to as 'polysomes' for simplicity), which consist of mRNAs loaded with one or more translating ribosomes, constitute a sizeable and abundant cytoplasmic crowder (*McGuffee and Elcock, 2010*). The proteome fraction allocated to ribosomes increases with growth rate across nutrient conditions (*Chure and Cremer, 2023*; *Dai et al., 2016*; *Dourado and Lercher, 2020*; *Hu et al., 2020*; *Molenaar et al., 2009*; *Scott et al., 2014*; *Si et al., 2017*). The proportion of ribosomes engaged in translation also increases with improving nutrient quality, attaining 70–80% under fast growth (*Dai et al., 2016*; *Forchhammer and Lindahl, 1971*; *Mohapatra and Weisshaar, 2018*; *Sanamrad et al., 2014*; *Varricchio and Monier, 1971*). These polysomes form structures (*Brandt et al., 2009*) of comparable or larger size than the average ~50 nm mesh size of the nucleoid (*Xiang et al., 2021*). Consistent with a steric clash between polysomes and the nucleoid meshwork, fluorescently labeled ribosomes accumulate in DNA-free regions such as the cell poles and between segregated nucleoids (*Azam et al., 2000*; *Bakshi et al., 2015*; *Bakshi et al., 2012*; *Chai et al., 2014*; *Gray et al., 2019*; *Lewis et al., 2000*; *Mohapatra and Weisshaar, 2018*; *Robinow and Kellenberger, 1994*; *Sanamrad et al., 2014*; *Xiang et al., 2021*). Interestingly, theoretical work suggests that excluded-volume effects alone may be sufficient for the DNA to spontaneously phase separate from polysomes and compact into its observed nucleoid form (*Mondal et al., 2011*). Furthermore, a recent nonequilibrium statistical physics model proposes that nucleoid positioning and segregation could potentially be explained by the steric interaction (i.e. repulsion) between polysomes and DNA and the nonequilibrium effects associated with mRNA synthesis and degradation (*Miangolarra et al., 2021*). In this theoretical model, polysome accumulation in the middle of the nucleoid emerges due to polysomes born in the middle of the nucleoid taking longer to diffuse out of the nucleoid than those born closer to the nucleoid edges. Polysome accumulation beyond a certain concentration threshold drives phase separation between the DNA and polysomes at the mid-nucleoid location, resulting in nucleoid splitting. However, we lack experimental validation of this model. Furthermore, the coupling between nucleoid segregation and growth rate has not been addressed either experimentally or theoretically.

Here, we quantitatively characterize the temporal and spatial dynamics of nucleoids and polysomes in *E. coli* under different conditions and perturbations. In this work, we refer to nucleoid segregation as a series of events observable by microscopy (*Figure 1A*): (i) the initiation of nucleoid splitting, marked by the depletion of a DNA marker near the mid position of the nucleoid, (ii) the end of nucleoid splitting, which results in the generation of two separable nucleoid objects, and (iii) the migration of the sister nucleoids away from each other, marked by the increasing distance between the centroid of the nucleoids. This sequence of events can be initiated in the preceding cell division cycle under nutrient-rich (fast-growth) conditions. Our experimental findings, combined with modeling, provide evidence that out-of-equilibrium and asymmetric polysome rearrangements in the cell result in both DNA compaction and the choreography of the nucleoid segregation cycle during growth. The dual involvement of polysomes in protein synthesis and nucleoid dynamics ensures self-regulating coordination between cell growth and chromosome segregation across a wide range of growth rates, even under conditions where *E. coli* divides faster than it can replicate its chromosome.

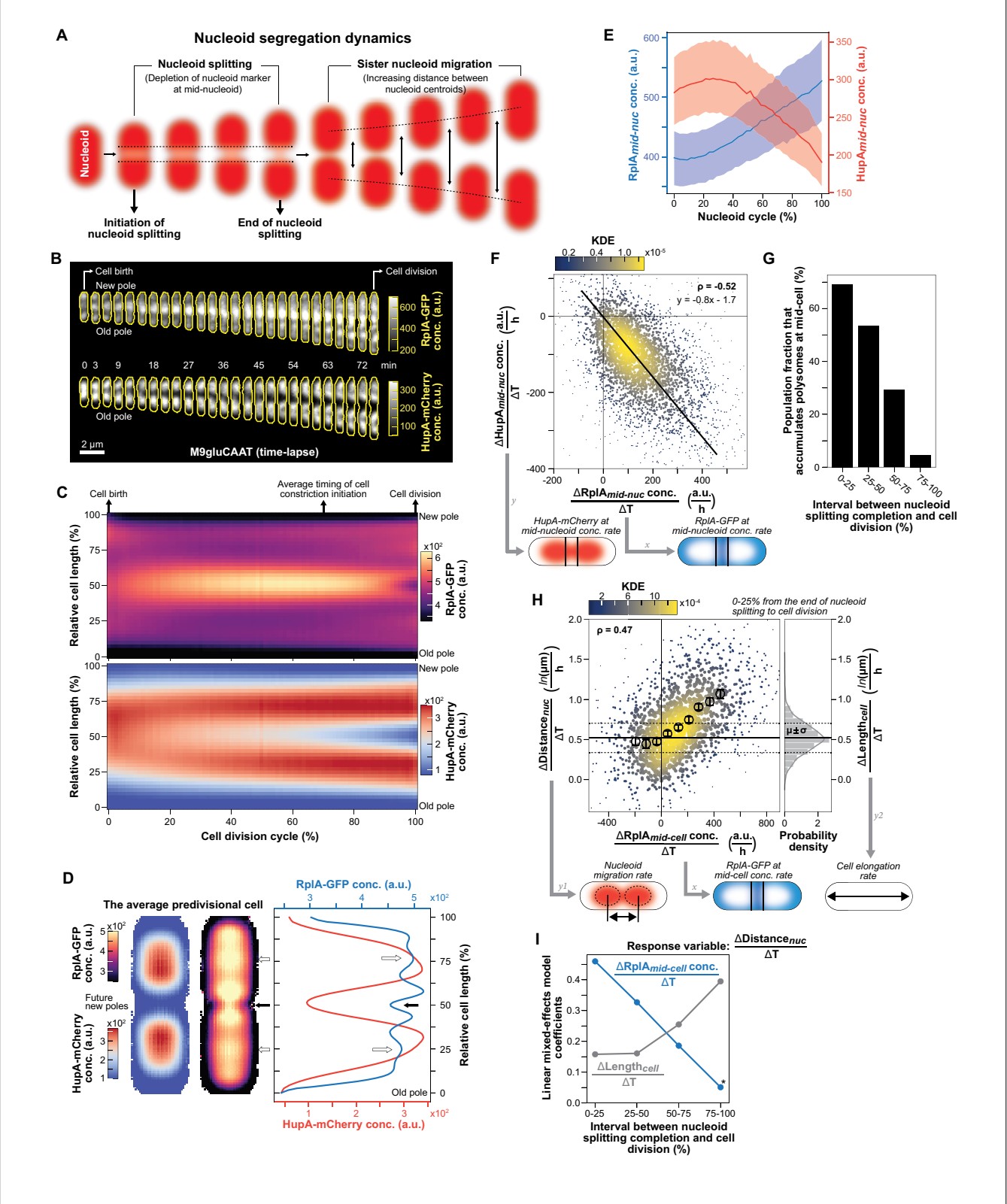

**Figure 1.** Correlations between polysome and nucleoid dynamics at the single-cell level. CJW7323 cells were grown in M9gluCAAT in a microfluidic device. (**A**) Schematic illustrating observable nucleoid segregation events. (**B**) Fluorescence images of RplA-GFP and HupA-mCherry for a representative cell (CJW7323) from birth to division. (**C**) Ensemble kymographs of the average RplA-GFP and HupA-mCherry fluorescence during the cell division cycle (>300,000 segmented cell instances from 4122 complete cell division cycles). The average relative timing of cell constriction initiation was estimated

*Figure 1 continued on next page*

*Figure 1 continued*

as shown in *Figure 1—figure supplement 2*. (**D**) Two-dimensional projections of the average RplA-GFP and HupA-mCherry fluorescence signals in predivisional cells (4564 cells with two nucleoid objects, from 1907 cell division cycles, 95–100% into the cell division cycle) and their intensity profiles. White arrows indicate RplA-GFP enrichments at the quarter cell positions, while the black arrow indicates the site of cell constriction. (**E**) Plot showing the dynamics of RplA-GFP accumulation and HupA-mCherry depletion at mid-nucleoid (median ± IQR) during the nucleoid cycle (see *Figure 1—figure supplement 4D*). Data from 3240 nucleoid segregation cycles are shown (40 nucleoid cycle bins, 2512–4823 segmented nucleoids per bin). (**F**) Correlation (Spearman $\rho$ =–0.52, p-value <$10^{-10}$) between the rate of RplA-GFP accumulation in the middle of the nucleoid ($\frac{\Delta RplA_{mid-nuc} \, conc.}{\Delta T}$) and the rate of HupA-mCherry depletion in the same region ($\frac{\Delta HupA_{mid-nuc} \, conc.}{\Delta T}$) between the initiation of nucleoid splitting and just before the end of nucleoid splitting (3214 complete nucleoid cycles). The color map and the marker size indicate the Gaussian kernel density estimation (KDE). Solid black line indicates the linear regression fit to the data. (**G**) Percentage of cells that continue to accumulate polysomes in the middle of the cell ($\frac{\Delta RplA_{mid-cell} \, conc.}{\Delta T}$) during four relative time bins (1335–1957 cell division cycles per bin) covering the period from the end of nucleoid splitting until cell division. (**H**) Correlation (Spearman $\rho$ =0.47, p-value <$10^{-10}$) between the rate of RplA-GFP accumulation at mid-cell ($\frac{\Delta RplA_{mid-cell} \, conc.}{\Delta T}$) and the rate of distance increase between the sister nucleoids ($\frac{\Delta Distance_{nuc}}{\Delta T}$) during the first quartile (0–25%) of the period between the end of nucleoid splitting and cell division (1376 cell division cycles). The black markers correspond to nine bins (mean ± SEM, 75–177 cell division cycles per bin) within the 5th-95th percentiles of x-axis range. Also shown is the distribution of the cell elongation rates ($\frac{\Delta Length_{cell}}{\Delta T}$) during the same time interval, with the mean and SD shown by the solid and dashed lines, respectively. (**I**) Plot showing the coefficients of a linear mixed-effects model (see *Equation 3* in Methods and *Figure 1—figure supplement 5B*) for four interval bins between the completion of nucleoid splitting and cell division. The coefficients quantify the relative contribution of polysome accumulation at mid-cell ($\frac{\Delta RplA_{mid-cell} \, conc.}{\Delta T}$) and cell elongation ($\frac{\Delta Length_{cell}}{\Delta T}$) to the rate of sister nucleoid migration ($\frac{\Delta Distance_{nuc}}{\Delta T}$). All coefficients are significant (Prob(<|Z|)<$10^{-9}$), except for the one marked with an asterisk that is marginally significant (Prob(<|Z|)=0.02).

The online version of this article includes the following figure supplement(s) for figure 1:

**Figure supplement 1.** Reproducibility analysis of the dynamic ribosome and nucleoid distributions between microfluidic experiments.

**Figure supplement 2.** Determination of the relative timing of cell constriction.

**Figure supplement 3.** Intracellular distributions of RplA-GFP in rifampicin-treated cells following rifampicin removal.

**Figure supplement 4.** Tracking nucleoid segregation cycles.

**Figure supplement 5.** Correlations used to calculate the relative contribution of polysome accumulation and cell elongation to nucleoid migration.

**Figure supplement 6.** Examination of the relative timing of the initiation of nucleoid constriction and the accumulation of polysomes at mid-nucleoid.

## Results

### The rate of polysome accumulation correlates with the rate of nucleoid segregation across cells

To carefully examine the dynamics between nucleoids and polysomes with high temporal resolution (every 1 or 3 min, *Figure 1—figure supplement 1A–D*), we ran time-lapse microfluidic experiments using an *E. coli* strain carrying the 50 S ribosomal protein RplA fused to GFP and the nucleoid-associated protein HupA tagged with mCherry (*Xiang et al., 2021*). Cells were grown at 30°C in M9 buffer supplemented with glucose, casamino acids, and thiamine (M9gluCAAT). While the ribosome signal displayed the expected polar and inter-nucleoid enrichments, the accrual of ribosome signal in the middle of the cells was particularly strong compared to the polar regions, as shown by representative individual cells (*Figure 1B* and *Video 1*) and average kymographs (*Figure 1C*, n=4122 cell division cycles). Cell constriction contributed to the apparent depletion of ribosomal signal from the mid-cell region at the end of the cell division cycle (*Figure 1B and C*, *Figure 1—figure supplement 2*). In predivisional cells, the ribosomal signal accumulated in the middle of the segregated nucleoids near the ¼ and ¾ cell positions, as shown in average one-dimensional (1D) and two-dimensional (2D) fluorescence profiles (n=1907 predivisional cells, *Figure 1D*). Drug treatment and single-ribosome tracking experiments have demonstrated that ribosomal accumulations inside *E. coli* correspond to polysomes, whereas free ribosomal (or ribosomal subunits) are homogeneously distributed (*Bakshi et al., 2012*; *Gray et al., 2019*; *Lewis et al., 2000*; *Linnik et al., 2024*; *Sanamrad et al., 2014*; *Xiang et al., 2021*). This is in agreement both with our own results obtained using rifampicin (*Figure 1—figure supplement 3*), a transcriptional inhibitor that causes polysome depletion over time (*Blundell and Wild, 1971*; *Campbell et al., 2001*; *Hartmann et al., 1967*), and with the polysome classification in cryo-electron tomograms of *E. coli* sections (*Xiang et al., 2021*). Since ribosomal enrichments consist of polysomes, we will, therefore, refer to these enrichments as polysome accumulations hereafter.

Under our relatively nutrient-rich growth conditions (M9gluCAAT), nucleoid splitting occurred early in the division cycle and occasionally initiated in the segregated nucleoids from the preceding

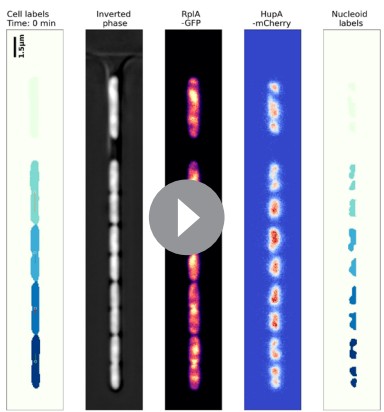

Cell labels
Time: 0 min

Inverted
phase

RplA
-GFP

HupA
-mCherry

Nucleoid
labels

**Video 1.** Example time-lapse sequence showing ribosome and nucleoid dynamics during cell growth under steady-state condition in a microfluidic channel. Shown are corresponding inverted phase contrast signal, RplA-GFP signal, HupA-mCherry signal, cell contours (based on phase contrast signals), and nucleoid contours (based on nucleoid signal segmentation) of *E. coli* cells (CJW7323) grown in a microfluidic channel supplemented with M9gluCAAT. The white circles indicate the centroid of the cell segmentation masks and the extending lines indicate the tracked cell traces.

https://elifesciences.org/articles/104276/figures#video1

division cycle (*Video 1* and *Figure 1B*). Given the variability in nucleoid segregation timing across cells (*Figure 1—figure supplement 4A*), we developed a computational method to track nucleoid dynamics independently of the cell division cycle (see Methods and *Figure 1—figure supplement 4B and C*). Specifically, we focused on the nucleoid cycle—defined as the period between the ends of two nucleoid splitting events (*Figure 1—figure supplement 4D*)—instead of the division cycle. By tracking the accumulation of RplA-GFP and the depletion of HupA-mCherry in the middle of nucleoids, we found that polysome accumulation and nucleoid splitting correlated in time (*Figure 1E*). Furthermore, the rate of polysome accumulation at mid-nucleoid ($\Delta RplA_{mid-nuc}/\Delta T$) correlated with the rate of DNA depletion in the same region ($\Delta HupA_{mid-nuc}/\Delta T$) at the single-cell level ($\rho$ =–0.52, *Figure 1F*). This indicates that cells that accumulated polysomes faster also split their nucleoids faster. Importantly, the fitted linear regression had an intercept close to zero for both axes (*Figure 1F*), indicating that when the rate of polysome accumulation approached zero, so did the rate of nucleoid splitting.

To examine what happens when sister nucleoids move away from each other, we divided the time between the end of nucleoid splitting and cell division into four bins. We found that right after nucleoid splitting (0–25% bin), most (~70%) cells continued to accumulate polysomes at mid-cell, i.e., between the sister nucleoids (*Figure 1G*). For these cells, the rate of polysome accumulation ($\Delta RplA_{mid-cell}/\Delta T$) positively correlated with the rate of nucleoid migration ($\Delta Distance_{nuc}/\Delta T$) ($\rho$ =0.47, *Figure 1H*). Thus, the faster that cells accumulated polysomes at mid-cell, the faster the sister nucleoids migrated apart (and vice versa).

## Cell elongation may also contribute to sister nucleoid migration near the end of the division cycle

As noted in the Introduction, previous work has shown that the rate of chromosomal loci is faster than that of cell elongation, indicating that cell elongation is not the predominant process driving chromosome segregation (*Kuwada et al., 2013*). This interpretation is consistent with our rate measurements of nucleoid segregation and cell elongation (*Figure 1H*). However, several observations suggest that cell elongation may play a complementary role to polysome accumulations in nucleoid segregation, particularly near the end of the division cycle. First, we noted that there was a substantial percentage (~30%) of cells with decreasing RplA-GFP signal at mid-cell right after nucleoid splitting ($\Delta RplA_{mid-cell}/\Delta T \leq 0$, *Figure 1H*). Interestingly, in these cells, nucleoid migration did not stop; instead, its average rate was similar to the average rate of cell elongation ($\Delta Length_{cell}/\Delta T$) (*Figure 1H*). In fact, when the cell elongation rate was subtracted from the nucleoid migration rate for each single-cell (*Figure 1—figure supplement 5A*), the positive correlation between polysome accumulation and nucleoid migration rates remained ($\rho$ =0.47), but now the average rate of nucleoid migration was near zero in cells with no polysome accumulation ($\Delta RplA_{mid-cell}/\Delta T \leq 0$).

This finding may suggest a mixed contribution between polysome accumulation and cell elongation to nucleoid migration. This was interesting considering that polysomes became less enriched at mid-cell but more enriched in the middle of sister nucleoids in predivisional cells (*Figure 1B–D*). This change corresponded to a redistribution of polysome enrichments, as the average ribosome

concentration remained constant during the cell division cycle (*Figure 1—figure supplement 1E*). This spatial change in polysome enrichments was accompanied by a steady decline in the percentage of cells that continued to accumulate polysomes at mid-cell between the end of nucleoid splitting and cell division (*Figure 1G*). Concurrent with this decline, the migration rate of sister nucleoids became less correlated with the rate of polysome accumulation and more correlated with the rate of cell elongation (*Figure 1—figure supplement 5B*). To examine the relative correlation of polysome accumulation and cell elongation with nucleoid migration over time, we used a linear mixed-effects model to analyze each relative nucleoid migration interval (see Methods). The coefficients of the fitted mixed linear regressions suggest the following hypothesis: Early during nucleoid migration (0–25% between the completion of nucleoid splitting and cell division), polysome accumulation contributes most to the measured variance in the displacement of the sister nucleoids (*Figure 1I*). This contribution progressively decreases over time, while that of cell elongation increases (*Figure 1I*). Such a switch in relative contribution would be consistent with the spatiotemporal dynamics of polysome accumulation and cell wall synthesis (see Discussion).

## Polysome accumulation at mid-cell correlates with the relative timing of nucleoid segregation across nutrient conditions and growth rates

If polysome production plays a role in nucleoid segregation, it predicts that the timing and amount of polysome accumulation at mid-cell will correlate with the timing of nucleoid segregation across nutrient conditions. To test this prediction, we analyzed images of fluorescently labeled ribosomes and nucleoids in cells grown under 30 different carbon source conditions (*Supplementary file 1*) that vary the doubling times (~40 min to ~4 hr) and average cell areas (~1.9 to ~3 μm$^2$) of *E. coli*. This dataset included both previously published (*Gray et al., 2019*) and new microscopy snapshots from our laboratory. Demographs generated from these images revealed that the polysome accumulation at mid-cell was reproducible across all conditions and strains, irrespective of the ribosomal subunit protein (RplA or RpsB) or the fluorescent tag (msfGFP, mEos2, or GFP) used to mark ribosomes (*Figure 2A*, *Figure 2—figure supplement 1*). Since these profiles were generated from snapshot images, they confirmed that the mid-cell polysome accumulation observed in the time-lapse microfluidic experiments (*Figure 1*) was not caused by a photobleaching effect.

For population snapshots, nucleoids were typically imaged using DAPI (rather than a fluorescent fusion to HupA), indicating that polysome accumulation at mid-nucleoid was independent of the DNA labeling method. Importantly, and consistent with our prediction, the richer the growth conditions (i.e. the larger the average cell area), the earlier the polysome accumulation and the nucleoid splitting occurred in the division cycle based on relative cell lengths (*Figure 2A*, *Figure 2—figure supplement 1*). In addition, the polysome accumulation at mid-cell was more pronounced in nutrient-rich media (e.g. M9malaCAAT) compared to nutrient-poor ones (e.g. M9mann) where the nucleoid segregated later in the division cycle (*Figure 2A*, *Figure 2—figure supplement 1*).

To quantify these phenotypes across strains and nutrient conditions, we extracted and correlated population-level polysome and nucleoid statistics. We found a strong correlation (Spearman correlation $\rho$ =–0.85) between the amplitude of the average ribosomal signal accumulation and the average nucleoid signal depletion at mid-cell (*Figure 2B*). Since the average cell area (colormap in *Figure 2B*) correlates with the growth rate of the population (*Schaechter et al., 1958*), this plot also confirmed that faster-growing populations displayed stronger polysome accumulation and greater DNA depletion at mid-cell on average (*Figure 2B*). This was also observed across cells within a population under the same nutritional condition ( $\rho$ =–0.84, *Figure 2—figure supplement 2*).

Both the average proteome fraction dedicated to ribosomes and the average fraction of ribosomes engaged in translation are known to correlate with growth rate across nutrient conditions (*Chure and Cremer, 2023*; *Dai et al., 2016*; *Dourado and Lercher, 2020*; *Hu et al., 2020*; *Molenaar et al., 2009*; *Scott et al., 2014*; *Si et al., 2017*). Thus, a role for polysome production in nucleoid segregation may provide a mechanistic link between growth rate and the relative timing of nucleoid splitting.

## Spatial polysome asymmetry correlates with nucleoid positioning

A surprising result was the apparent higher polysome accumulation at mid-cell relative to the cell pole regions (*Figures 1B and C, and 2A*). This was not artificially created by the smaller cytoplasmic volumes at the cell poles or constriction sites due to membrane curvature (*Figure 2—figure supplement 3*). It

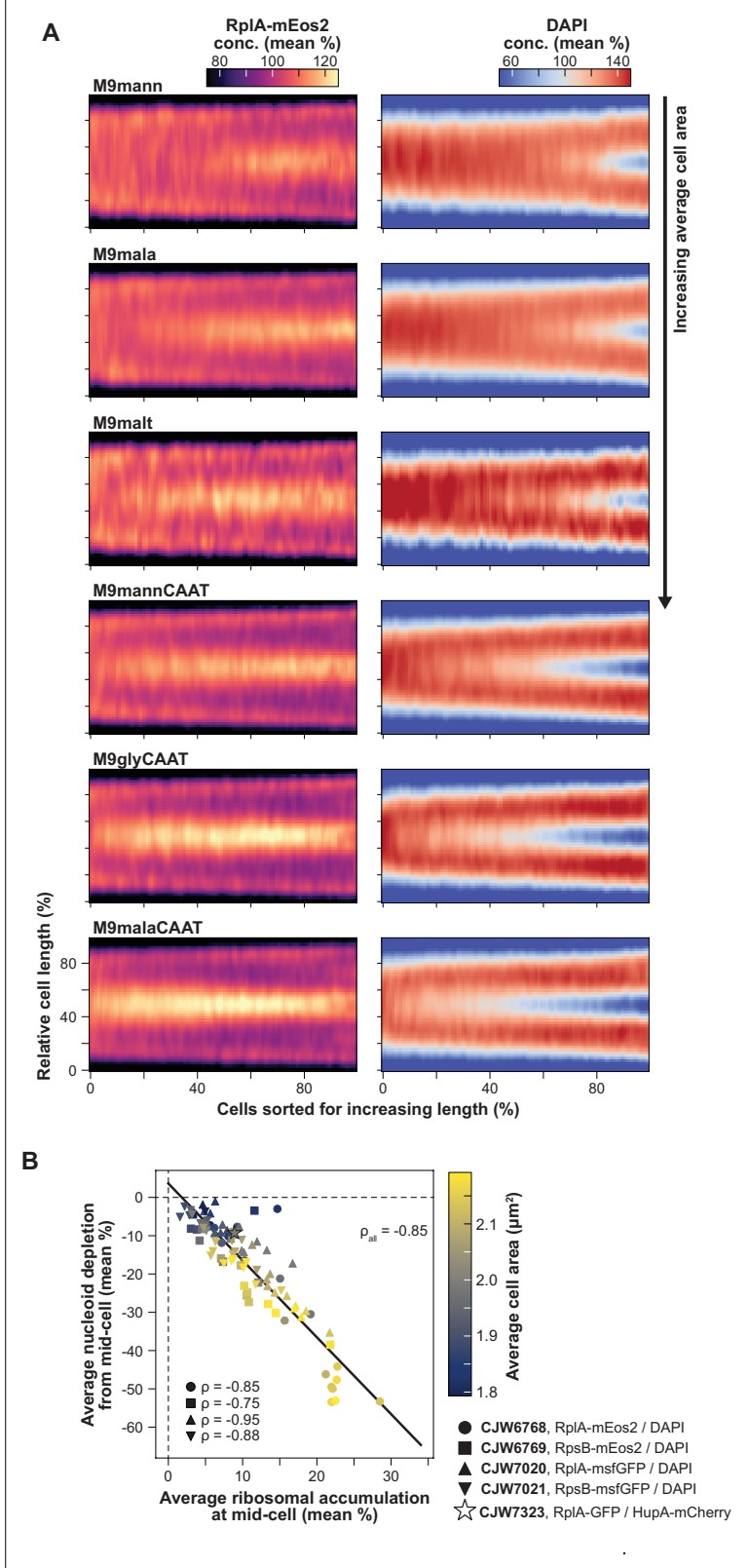

**Figure 2.** Correlation of the extent and relative timing of polysome accumulation with nucleoid segregation at the population level. (**A**) RplA-mEos2 and DAPI (scaled by the whole cell average) demographs constructed from snapshots of DAPI-stained CJW6768 cells (815–2771 cells per demograph) expressing RplA-mEos2 and growing in different nutrient conditions (see *Supplementary file 1* for abbreviations). The demographs were arranged from

*Figure 2 continued on next page*

*Figure 2 continued*

smallest (M9mann, top) to biggest average cell area (M9malaCAAT, bottom). Additional demographs for different ribosomal reporters and nutrient conditions are shown in *Figure 2—figure supplement 1*. (**B**) Correlation between average polysome accumulation and average nucleoid depletion at mid-cell for all tested strains (Spearman $\rho_{all}$ = -0.85, p-value <10$^{-10}$) with different ribosomal and nucleoid reporters, and within each strain (–0.75 ≤ Spearman $\rho_{strain}$ ≤ -0.95, p-values <10$^{-3}$) across nutrient conditions. A linear regression was fitted to all the data.

The online version of this article includes the following figure supplement(s) for figure 2:

**Figure supplement 1.** Demographs of scaled ribosome and nucleoid fluorescence for strains with different ribosome markers and under various nutrient conditions.

**Figure supplement 2.** Correlation of the extent and relative timing of polysome accumulation with nucleoid segregation at the single-cell level.

**Figure supplement 3.** Calculation of RplA-msfGFP concentration after cell curvature correction.

was also not caused by a photobleaching artefact during the timelapse microscopy since it was also observed from snapshot images, particularly under nutrient-rich conditions (*Figure 2—figure supplements 1 and 3*). The uneven distribution of polysomes suggested limited diffusion-driven equilibration of polysome concentration between the DNA-free regions. At division, such a disequilibrium could lead to a higher concentration of polysomes at the new cell pole relative to the old pole in daughter cells through inheritance. To examine this possibility, we went back to the microfluidic experiments in which we traced cell lineages from mother to daughter cells (*Figure 1—figure supplement 1A*), determined the pole identity (new vs. old) of each tracked cell (*Figure 1—figure supplement 4B*), and compared the polysome accumulations between the new and old poles (see Methods and *Figure 3—figure supplement 1*). Old mother cells (located at the end of the microfluidic channels) and their daughters were excluded from our analysis to avoid cell aging effects (*Chao et al., 2024*; *Coquel et al., 2013*; *Koleva and Hellweger, 2015*; *Lapińska et al., 2019*; *Lindner et al., 2008*; *Proenca et al., 2019*). We found that newborn cells had more polysomes at the new pole compared to the old one on average (*Figure 3A*), consistent with the notion that polysomes do not rapidly equilibrate in concentration between DNA-free regions through diffusion.

Across cells, the polysome distribution asymmetry between cell poles negatively correlated with the position of the nucleoid in newborn daughter cells (*Figure 3B*, cells always oriented with the new pole to the right). The x and y intercepts of a fitted linear regression to the data were not zero, indicating that a nucleoid positioned precisely at the cell center did not equate with an even distribution of polysomes between poles. Instead, newborn cells with centrally located nucleoids tended to have more polysomes at the new pole compared to the old pole (example #1 in *Figure 3B and C*), whereas cells born with symmetric polysome enrichments between poles tended to display an off-center nucleoid closer to the new pole (example #2 in *Figure 3B and C*).

## Spatial polysome asymmetry correlates with asymmetric nucleoid compaction in newborn cells

Construction of average 2D cell projections of the RplA-GFP signal in cells sorted based on their relative timing to cell division confirmed the polysome asymmetry between poles in newborn cells (*Figure 3D* and *Video 2*). Strikingly, the corresponding 2D projections of the HupA-mCherry signal revealed another spatial asymmetry, this time, in the average DNA mass distribution along the nucleoid length (axial asymmetry). The average HupA-mCherry signal concentration was higher toward the new pole right after birth or, correspondingly, toward the middle of the cell (future new pole) in the period prior to division (*Figure 3D* and *Video 2*). Quantification of this axial asymmetry in newborn cells revealed that the HupA-mCherry concentration is ~20% higher toward the new pole on average (*Figure 3E*), suggesting that the DNA density is uneven along the nucleoid length. This nucleoid mass asymmetry emerged late in the nucleoid migration cycle, typically before cell division such that it was inherited by newborn cells (*Figure 3D*).

To examine if features of polysome accumulations (e.g. position, amplitude, or fraction of cell length covered) correlate with the asymmetric nucleoid density and its variability among cells, we used a linear mixed-effects model (see Methods). We found that in newborn cells, the positions of polysome enrichments in the old-pole and mid-cell regions significantly correlated with the HupA-mCherry

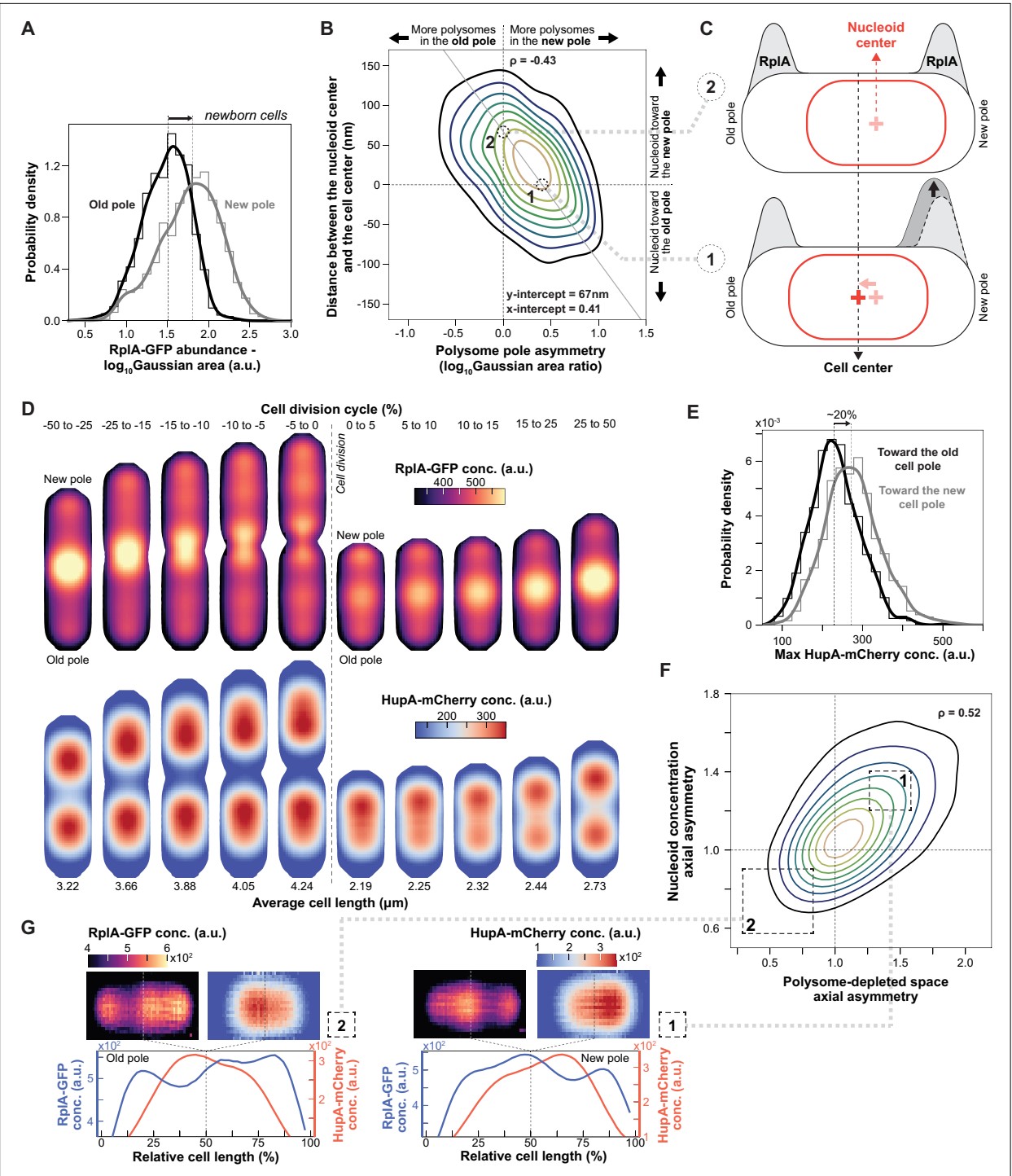

**Figure 3.** Correlations between polysome and nucleoid asymmetries. (**A**) Distributions of RplA-GFP concentration in the new (gray) and the old (black) pole regions of newborn cells (0–2.5% of the cell division cycle, n=912 cell division cycles). The histograms were smoothed using Gaussian kernel density estimations. (**B**) Correlation (Spearman $\rho$ =−0.41, p-value <$10^{-10}$) between the polar polysome asymmetry and the position of the nucleoid centroid around the cell center of cells at the beginning of the division cycle (0–10%, n=1179 cell division cycles). The isocontour plot consists of nine levels with a lower data density threshold of 25%. The polar polysome profiles for cells at the x and the y intercept, indicated by the numbers 1 and 2 respectively, is schematically illustrated in the next panel. A linear regression (solid gray line) was fitted to the data. (**C**) Schematic illustrating the effects of the relative polysome abundance between the poles on the position of the nucleoid. (**D**) Average 2D projections of the RplA-GFP and HupA-mCherry concentration (conc.) at different cell division cycle intervals (~9440–47240 cell images per cell division cycle interval from 4122 cell division cycles). The dotted line indicates the boundary between two cell division cycles. (**E**) Density plot comparing the distribution of the HupA-mCherry maximum concentration

*Figure 3 continued on next page*

*Figure 3 continued*

toward the new pole (gray) to that toward the old pole (black) in newborn cells (0–2.5% of the division cycle, n=912 cell division cycles). The histograms were smoothed using Gaussian kernel density estimations. (**F**) Correlation (Spearman $\rho$ =0.52, p-value <$10^{-10}$) between the nucleoid density asymmetry and the relative availability of polysome-free space between the two cell halves early in the cell division cycle (0–10% of the division cycle, n=2150 cell division cycles). The isocontour plot consists of nine levels with a lower data density threshold of 25%. Values above 1 on the x-axis indicate more polysome-free space toward the new pole, and values below 1 correspond to cells with more polysome-free space toward the old pole. On the y-axis, values above 1 indicate higher DNA density toward the new pole and values below 1 indicate higher DNA density toward the old pole. (**G**) Average 2D projections of newborn cells (0–10% into the cell division cycle) from the lower-left quartile in panel C (region 2, n=223 cell division cycles) and the upper right quartile in panel C (region 1, n=557 cell division cycles) and their 1D intensity profiles.

The online version of this article includes the following figure supplement(s) for figure 3:

**Figure supplement 1.** Quantification of the polysome and nucleoid asymmetries using fitted Gaussian functions.

**Figure supplement 2.** Correlations between the polysome distribution statistics and the nucleoid compaction asymmetries.

density asymmetry (*Figure 3—figure supplement 2*). Guided by this finding, we hypothesized that the positions of the accumulating polysomes along the cell length determine the space available for the chromosome to occupy and thereby dictate the DNA density distribution along the nucleoid. To examine this hypothesis, we combined the correlated polysome accumulation characteristics into a compound statistic that describes the relative polysome-free space between two cell halves (see Methods). Compared to other polysome statistics, the relative availability of cell space depleted of polysomes (RplA-GFP signal) correlated most strongly with the asymmetric distribution of DNA (HupA-mCherry signal) in newborn cells (*Figure 3F*). In other words, the DNA concentration was higher in cell regions with more space available between polysome enrichments. In most (~60%) cells, the distance between polysome enrichments, and thus the DNA concentration along the nucleoid, was greater between the mid-cell position and the new pole (i.e. x and y values >1, example #1 in *Figure 3F and G*). The opposite pattern was true for a small fraction (~14%) of cells, where the larger distance between polysome enrichments was located between the mid-cell position and the old pole (x and y values <1, example #2 in *Figure 3F and G*). Thus, polysome asymmetry correlates with asymmetric nucleoid compaction.

## A minimal reaction-diffusion model generates experimentally observed cellular asymmetries and growth rate-dependent nucleoid segregation

Our single-cell correlative studies were consistent with the exclusion between polysomes and DNA contributing to nucleoid compaction and segregation. However, it remained unclear whether the same mechanism could also explain the growth rate-dependent trends and cellular asymmetries that we observed (*Figures 2 and 3*). Therefore, we built a minimal reaction-diffusion model (see Methods) to describe the dynamics of polysomes and DNA during the cell cycle based on previous work (*Miangolarra et al., 2021*). Our model takes into account two ingredients important for nucleoid segregation: effective repulsion between DNA and polysomes from steric effects (described by the Cahn-Hilliard theory) and the nonequilibrium processes of polysome synthesis and degradation (described by linear reaction kinetics). The model is based on realistic parameters of polysome diffusion, production, and degradation (see Methods). We assume polysome production (i.e. mRNA synthesis and ribosome loading) to be uniform within the nucleoid and polysome degradation (i.e. mRNA decay) to be uniform across the entire cell (from pole to pole). These assumptions consider the most trivial scenarios (see Discussion for other scenarios). We also assume that the cell grows exponentially and that the nucleoid expands in size proportionally to the cell during growth, which has been experimentally verified (*Campos et al., 2014*; *Govers et al., 2024*; *Gray et al.,*

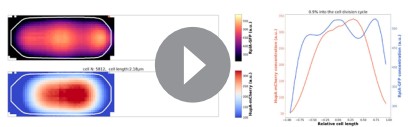

**Video 2.** Video showing the average subcellular distribution of ribosome and nucleoid signals from birth to division. Shown are 2D average RplA-GFP (top) and HupA-mCherry (bottom) projections from birth to division, with corresponding average intensity profiles (right). The cell projections are oriented from the old pole on the left to the new pole on the right. The average cell contour is also drawn. Ensemble data from 4122 division cycles of CJW7323 cells growing in M9gluCAAT are shown.

https://elifesciences.org/articles/104276/figures#video2

*2019*). Since *E. coli* grows along its long axis and polysomes do not readily diffuse around the nucleoid to equilibrate (*Figure 3A–D*, *Figure 2—figure supplement 3*), we reduced the problem to one dimension, the cell length. Importantly, the system is driven out of equilibrium by the continuous production and degradation of polysomes. It operates at a nonequilibrium steady state even at fixed cell length.

First, for simplicity and illustrative purposes, we considered the case of a non-growing virtual cell with the nucleoid initially spread throughout most of the cell to show that the repulsion between polysomes and DNA is sufficient to both demix (phase separate) these two cytoplasmic components and compact the nucleoid when the simulation reaches steady state (*Figure 4A*). The compaction force originates from the nucleoid/polysome steric repulsion (which drives the phase separation). The resulting higher concentration of polysomes on each side of the nucleoid produces a difference in osmotic pressure that condenses the nucleoid. This compaction force is consistent with drug experiments. Depletion of polysomes through inhibition of transcription with rifampicin leads to nucleoid expansion, whereas stabilization of polysomes through inhibition of ribosome translocation with chloramphenicol results in greater nucleoid compaction (*Bakshi et al., 2014*; *Cabrera et al., 2009*; *Farrar et al., 2025*; *Spahn et al., 2023*; *Spahn et al., 2018*; *Spahn et al., 2018*; *Stracy et al., 2015*; *Xiang et al., 2021*). In the case of chloramphenicol, fusion of nucleoids has been reported (*Bakshi et al., 2014*; *Spahn et al., 2023*). This phenomenon is expected to occur for nucleoids in close proximity at the time of chloramphenicol treatment. These nucleoids may touch through diffusion (thermal fluctuation) and fuse to reduce their interaction with the polysomes and minimize their conformational energy. Well-separated nucleoids typically did not fuse. Since division could still occur during chloramphenicol treatment, the lack of fusion between well-separated condensed nucleoids was more evident in filamenting cells inhibited for cell division by cephalexin (*Figure 4—figure supplement 1*).

Next, we tested whether adding cell growth to our model recapitulates the experimental observations. Simulations showed that polysomes accumulate in the middle of the nucleoid during growth (*Figure 4B* and *Video 3*, left panel). This is followed by the division of the nucleoid into two entities, which then move apart from each other as polysomes accumulate between them. Thus, the model provides a minimal mechanism for nucleoid segregation: At any given point, polysomes that form in the middle of the nucleoid have a lower probability of escaping the nucleoid through diffusion compared to polysomes born at the edge of the nucleoid (*Figure 4C*). Consequently, the polysome concentration rises monotonically towards the center of the nucleoid, with its level increasing with nucleoid length (quadratically in the quasi-steady-state limit). Once the mid-cell polysome concentration reaches a threshold (the spinodal concentration), phase separation occurs spontaneously (i.e. via spinodal decomposition), creating a new polysome-rich phase that splits the nucleoid in two. Compared to regions near the poles, this new phase has a higher polysome concentration and, therefore, a higher osmotic pressure. This pressure difference results in a net poleward force on the sister nucleoids that drives their migration toward the poles (*Figure 4B* and *Video 3*, left panel). Therefore, both phase separation (due to the steric repulsion described above) and nonequilibrium polysome production and degradation (which create the initial accumulation of polysomes around mid-cell) are essential ingredients for nucleoid segregation.

We wondered whether our simple model could also explain the correlation between growth rate and the relative timing of nucleoid segregation (*Govers et al., 2024*). Therefore, we performed simulations for different growth rates, matching the cell and nucleoid length at birth with population-level measurements (*Figure 4—figure supplement 2*). To initialize each simulation in a realistic fashion, we used the last timepoint (i.e. half of the predivisional cell) of the previous simulation as initial conditions, capturing the new/old pole identity as well as any cellular asymmetries inherited between generations. The relative timing of nucleoid splitting was measured as nucleoid depletion at mid-cell. We found that our model successfully captures the negative trend between the growth rate and the relative timing of nucleoid splitting. Simulated cells that grew faster also split their nucleoids earlier from birth to division, agreeing with population-level data (*Figure 4D* and *Video 3*). The same simulations also reproduced the relatively constant cell length at which nucleoid splitting occurs across different growth rates (*Figure 4—figure supplement 3*), which was previously discovered in population-level measurements (*Govers et al., 2024*). These phenomenological principles are expressions of the link between the absolute nucleoid length and the rate of polysome accumulation in the middle of the nucleoid, which is explained by our mechanistic model.

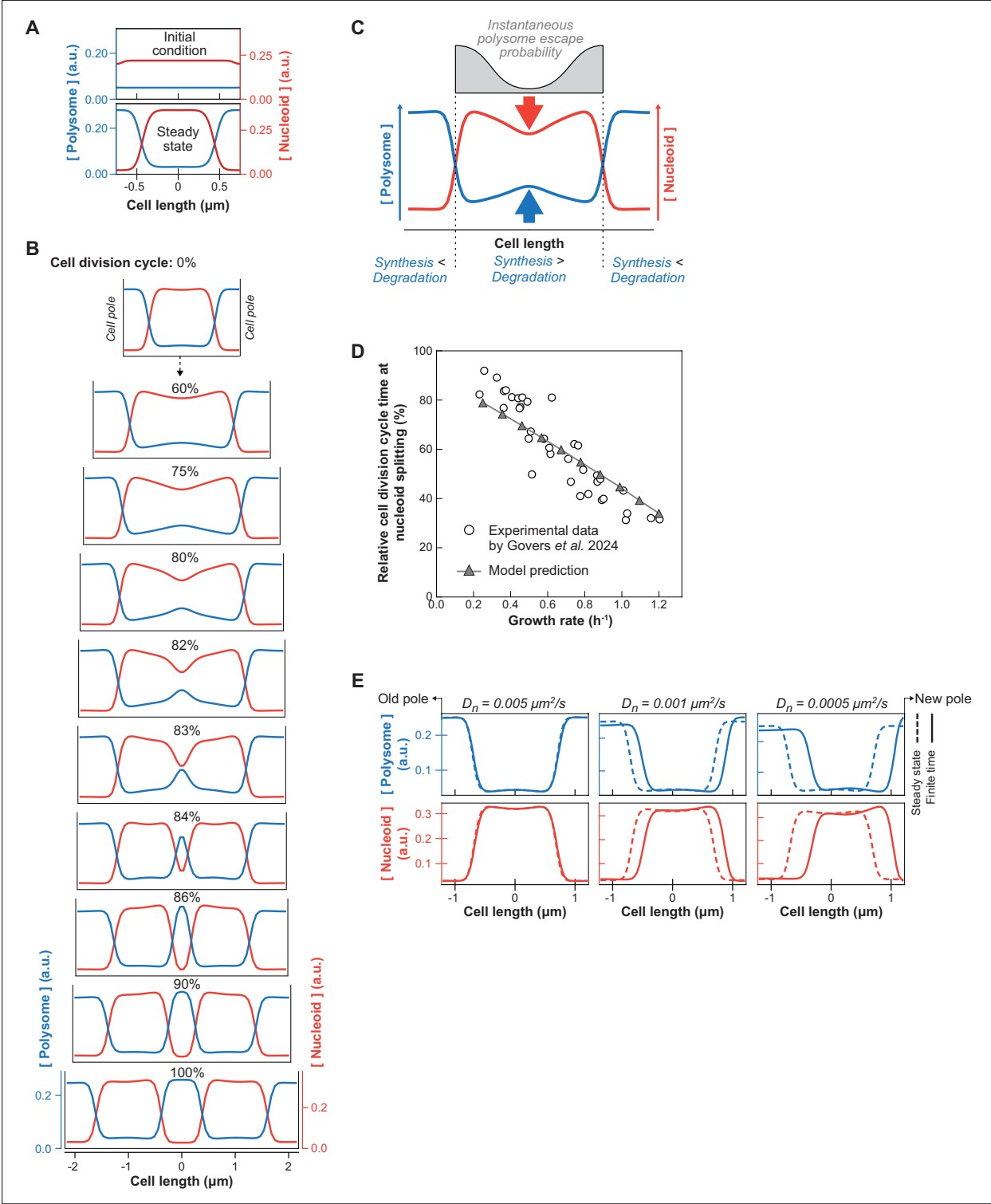

**Figure 4.** Simulation results of the reaction-diffusion model for different growth rates or nucleoid diffusion rates. (**A**) Simulation of a non-growing virtual cell, initialized with homogeneous polysome concentration and a nucleoid spread between the two poles ($t$=0 s). It reaches steady state with the nucleoid compacted at mid-cell at $t$=998 s. (**B**) 1D simulation of polysome (blue) and nucleoid (red) dynamics during slow growth (growth rate = 0.25 h$^{-1}$, $D_n$ = 0.001 μm$^2$/s, cell length at birth = 2.2 μm) at different relative cell division cycle timepoints. The simulation was initialized from the equilibrium polysome and nucleoid distribution (at 0%). (**C**) Schematic summarizing how polysomes accumulate in the middle of the elongating nucleoid, causing nucleoid splitting. (**D**) Correlation between the relative timing of nucleoid splitting and the growth rate as captured by our reaction-diffusion model ($D_n$ = 0.001 μm$^2$/s) across six growth rate bins. The cell and nucleoid lengths for each growth rate bin matched previously published population-averaged

*Figure 4 continued on next page*

*Figure 4 continued*

data (**Govers et al., 2024**). (**E**) Deviation between the steady state after infinite relaxation time (dashed curves) and the polysome or nucleoid profiles in newborn cells after one simulation round (solid curves) for increasing nucleoid diffusion constants. The simulations were performed for a growth rate of 0.57 h$^{-1}$, which is comparable to the average growth rate in our microfluidic experiments (**Figure 2—figure supplement 2A**).

The online version of this article includes the following figure supplement(s) for figure 4:

**Figure supplement 1.** Effects of chloramphenicol treatment on nucleoids in non-dividing cells.

**Figure supplement 2.** Determination of simulated cell growth parameters based on experimental measurements.

**Figure supplement 3.** Comparison of cell length at nucleoid splitting between simulations and experimental data.

**Figure supplement 4.** Simulation of an extended model with three different polysome species.

To examine a potential origin for the asymmetries in polysome distribution and nucleoid compaction that we observed (**Figure 3**), we examined the effect of the nucleoid diffusion coefficient $D_n$, which is a model parameter that describes how fast the nucleoid relaxes towards its equilibrium configuration. Large $D_n$ represents the quasi-steady-state limit where the nucleoid relaxation time scale is much shorter than the cell doubling time, and the nucleoid always assumes its (symmetric) equilibrium state. Conversely, small $D_n$ (i.e. slower relaxation time) can lead to asymmetric concentration profiles that will be inherited by the daughter cells. Consistent with this expectation, we found that for average growth rate (~ 70 min doubling time), a lower nucleoid diffusion coefficient results in a larger deviation from the equilibrium concentration profiles for the nucleoid and polysomes (dotted curves vs. solid curves, **Figure 4E**). In fact, at diffusion coefficients below 0.005 µm$^2$/s, the model (**Figure 4E**) reproduced the experimentally observed asymmetries, including the nucleoid position offset towards the new pole at birth, the higher polysome concentration at the new pole compared to the old one, and the asymmetric nucleoid compaction (**Figure 3**). Our minimal model thus suggests that the material properties of the nucleoid (e.g. stiffness) may contribute to the observed nucleoid and polysome asymmetries in *E. coli* (see Discussion).

## Polysomes accumulate at mid-nucleoid in DNA regions inaccessible to freely diffusing particles of similar sizes

In the model, the early polysome accumulation in the middle of the nucleoid is caused by the nonequilibrium processes of polysomes being born within the nucleoid while being degraded uniformly across the cell (due to mRNA turnover). It predicts that the early mid-nucleoid enrichment of polysome signal observed in our experiments is the product of such nonequilibrium processes rather than of polysomes simply diffusing into undetected DNA-free space. If this is correct, freely diffusing objects of similar sizes to polysomes should accumulate at mid-cell after polysomes accumulate there, i.e., after DNA-free space has been generated through nucleoid splitting. To test this expectation, we compared the average distribution of RplA-msfGFP with that of freely diffusing mCherry-labeled µNS particles from snapshot images of DAPI-stained cells (**Figure 5A**). These µNS particles consist of a fragment of a mammalian reovirus protein (**Broering et al., 2005**; **Broering et al., 2002**) that self-assembles into a particle, typically one per cell, when produced orthogonally in *E. coli* (**Parry et al., 2014**). They have sizes between 50 and 200 nm (**Parry et al., 2014**; **Xiang et al., 2021**) similar to polysomes (**Brandt et al., 2009**; **Slayter et al., 1968**) and are, therefore, largely excluded by the nucleoid mesh (**Xiang et al., 2021**). After sorting cells by length into four bins, the positions of mCherry-labeled µNS particles from approximately 2580 cells per bin were superimposed using the relative cellular coordinates to construct particle density maps. Cells were randomly oriented in this analysis (meaning that asymmetries between poles cannot be observed), as the pole identity cannot be assigned from snapshot images.

We found that in the shortest (i.e. newborn) cells, RplA-msfGFP-labeled polysomes had already accumulated within the nucleoid (**Figure 5B**, bin 1). In contrast, mCherry-µNS

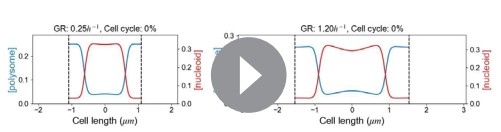

**Video 3.** Video showing simulated 1D profiles of polysome and nucleoid concentration during slow and fast cell growth. Simulations during slow (left) and fast (right) growth are shown. The simulations were initialized from the equilibrium configuration, with a compact symmetric nucleoid (Dn = 10$^{-3}$ µm$^2$/s) at the cell center.

https://elifesciences.org/articles/104276/figures#video3

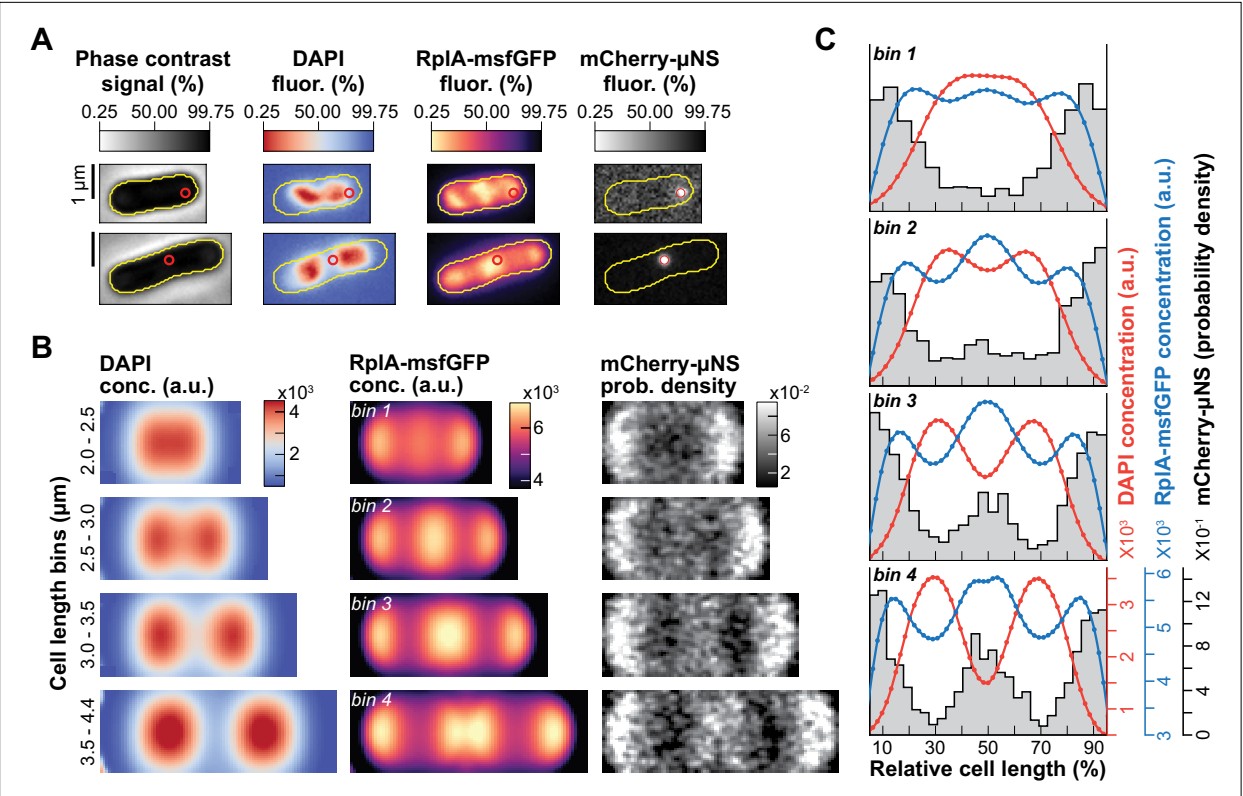

**Figure 5.** Comparison of the nonequilibrium polysome accumulation with freely diffusing particles. (**A**) Phase contrast and fluorescence (fluor.) images of two representative single cells (CJW7651). The red circles indicate the position of the mCherry-µNS particle in each cell. (**B**) Two-dimensional average cell projections of the DAPI concentration (conc.) and the RplA-msfGFP concentration, and 2D histogram of the mCherry-µNS particle density for four cell length bins of CJW7651 cells (~2580 cells per bin) grown in M9gluCAAT and spotted on an agarose pad containing the same medium. Since the cell pole identity cannot be inferred from snapshot images, pole assignment was random. (**C**) Average 1D profiles of the DAPI and RplA-msfGFP concentrations and the mCherry-µNS probability density.

particles were restricted to the cell poles and were not able to access the mid-cell region until after nucleoid splitting was clearly visible (*Figure 5B*, bins 2–4). This was also shown in the corresponding 1D average concentration and particle density profiles (*Figure 5C*). These observations support the notion that the early mid-nucleoid accumulation of RplA-msfGFP is caused by nonequilibrium effects associated with polysome synthesis and degradation rather than polysome diffusion into DNA-free space.

## Arrest of polysome production immediately stops nucleoid segregation, while polysome depletion gradually reverses it

Our correlative analyses and model (*Figures 1–4*) support the hypothesis that the interactions and ensuing exclusion between polysomes and nucleoids promote nucleoid segregation and macromolecular asymmetries along the cell length. To probe causality, we used two tests. The first one aimed to disrupt the proposed mechanism using rifampicin. Rifampicin treatment is known to homogenize ribosome distribution and expand the nucleoid over time through polysome depletion (*Bakshi et al., 2014*; *Dworsky and Schaechter, 1973*; *Koch and Gross, 1979*; *Pettijohn and Hecht, 1974*; *Xiang et al., 2021*). However, our hypothesis predicts a faster effect on nucleoid segregation. Blocking transcription should instantly reduce the rate of polysome production to zero, causing an immediate arrest of nucleoid segregation. Gradual depletion of the existing polysomes due to mRNA decay should then cause, on a slower time scale, a dissipation of the phase separation between DNA and ribosomes.

To test these predictions, we subjected cells growing in M9gluCAAT in microfluidic channels to two rounds of rifampicin treatment (*Figure 6A* and *Video 4*). Rifampicin resulted in growth rate inhibition

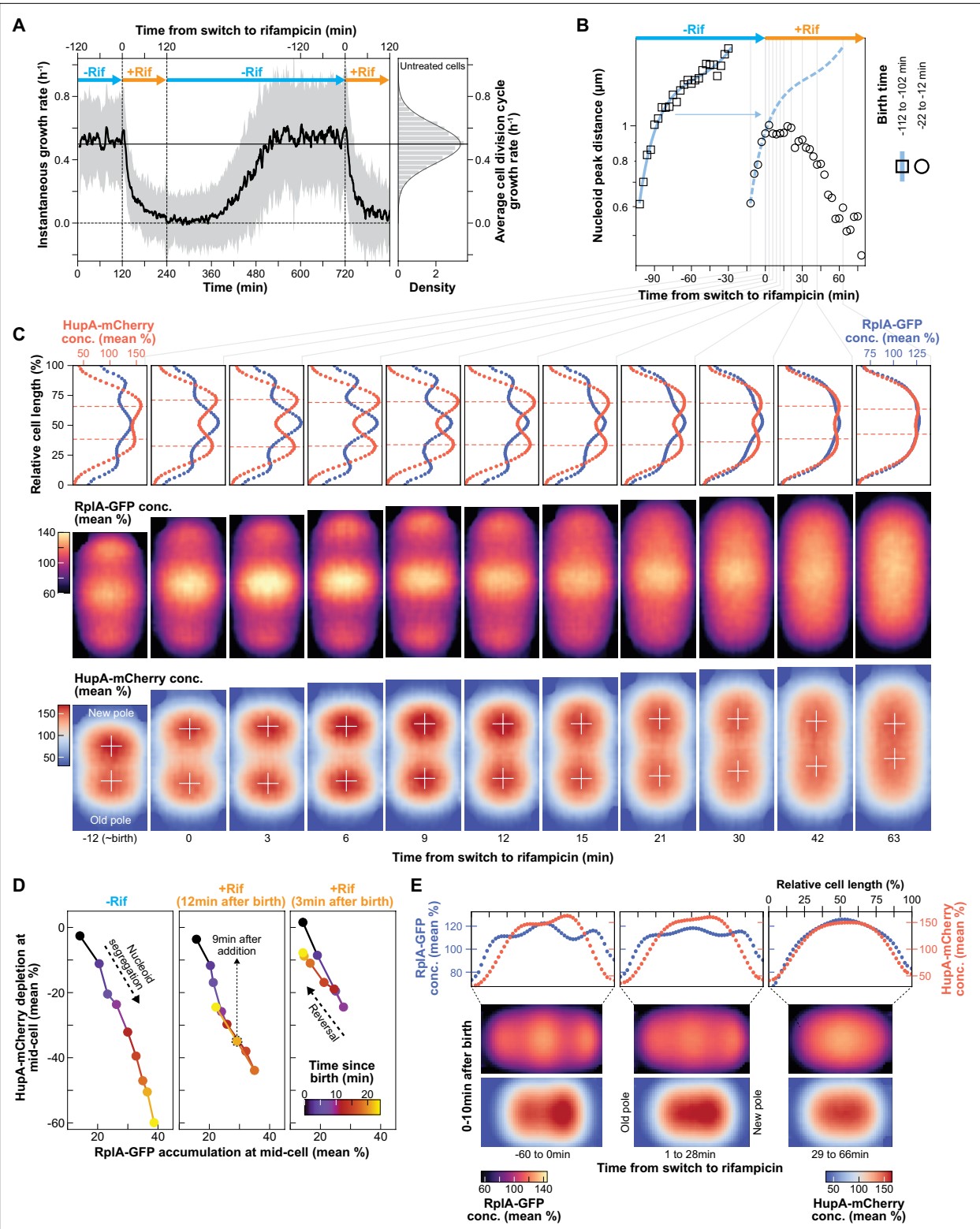

**Figure 6.** Effects of rifampicin treatment and polysome depletion on nucleoid segregation and compaction. (**A**) Plot showing the average instantaneous growth rate (mean ± SD shown by the solid black curve and gray shaded region, respectively) of a cell population (n=2629 cell division cycles) undergoing two rounds of rifampicin treatment in a microfluidic device supplemented with M9gluCAAT. The distribution of the average cell cycle growth rate of unperturbed populations is also shown on the right (n=4122 cell division cycles from a different microfluidics experiment). The solid horizontal line indicates the average growth rate. (**B**) Plot showing the average distance between nucleoid peaks for a population of cells (squares,

*Figure 6 continued on next page*

*Figure 6 continued*

114 cell division cycles) that were born (−112 to −102 min) and divided before the addition of rifampicin, and for a population of cells (circles, 112 cell division cycles) that were born just before (−22 to −12 min) and divided after the addition of rifampicin. A third-degree polynomial was fitted to the data from the unperturbed population (solid blue curve) and juxtaposed (dashed blue curve) with the data from the interrupted population. (**C**) Average 1D profile and 2D projections of the scaled (divided by the whole cell average concentration) RplA-GFP and HupA-mCherry signals for cells before and after rifampicin addition (n=112 cell division cycles). The red dashed horizontal lines in the 1D intensity profiles and the white crosses in the 2D profiles mark the nucleoid peaks. (**D**) Plot showing the RplA-GFP accumulation relative to the HupA-mCherry depletion at mid-cell from 0 to 24 min after birth (colormap) for cells that completed their division cycle before the addition of rifampicin (n=114 cell division cycles) and for cells that were subjected to rifampicin 12 min (n=112 cell division cycles) or 3 min (n=99 cell division cycles) after birth. (**E**) Average 1D and 2D scaled RplA-GFP and HupA-mCherry intensity profiles for newborn cells (0–10 min after birth) before (left, n=726 cell division cycles), just after (middle, n=367 cell division cycles), and much after (right, n=235 cell division cycles) rifampicin addition.

The online version of this article includes the following figure supplement(s) for figure 6:

**Figure supplement 1.** Phenotypic effects of transcription inhibition.

(*Figure 6A*) and changes in the nucleoid area and nucleoid-to-cell area (NC) ratio (*Figure 6—figure supplement 1*), as previously described (*Bakshi et al., 2014*; *Dworsky and Schaechter, 1973*; *Koch and Gross, 1979*; *Pettijohn and Hecht, 1974*; *Xiang et al., 2021*). In cells that completed their division cycle before antibiotic addition (squares and fitted blue curve in *Figure 6B*), the distance between the intensity peak of each sister nucleoid increased monotonically between birth and division, displaying the dynamics of normal, unperturbed nucleoid segregation. Cells born 22 min to 12 min before the treatment (circles in *Figure 6B*) experienced the same nucleoid segregation dynamics up to the time of rifampicin addition. Exposure to rifampicin led to the near-immediate arrest of nucleoid segregation (*Figure 6B*), consistent with our prediction.

Interestingly, the distance between the sister nucleoids remained the same for close to 30 min into rifampicin treatment, after which it started to decrease (*Figure 6B*). Average 1D and 2D cell projections of the scaled (divided by the whole cell average) RplA-GFP and HupA-mCherry concentration suggest that the time delay between the arrest in nucleoid segregation and its reversal is likely due to the compressible nature of the nucleoid, which has been demonstrated in vitro (*Pelletier et al., 2012*). As the polysomes in the middle of the cells started to visibly deplete (>6 min after rifampicin addition), the DNA signal expanded to fill the emerging available space without affecting the distance between the peaks of the sister nucleoids (white crosses, *Figure 6C*). This is consistent with the removal of a compaction force exerted by the accumulating polysomes on the soft nucleoid. About 30 min after rifampicin addition and further polysome depletion, the peak signals of the sister nucleoids (white crosses, *Figure 6C*) started migrating closer to each other. Eventually, after 1 hr of rifampicin treatment, when the RplA-GFP fluorescence was homogeneous, the two nucleoid objects fused into one (*Figure 6C*), consistent with the dissipation of phase separation.

These results support the notion that in untreated cells, polysome accumulation effectively exerts a force on the compressible nucleoid, which translates into its observed compaction and translocation (hence, segregation). Gradual polysome depletion through rifampicin treatment progressively decreased this effect, reversing the process. This reversal became obvious when we plotted the correlation between the relative polysome accumulation and nucleoid depletion at mid-cell for two cell lineages that experienced rifampicin at different times after birth (*Figure 6D*). Irrespective of their birth time (12 or 3 min before the addition of rifampicin), the negative correlation between the two variables was reversed ~9 min after the addition of the antibiotic, following the same path as for the untreated

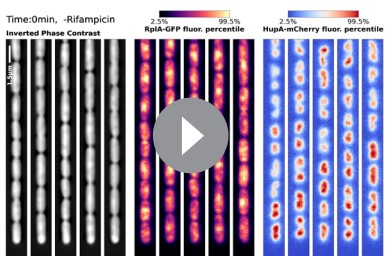

**Video 4.** Video showing the effects of rifampicin addition on cell growth, ribosome signal heterogeneity, nucleoid segregation, and nucleoid compaction. Examples of five microfluidic channels showing the corresponding inverted phase contrast, RplA-GFP, and HupA-mCherry signals (from left to right) of cells (CJW7323) growing within microfluidic channels. Rifampicin was added at 120 and 720 min. Each antibiotic treatment lasted 120 min.
https://elifesciences.org/articles/104276/figures#video4

cells but in the opposite direction (*Figure 6D*). Polysome depletion during rifampicin treatment also resulted in correlated loss of asymmetric nucleoid compaction in newborn cells (*Figure 6E*). Altogether, these results support the notion that the asymmetric accumulation of polysomes results in an anisotropic force that asymmetrically compacts and segregates nucleoids.

## Ectopic polysome production redirects nucleoid dynamics

Our second approach to test causality experimentally was to redirect polysome production away from the chromosome to achieve polysome accumulation at an ectopic site. To achieve polysome accumulation at ectopic sites in experiments, we overexpressed a useless protein (mTagBFP2) for the cell from a T7 promoter on a multi-copy pET28 plasmid (*Figure 7A*). The resulting CJW7798 strain also carried the ribosome (RplA-msfGFP) and DNA (HupA-mCherry) markers. We reasoned that high expression of BFP2 from the plasmid would slow polysome production within the nucleoid and create polysome accumulations at ectopic cellular locations through the recruitment of ribosomes to plasmid transcripts. This, in turn, should affect nucleoid dynamics according to our model.

Plasmid expression of mTagBFP2 was induced by the addition of 100 µM IPTG and expression of a chromosomally encoded T7 RNA polymerase (*Figure 7B*). The gradual increase of mTagBFP2 fluorescence in the cells was associated with a concomitant decrease in cell growth rate (*Figure 7C*), consistent with reduced gene expression from the chromosome. We verified by flow cytometry that induction did not block DNA replication. IPTG-induced cells displayed a similar scaling relationship between the intensity of DNA (labeled with DRAQ5) and cell size (side scatter area) to uninduced cells (*Figure 7—figure supplement 1*).

We found that induced cells displayed various patterns of polysome accumulations (*Video 5*), presumably due to stochastic clustering of plasmids in DNA-free regions as previously reported for multi-copy plasmids devoid of DNA partitioning genes (*Hsu and Chang, 2019*; *Reyes-Lamothe et al., 2014*; *Yao et al., 2007*). Importantly, the ectopic accumulations of polysomes had a drastic effect on nucleoid dynamics. In some cells, polysomes accumulated at one pole instead of the middle of the nucleoid, preventing the nucleoid from splitting (*Figure 7D*). Expansion of the polysome accumulation at a pole effectively pushed the nucleoid toward the opposite pole of the cell. In other cells, polysome accumulation occurred between sister nucleoids, but did not relocate to the segregated nucleoids at the ¼ and ¾ cell positions. Rather, polysome accumulation persisted and expanded between the sister nucleoids, effectively further pushing them apart (*Figure 7E*). Both of these phenotypes were reproduced by our model when we intentionally caused an accumulation of polysomes, either at a pole or between segregated nucleoids (*Figure 7—figure supplement 2*). In our experiments, we also observed filamenting cells with correlated polysome and nucleoid dynamics that changed in time (*Figure 7F*), resulting in transient events of nucleoid fusion, splitting, or changes in migration direction depending on where polysomes accumulated. *Video 5* shows additional examples of such dependency.

Upon induction, we observed the appearance of diffuse mTagBFP2 fluorescence (*Figure 7B*). We also observed the formation of inclusion bodies (bright phase contrast) that typically remained at a pole or sometimes moved along the edge of a growing polysome accumulation (*Figure 7F*). After a long period of IPTG induction (>8 hr), polysome accumulation eventually decreased, leading to nucleoid decompaction (*Figure 7—figure supplement 3*).

This experiment effectively decoupled polysome accumulation from cell growth. By redirecting a substantial fraction of chromosome gene expression to a single plasmid-encoded gene, we reduced the rate of cell growth but still created a large accumulation of polysomes at an ectopic location. This ectopic polysome accumulation was sufficient to affect nucleoid dynamics in a correlated fashion. Altogether, these results support the notion that ectopic polysome accumulation drives nucleoid dynamics.

## Cell width enlargement leads to nucleoid splitting along the incorrect cell axis and to the fusion of polysome accumulations from distinct DNA-free regions

A previous study on *Bacillus subtilis* L-forms suggests that the width of the cell is also important for nucleoid segregation (*Wu et al., 2020*). In that study, L-forms, which are spherical cells stripped of their cell wall, were squeezed into narrow microfluidic channels of similar width to the diameter of

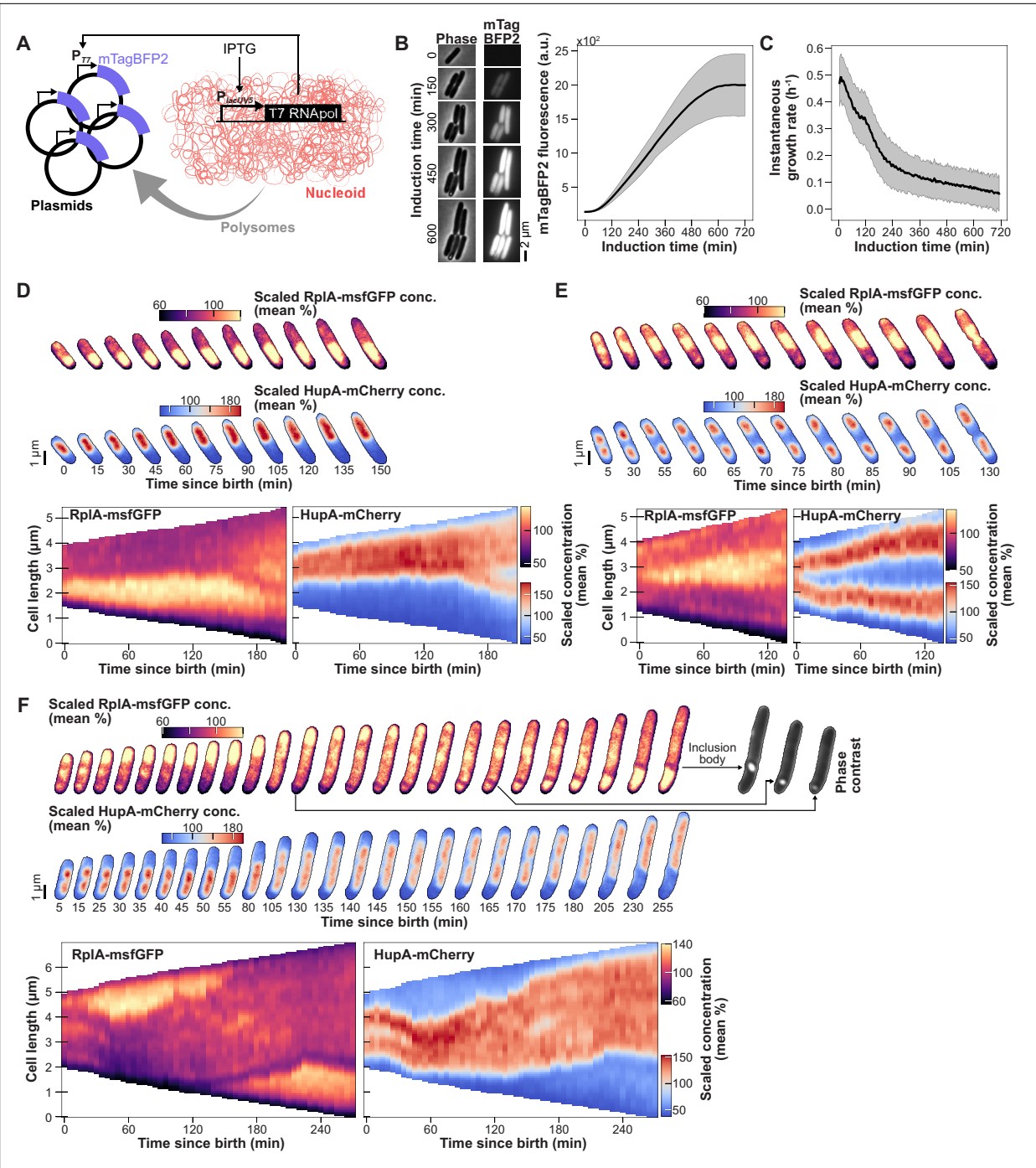

**Figure 7.** Effects of ectopic polysome accumulation on nucleoid dynamics. (**A**) Schematic summarizing the experiment. (**B**) Representative phase contrast and mTagBFP2 fluorescence images at different times after induction with IPTG (100 μM) are shown, next to a plot showing the mTagBFP2 fluorescence of the entire population (mean ± SD, n=3624 cell trajectories) over time. (**C**) Plot showing how instantaneous growth rate (mean ± SD, n=3624 mTagBFP2 induction trajectories) decreases following induction of mTagBFP2 synthesis. (**D–F**) Representative kymographs and images of the normalized (divided by the whole cell average) RplA-msfGFP and HupA-mCherry fluorescence signals in cells (CJW7798) born during mTagBFP2 over-expression. (**F**) Phase contrast images are shown to illustrate the formation of inclusion bodies (see also *Figure 7—figure supplement 3*). Additional cell examples are shown in *Video 5*.

The online version of this article includes the following figure supplement(s) for figure 7:

**Figure supplement 1.** DNA content during protein overexpression from plasmids.

**Figure supplement 2.** Simulations of ectopic polysome formations away from the nucleoid.

**Figure supplement 3.** Phenotypic effects of prolonged protein overexpression from plasmids.

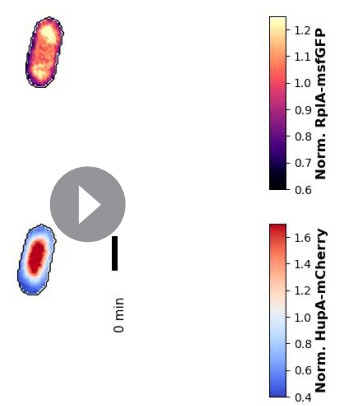

**Video 5.** Video showing the effects of ectopic polysome formation on nucleoid dynamics over time. The RplA-GFP and HupA-mCherry fluorescence normalized (norm.) by the average cell fluorescence are shown, together with their corresponding phase contrast image, for multiple cells (CJW7798) in succession following induction of mTagBFP2 expression from a T7 promoter on a multi-copy plasmid. The scale bar indicates 1 μm, and the time since cell birth is shown in minutes. The cell contours indicate the boundaries of the cell masks obtained by cell segmentation of the corresponding phase contrast images.

https://elifesciences.org/articles/104276/figures#video5

walled rod-shaped cells. Growth in these channels resulted in elongated cells with improved efficiency in nucleoid segregation (*Wu et al., 2020*). In normal (walled) cells, the nucleoid is kept close to the cytoplasmic membrane across the cell width, likely due to transertion (co-transcriptional translation and translocation of membrane and secreted proteins) (*Bakshi et al., 2014*; *Spahn et al., 2023*; *Youngren et al., 2014*). We reasoned that this geometric constraint reproduces some important aspects of our simple model. The rod-shaped cell morphology ensures that the nucleoid grows exclusively along one dimension, the cell length. Furthermore, the nucleoid attachment to the membrane along this radial cell axis, through transertion, may act as a diffusion barrier and limit polysome exchange between distinct DNA-free regions.

To test these ideas, we first treated cultures with A22 to inhibit cell width control through the inactivation of MreB (*Bean et al., 2009*; *Iwai et al., 2002*). This resulted in cells with a polysome phase at the cell center surrounded by a nucleoid phase around the cell periphery (*Figure 8—figure supplement 1*, top). The peripheral nucleoid localization is likely due to membrane attachment through transertion, as previously shown (*Spahn et al., 2023*). To further increase the cell width (>2.5 fold), we exposed cells to the cell division inhibitor cephalexin in addition to A22 (*Figure 8A*, *Figure 8—figure supplement 1*, bottom). As a control, we showed that cephalexin treatment alone resulted in filamentous cells (of constant cell width) with multiple nucleoids separated by polysome accumulations (*Figure 8B*), consistent with previous reports (*Chai et al., 2014*; *Gray et al., 2019*; *Thappeta et al., 2024*). In cells treated with both drugs, we observed two types of subcellular rearrangements. In smaller cells, which started with a single nucleoid, drug treatment resulted in a single large polysome accumulation at the cell center, with the nucleoid displaying a toroidal shape at the cell periphery (*Figure 8C* and *Video 6*). In longer cells with two segregated nucleoids, the nucleoids expanded and were often aberrantly segregated along the cell width concomitant with polysome accumulation at the site of nucleoid splitting (white arrowheads, *Figure 8D*). Consequently, a polysome 'bridge' was formed between the polysome accumulations flanking the nucleoid. These polysome bridges resulted in a characteristic cross-like polysome pattern, marking the two axes (longitudinal and radial) of nucleoid segregation (*Figure 8E*). Cell width enlargement led to the coalescence of polysome accumulations toward the cell center and the fusion of nucleoids around the cell periphery (*Figure 8C–E*, *Figure 8—figure supplement 1*, bottom). As a result, the nucleoids and polysome accumulations decreased in number while increasing in size in cephalexin/A22-treated cells compared to cephalexin-treated cells with the same cell area distribution but normal cell width (*Figure 8F*). These results suggest a critical role for cell width regulation in limiting the diffusion of polysomes around the nucleoid, thereby promoting nucleoid segregation specifically along the cell length.

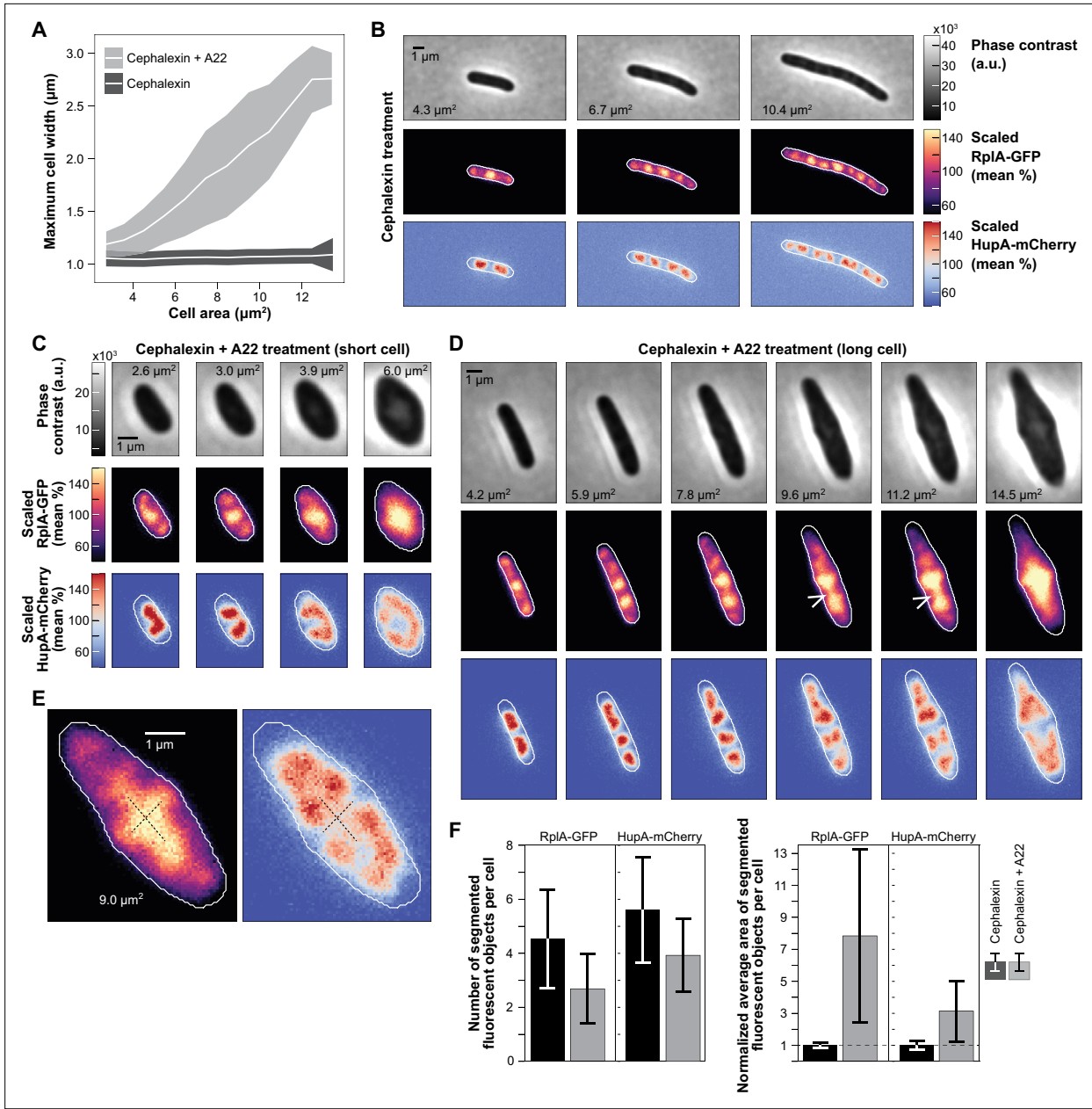

**Figure 8.** Effects of cell width increase on polysome and nucleoid dynamics. (**A**) Comparison of the cell width increase during cell growth between CJW7323 cells treated with cephalexin (mean ± SD, 360 cell growth trajectories, 418–1511 segmented cells per bin) and cells treated with both cephalexin (50 µg/mL) and A22 (4 µg/mL) (gray, mean ± SD, 309 cell growth trajectories, 51–1684 segmented cells per bin). The same cell area bins are compared between the two populations. (**B**) Phase contrast and fluorescence images of a representative cephalexin-treated cell expressing RplA-GFP and HupA-mCherry. (**C**) Same as B but for a short cell growing in the presence of A22 and cephalexin. (**D**) Same as C but for a longer cell. The white arrowheads indicate the polysome bridges that connect polysome accumulations between two DNA-free regions. Additional examples are shown in *Video 6* (**E**) Representative fluorescence images of RplA-GFP and HupA-mCherry in a cell treated with A22 and cephalexin. The dotted lines indicate the representative cross-like polysome accumulation, which forms during the fusion of the polysome accumulations towards the center (see also *Video 6*). (**F**) Comparison (mean ± SD) of the segmented polysome accumulations and nucleoid objects between A22 +cephalexin (150 sampled segmented cells from 47 growth trajectories) and cephalexin-treated (150 sampled segmented cells from 100 growth trajectories) cells. The polysome and nucleoid areas per cell were normalized by the population-average statistic from cephalexin-treated cells. All differences between the two populations are statistically significant (Mann-Whitney p-value $<10^{-10}$).

The online version of this article includes the following figure supplement(s) for figure 8:

**Figure supplement 1.** Phenotypic effects of cell growth under A22 and cephalexin treatment.

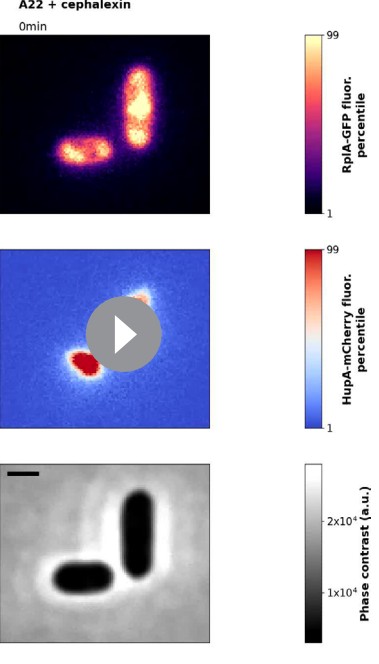

**A22 + cephalexin**
0min

RplA-GFP fluor. percentile

HupA-mCherry fluor. percentile

Phase contrast (a.u.)

**Video 6.** Video showing how the loss of cell width confinement due to cephalexin and A22 treatment affects ribosome and nucleoid distributions over time. Fluorescence images of RplA-GFP and HupA-mCherry fluorescence (fluor.) normalized by the average cellular fluorescence images, together with their corresponding phase contrast images, are shown for cells (CJW7323) following treatment with A22 (4 µg/mL) and cephalexin (50 µm/mL). The scale bar indicates 1 µm. The time since drug addition is shown in minutes.

https://elifesciences.org/articles/104276/figures#video6

## Discussion

### The flow of genetic information intrinsically couples nucleoid segregation to cell growth

This study provides experimental and theoretical evidence (*Figures 1–8*) that polysome production within the nucleoid—an inherent product of chromosomal gene expression—contributes to nucleoid segregation and positioning in *E. coli* cells. This may also be true in other bacteria, as reduced or abrogated transcription via gene deletion or antibiotic treatment causes chromosome segregation defects in *Streptococcus pneumoniae* (*Kjos and Veening, 2014*) and *Bacillus subtilis* (*Dworkin and Losick, 2002*).

An appealing feature of this proposed model is that polysomes inherently integrate the rate of nucleoid segregation with that of gene expression and cell growth. The concentration of polysomes and their rate of accumulation in the cell directly reflect the transcriptional and translational activities in the cell (*Balakrishnan et al., 2022*). The higher the concentration of polysomes, the faster the growth rate becomes. Thus, in our proposed model, an increase in polysome concentration not only leads to more protein synthesis and faster cell growth but also results in faster nucleoid segregation. The reverse is true for a decrease in polysome concentration, inherently coupling these processes without the help of a dedicated regulatory system. Such coupling was observed across isogenic cells with variable growth rates in the same nutrient condition (*Figure 2—figure supplement 2*) as well as across nutrient conditions that led to a wide range of growth rates (*Figure 2*, *Figure 2—figure supplement 1*).

### Directional nucleoid splitting requires DNA/polysome exclusion and cell width control

Polysomes form within nucleoids due to chromosomal gene expression, which, together with polysome turnover due to mRNA decay, creates an out-of-equilibrium system (*Figure 4*; *Miangolarra et al., 2021*). Redirecting polysome formation to plasmid gene expression leads to ectopic polysome accumulation that is sufficient to alter nucleoid dynamics (*Figure 7* and *Video 5*).

In normal cells, the effective force that segregates nucleoids appears to be linked to the propensity of the chromosomal meshwork and polysomes to separate from each other. Mutual exclusion between these two cytoplasmic components is, at least in part, caused by steric repulsion. While ribosomes or ribosomal subunits freely diffuse across the cell unobstructed by the presence of the nucleoid (*Bakshi et al., 2012*; *Sanamrad et al., 2014*), the larger polysomes are impeded by the chromosomal mesh based on size considerations alone (*Xiang et al., 2021*). Modeling studies have suggested that such steric hindrance between large crowders (polysomes) and a polymeric meshwork (chromosome) can result in phase separation and polymer compaction (*Bakshi et al., 2014*; *Castellana et al., 2016*; *Miangolarra et al., 2021*; *Mondal et al., 2011*; *Wu et al., 2019a*). Theoretically, mRNAs alone are large enough to phase separate from DNA, though to a lesser degree than polysomes (*Miangolarra et al., 2021*). It is also possible that phase separation between nucleoids and polysomes (or mRNAs)

involves non-steric interactions such as electrostatic repulsion between the negatively charged DNA and RNA (mRNA and rRNA), as previously hypothesized (*Joyeux, 2015*). Such steric and non-steric interactions may contribute to the effective poor solvent quality of the polysome-rich cytoplasm for the chromosome (*Xiang et al., 2021*). Future research will be necessary to elucidate the precise nature of the interaction between chromosomes and mRNAs, whether individually or in complex with ribosomes.

We found that the width of the cell controls the exclusion dynamics between polysomes and nucleoids by directing nucleoid growth and segregation along a single cellular dimension, cell length (*Figure 8*). The close proximity of the nucleoid to the membrane, presumably due to transertion (*Bakshi et al., 2014*; *Rabinovitch et al., 2003*; *Spahn et al., 2023*) effectively limits the diffusion and fusion of polysome accumulations from distinct DNA-free regions, which leads to alternating enrichments of polysomes and DNA along the length of elongating cells (*Figure 8B*). Limited polysome diffusion around the nucleoids is consistent with the observed differences in ribosome concentrations between DNA-free regions (*Figure 3A, B and D*). Otherwise, we would expect polysome enrichments on each side of the nucleoids to rapidly equilibrate in concentration. Diffusion limitation around nucleoids is consistent with our previous report that polysomes diffuse much faster over short distances (within DNA-free domains) than long distances (across DNA-free domains) (*Gray et al., 2019*). Control of cell width appears vital to uphold this constraint and promote that nucleoid grows along a single-cell axis. This explains the drastic improvement in nucleoid segregation of wall-less cells (L-forms) when compressed to normal width (*Wu et al., 2020*). It highlights the importance of cell width regulation and suggests that nucleoid segregation may have imposed an evolutionary constraint on cell width control.

## Nucleoid segregation likely involves multiple factors

The model shows that the most trivial case of uniform production of polysomes (i.e. uniform mRNA synthesis) within the nucleoid is sufficient to cause an enrichment of polysomes at mid-nucleoid (*Figure 4*; *Miangolarra et al., 2021*). Inside cells, this polysome enrichment at mid-nucleoid may be enhanced by a bias in mRNA synthesis across the nucleoid. For instance, the chromosomal region close to the origin of replication has been shown to be more highly expressed per gene copy than other regions on the chromosome (*Scholz et al., 2019*) and this region is located in the middle of the nucleoid prior to DNA replication (*Bates and Kleckner, 2005*; *Cass et al., 2016*; *Fisher et al., 2013*; *Kuwada et al., 2013*; *Kuwada et al., 2013*; *Mäkelä et al., 2021*; *Sadhir and Murray, 2023*; *Wang et al., 2006*). In addition, this highly expressed chromosomal region is the first one to replicate, which should lead to a further local increase in mRNA expression due to a doubling in gene dosage (*Pountain et al., 2022*). We did not consider such localized mRNA synthesis in our model. However, if we did, it would only help the mechanism that we proposed by increasing the polysome built-up at mid-nucleoid.

Other factors are likely involved in nucleoid segregation. In fact, our data revives the largely abandoned 60-y-old hypothesis by *Jacob et al., 1963* that cell growth separates the sister nucleoids through their potential attachment to the cell wall, but with two notable differences. First, the contribution of cell growth to nucleoid splitting would be minor relative to the polysome contribution (*Figure 1I*, *Figure 1—figure supplement 5*). Second, cell growth would contribute predominantly near the end of the division cycle (*Figure 1I*). This late timing would be attractive for two reasons. First, it corresponds to the time when polysomes stop accumulating between the separated sister nucleoids and polysome accumulations emerge at the middle of these nucleoids (i.e. at the ¼ and ¾ cell positions) to start the next round of segregation (*Figure 1D* and *Video 2*). Second, this is also when *E. coli* switches its cell wall growth pattern from dispersed along the cell body to zonal and divisome-dependent at mid-cell (*Cooper and Hsieh, 1988*; *Gray et al., 2015*; *Navarro et al., 2022*; *Wientjes and Nanninga, 1989*; *Woldringh et al., 1987*). Indeed, zonal cell growth between the sister nucleoids was a key assumption of the 1963 model (*Jacob et al., 1963*). What is not entirely clear is how the DNA would be attached to the peptidoglycan cell wall. Transertion links the DNA to the cytoplasmic membrane. Perhaps the coupling between transcription, translation, and membrane insertion extends to peptidoglycan binding.

Beyond cell elongation, thermodynamic demixing and other cellular processes such as DNA replication, loop extrusion, supercoiling, and preferential loading of DNA remodeling complexes are also

likely to be important for robust chromosome segregation and organization (*Danilova et al., 2007*; *Harju et al., 2024*; *Hofmann et al., 2019*; *Holmes and Cozzarelli, 2000*; *Lemon and Grossman, 2000*; *Mäkelä et al., 2021*; *Minnen et al., 2011*; *Sawitzke and Austin, 2000*; *Weitao et al., 1999*; *Wu et al., 2019b*; *Youngren et al., 2014*).

We note that in our time-lapse experiments, the accumulation of polysome signal appeared to slightly precede the depletion of DNA signal that marked the initiation of nucleoid splitting (*Figure 1E*). Polysome enrichment in the middle of unconstricted nucleoids was also occasionally observed in snapshot images of cells growing on glycerol, a slow growth condition that results in a single nucleoid segregation event late during the cell division cycle (*Figure 1—figure supplement 6*). This is not seen in our model in which polysome accumulation and nucleoid splitting occur at the same time (*Figure 4B* and *Video 3*). This small discrepancy may reflect a limitation of our experimental or modeling approach. For example, it is possible that the point spread function of our fluorescent DNA marker slightly delays the moment at which we can detect signal depletion at mid-nucleoid and thereby the initiation of nucleoid splitting. Alternatively, the small difference in timing may be associated with a model simplification. In our model, the nucleoid is effectively a solution of DNA fragments. In reality, the nucleoid consists of a circular polymer, crosslinked by nucleoid-associated proteins. These DNA crosslinks may cause a small resistance that marginally delays the initiation of nucleoid splitting relative to the polysome enrichment at mid-nucleoid.

### *E. coli* is an asymmetric organism

The spatial molecular asymmetries uncovered in our study contrast with the common perception of *E. coli* as a symmetric organism. In our relatively rich growth medium (M9gluCAAT), the distribution of polysomes at the new pole in newborn cells was, on average, higher than at the old pole through inheritance of the large mid-cell accumulation of polysomes from their mother cells (*Figure 3A and D*). We also observed an asymmetric distribution in DNA density within nucleoids, which correlated with the availability of polysome-free space along the cell length and width (*Figure 3E–G*). This asymmetry emerged before cell division (*Figure 3D* and *Video 2*). Simulations of our reaction-diffusion model suggest that slower diffusing/relaxing nucleoids are more likely to reproduce these nucleoid position and compaction asymmetries during the finite course of the cell division cycle (*Figure 4E*). A reduction in the apparent nucleoid diffusivity has been linked to nucleoid-associated proteins, which bridge and thus stiffen the DNA polymer (*Subramanian and Murray, 2023*). The physiological significance of these dynamic cellular asymmetries is not clear at this time, though it is conceivable that a difference in DNA compaction within the nucleoid may affect gene expression. Regardless, our study illustrates how spatial and temporal asymmetries in the cytoplasm can emerge from the interactions between two of the most important cellular components.

## Materials and methods

### Strains and constructs

Strains and plasmids used for this study are listed in *Supplementary file 2* and *Supplementary file 3*, respectively, while the sequences of the oligonucleotides used to make constructs can be found in *Supplementary file 4*.

To measure the concentration and spatial heterogeneity of ribosomes inside *E. coli*, we used strains in which the RplA 50 S ribosome subunit protein (strains CJW7323, CJW6768, CJW7020 and CJW7651) or the RpsB 30 S ribosome subunit protein (strains CJW6769 and CJW7021) are fused with mEos2 (strains CJW6768 and CJW6769), msfGFP (strains CJW7020, CJW7021, CJW7651, and CJW7798) or GFP (CJW7323) (*Gray et al., 2019*; *Xiang et al., 2021*). Nucleoid characteristics were measured using strains in which HupA, a nucleoid-associated protein, is fused with mCherry (strains CJW7323, CJW6723, and CJW7798) (*Xiang et al., 2021*). Alternatively, DAPI was used to stain the DNA (strains CJW6768, CJW6769, CJW7020, CJW7021, and CJW7651).

The mCherry-µNS (CJW7651) particles were chromosomally expressed from the native Lac promoter after induction with 150 µM IPTG (for 3 hr for agarose pad experiments). Strain CJW7651 was constructed as follows. The *gfp* coding sequence in the pER12 (pBAD322A-gfp-µNS) plasmid (kind gift from Dr. A. Janakiraman, City College of New York) was swapped with the mCherry-coding sequence using megaprimer whole plasmid (MEGAWHOP) cloning (*Bryksin and Matsumura, 2010*)

and the primer pair ER12-MCR-fwd/ER12-MCR-rev2 to generate plasmid pER12-mCherry. The mCherry-μNS coding sequence and the *rrnB* transcriptional terminator were amplified from the pER12-mCherry plasmid (primer pair μNSmCherry fwd/rev) and assembled with the *frt* site flanked with a kanamycin resistance cassette (amplified from pKD13 *Datsenko and Wanner, 2000* using the primer pair FRT_KanR fwd/rev) and the ColE1 origin of replication (PCR amplified from the pBAD22A plasmid (*Guzman et al., 1995*) using the primer pair ColE1 fwd/rev) using Gibson DNA assembly (*Gibson et al., 2009*) to form the pAPG1 plasmid. The pAPG1 plasmid also included two 50 bp sites homologous to the *attB* region of the *E. coli* chromosome, introduced as overhangs in the primers used for Gibson DNA assembly. The *attB* homologous regions allowed for the integration of the arabinose-inducible mCherry-μNS expression cassette into the respective site, though this was not used in this study. The pAPG1 plasmid was verified by sequencing using the pAPG1 seq1-5 primers. The mCherry-μNS coding sequence was then PCR amplified from the pAPG1 plasmid using primer pair lacZYA_redμNS fwd/rev, which includes 50 bp overhangs homologous to the region upstream and downstream of the *lacZYA* operon, and integrated downstream of the *lac* promoter in the MG1655 strain (*Guyer et al., 1981*; *Jensen, 1993*) using lambda red recombination and the pKD46 plasmid (*Datsenko and Wanner, 2000*) for the construction of the CJW7144 strain. Correct insertion of the mCherry-μNS coding sequence to substitute the *lacZYA* operon coding sequences was confirmed by colony PCR using primer pairs LacI_fwd/CynX_rev and LacI_fwd/mCherry_rev. Transduction with a P1 phage lysate of the CJW7144 strain was used to transfer the mCherry-μNS expression cassette into the CJW7020 strain using kanamycin as a selection marker. As a result, the CJW7145 strain was constructed, which was verified using colony PCR and the primer pairs LacI_fwd/CynX_rev and LacI_fwd/mCherry_rev, KanR_fwd/CynX_rev. The CJW7145 strain was then transformed using the pCP20 plasmid (*Datsenko and Wanner, 2000*), which encodes the FLP recombinase (*Cherepanov and Wackernagel, 1995*) to remove the kanamycin resistance cassette, yielding the CJW7651 strain.

The CJW7798 strain was derived from the MG1655 (DE3) strain (*Tseng et al., 2010*), which carries the T7 RNA polymerase-encoding gene under *lacUV5* control. Transduction with a P1 phage lysate of the CJW5158 strain (*Gray et al., 2019*) was used to transfer the gene encoding the HupA-mCherry fusion into the MG1655 (DE3) strain using kanamycin as a selection marker. The pCP20 plasmid (*Datsenko and Wanner, 2000*), which encodes the FLP recombinase (*Cherepanov and Wackernagel, 1995*), was used to remove the kanamycin resistance cassette, yielding the CJW7466 strain. Transduction with a P1 phage lysate of the CJW7019 strain, a kanamycin-resistant intermediate of strain CJW7020 (*Gray et al., 2019*), was used to transfer the gene encoding the RplA-msfGFP fusion into the CJW7466 strain using kanamycin as a selection marker. The resulting strain (CJW7766) was grown in the presence of kanamycin since the cells tend to lose msfGFP fluorescence in the absence of the antibiotic. The sequence of the CJW7766 strain was confirmed by whole genome sequencing.

The pET28:mTagBFP2 plasmid variant was derived from the pET28:GFP plasmid (*Shis and Bennett, 2013*), which was a gift from Mathew Bennett (Addgene plasmid # 60733; http://n2t.net/addgene:60733; RRID:Addgene_60733). First, the GFP coding sequence was substituted with the mTagBFP2 coding sequence from the pBAD-mTagBFP2 plasmid (*Subach et al., 2011*), a gift from Vladislav Verkhusha (plasmid # 34632; http://n2t.net/addgene:34632; RRID:Addgene_34632). Specifically, the mTagBFP2 coding sequence was amplified using the primer pair mTagBFP2 fwd/rev and assembled (Gibson DNA assembly) with the pET28 backbone, which was amplified in two pieces using the pET28_one fwd/rev and the pET28_two fwd/rev primer pairs to derive the pET28:mTagBFP2 plasmid. Then, the kanamycin resistance cassette that was originally present in the pET28 backbone was substituted with the chloramphenicol resistance cassette from the pSB3C5-proA-B0032-E0051 plasmid (*Davis et al., 2011*) which was a gift from Joseph Davis and Robert Sauer (Addgene plasmid # 107244; http://n2t.net/addgene:107244; RRID:Addgene_107244). Specifically, the backbone of the pET28:mTagBFP2 plasmid excluding the kanamycin resistance cassette was amplified using the pET28mTagBFP2 fwd/rev primers and assembled (Gibson DNA assembly) with the coding sequence of the chloramphenicol resistance cassette that was amplified using the cmR fwd/rev primer pair. As a result, the pET28:mTagBFP2-CmR plasmid was created. The proper assembly of the pET28:mTagBFP2 and pET28:mTagBFP2-CmR plasmids was confirmed by sequencing.

The pET28:mTagBFP2-CmR plasmid was introduced into CJW7766 by electroporation to create the CJW7798 strain, which was used to redirect the ribosomes away from the nucleoid onto the plasmid-expressed mTagBFP2 mRNAs after inducing the expression of T7 RNA polymerase with IPTG.

## Growth conditions

Strains CJW6769, CJW7020, and CJW7021 used in *Figure 2*, *Figure 2—figure supplement 1* were grown as previously described (*Gray et al., 2019*) using a basic M9 medium formulation without trace elements and supplemented with 0.2% (w/v) carbon source and when specified with 0.1% (w/v) casamino acids (CAA) and 1 µg/mL thiamine (T). The strain CJW6768 (used in *Figure 2* and *Figure 2—figure supplement 1*) was grown using the same medium formulation. The abbreviations of the medium growth conditions presented in *Figure 2A*, *Figure 2—figure supplement 1* are defined in *Supplementary file 1*. For the rest of our experiments on agarose pads or in a microfluidic device, strains CJW7323, CJW7651, and 7798 were grown in M9 salts (final concentrations: 33.7 mM $Na_2HPO_4$, 22 mM $KH_2PO_4$, 8.55 mM NaCl, 9.35 mM $NH_4Cl$, 1 mM $MgSO_4$, 0.3 mM $CaCl_2$) supplemented with trace elements (Fe, Zn, Cu, Co, B, Mn), 0.1% (w/v) thiamine, and, when specified, 0.4% (w/v) casamino acids. The pH of the 10x M9 salts was adjusted to 7.2 with NaOH. The trace elements were added at a final concentration of 13.4 mM ethylene-diamine-tetra-acetic-acid, 3.1 mM of $FeCl_3$-$6H_2O$, 0.62 mM of $ZnCl_2$, 76 µM of $CuCl_2$-$2H_2O$, 42 µM of $CoCl_2$-$2H_2O$, 162 µM of $HBO_3$-$2H_2O$, 8.1 µM of $MnCl_2$-$4H_2O$.

To achieve steady-state exponential growth prior to imaging, a stationary phase liquid culture in the appropriate growth medium was diluted at least 10,000 times and grown to an optical density at 600 nm ($OD_{600}$) between 0.1 and 0.3. These cells were either loaded in a mother-machine-type microfluidics device (*Lin and Jacobs-Wagner, 2022*; *Wang et al., 2010*) where they were grown under the constant flow of medium (~0.5 µL/s), or spotted on 1% agarose pads prepared with the same growth medium. In the microfluidic device, cells were grown for at least 3 hr prior to image acquisition. All precultures and experiments were performed at 30 °C.

The mCherry-µNS particles (CJW7651 strain) were imaged on agarose pads after inducing exponentially growing cells (in M9gluCAAT) with 150 µM IPTG for 3 hr (*Figure 4E–G*). Before spotting on the 1% agarose pad, the induced cells were stained with DAPI (1 µg/mL) for 5 min.

To redirect ribosomes to plasmid-expressed, T7 promoter-driven mTagBFP2 mRNAs, exponentially growing CJW7798 cells grown in M9glyCAAT to an OD ~0.2 were spotted on a 1% agarose pad containing M9glyCAAT and 100 µM IPTG, the latter to induce the expression of the T7 RNA polymerase.

## Rifampicin treatment

The antibiotic rifampicin (see *Supplementary file 5*) was used to block transcription, deplete mRNAs, and release the mRNA-bound ribosomes (polysomes). Rifampicin treatment was performed either in batch culture (*Figure 1—figure supplement 3*) or in microfluidics (*Figure 6* and *Video 4*).

For the microfluidic experiment, two rounds of rifampicin treatment were performed by switching between M9gluCAAT containing antibiotic (100 µg/mL) and antibiotic-free medium at 30 °C. For the medium switches, solenoid valves were used in a custom-built pressurized perfusion system, achieving fast (within 1 min) changes in the cellular environment. The first switch to rifampicin occurred 2 hr after normal growth in an antibiotic-free medium (*Figure 6A* and *Video 4*). Rifampicin treatment lasted 2 hr until the system switched back to an antibiotic-free medium, where cells were left to recover for 8 hr. Twelve hours into the experiment, the system switched again to a rifampicin-containing medium.

## A22 and cephalexin treatment

Cephalexin (50 µg/mL) and A22 (4 µg/mL) (see *Supplementary file 5*) were added to exponentially growing CJW7323 cell cultures ($OD_{600}$ ~0.1) in M9gluCAAT just prior to spotting on a 1% agarose pad, which was made with the same growth medium and contained the same antibiotic concentrations. The radial expansion of the cells and the polysome and nucleoid signals were tracked over time in a time-lapse experiment. The selected concentration of A22 has previously been shown to not affect cell growth (*Takacs et al., 2010*). All precultures and time-lapse imaging experiments were performed at 30 °C.

## Chloramphenicol and cephalexin treatment

Exponentially growing CJW7323 cells ($OD_{600}$ ~0.1) in M9gluCAAT were pre-treated with cephalexin (50 µg/mL) for 1 hr in the shaking flask. Then the cells were spotted on a 1% agarose pad, which was made with the same growth medium and contained cephalexin (50 µg/mL) and chloramphenicol (75 µg/mL). All precultures and time-lapse imaging experiments were performed at 30 °C.

## Ectopic polysome accumulation experiment

Exponentially growing CJW7798 cells in M9glyCAAT supplemented with kanamycin (50 µg/mL) and chloramphenicol (35 µg/mL) were washed one time with M9glyCAAT minimal medium lacking antibiotics but supplemented with 100 µM IPTG (see *Supplementary file 5*). The washed cells were spotted on a 1% agarose pad prepared with M9glyCAAT also supplemented with IPTG (100 µM) for time-lapse imaging. All pre-cultures and time-lapse imaging experiments were performed at 30 °C. Since chloramphenicol affects protein synthesis and thus polysome formation, the agarose pads lacked chloramphenicol, resulting in a fraction that lost the plasmids based on the absence of blue fluorescence. Therefore, only cells that expressed blue fluorescence (and thus carried the pET28:mTagBFP2-CmR plasmid) were analyzed.

## DNA quantification using flow cytometry

Exponentially growing CJW7798 cells ($OD_{600}$~1) in M9glyCAAT supplemented with kanamycin (50 µg/mL) and chloramphenicol (35 µg/mL) were washed with M9glyCAAT minimal medium lacking antibiotics. The washed cells were used to start two cultures in M9glyCAAT without antibiotics. In one culture, the expression of the mTagBFP2 was induced with 100 µM ITPG. In the second culture, the cells were not supplemented with IPTG (i.e. no induction). After 200 min, the cells were fixed with 70% cold ethanol for 30 min, and stained with 5 µM DRAQ5 (Invitrogen eBioscience Cat. 65-0880-92) for 30 min at 37 °C as previously described (*Silva et al., 2010*). The stained cells were washed with 0.1 Tris, 2 mM $MgCl_2$ buffer (pH 7.4), and then used for flow cytometry. The Attune CytPix flow cytometer (Invitrogen) was used to quantify the DRAQ5 fluorescence in the red laser 2 (RL2) channel (637 nm excitation, 690DLP beamsplitter, 720/30 emission, 440 voltage) and the cell size in the side scatter (SSC) channel (320 voltage). A low threshold of $0.1 \times 10^3$ was set for the forward scatter and $0.3 \times 10^3$ for the side scatter signal. The flow was set at 12.5 µL/min for a total of $10^5$ counted events per sample. The stained cells were diluted appropriately (usually 100-fold) to ensure less than $10^3$ events per second. For each biological replicate, the DNA content (RL2-Area) and the cell size (SSC-Area) were quantified. Lower thresholds in the RL2-Area and the SSC-Area channels were applied to exclude very small events without fluorescence, as well as a polygon gate in the SSC-Area vs. SSC-Height statistics to exclude events with more than one cell. These gates were first applied on the data from the uninduced sample and then replicated on the data from the induced sample for each biological replicate. As a result, the exact same thresholds were applied between the induced and uninduced samples per biological replicate for fair comparison. The analysis of the flow cytometry data was performed using a custom Python package (*flowio_to_pandas*), which is based on the FlowIO flow cytometry standard (FCS) file parser (*White et al., 2021*) and which allows for interactive gating.

## Microscopy

Strains CJW6769, CJW7020, and CJW7021 used in *Figure 2A-B*, *Figure 2—figure supplement 1*, were imaged on agarose pads using the same microscopy set-up and optical configurations as previously described (*Gray et al., 2019*). Snapshots of the CJW6768 strain (used in *Figure 2*, *Figure 2— figure supplement 1*) were taken using a Nikon Ti-E microscope equipped with a 100 x Plan Apo 1.45NA Ph3 oil objective, a Hamamatsu Orca-Flash4.0 V2 CMOS camera (16-bit, Slow Scan sensor mode) and a Lumencor Spectra X LED (Light Emitting Diode) engine. The 395/25 nm LED was used to excite DAPI and the 470/24 nm LED was used to excite GFP. DAPI fluorescence was acquired using an ET350/50 x (excitation filter), RT400lp (dichroic mirror), ET460/50 m (emission filter) filter cube from Chroma. The ET470/40 x (excitation filter), T495lpxr (dichroic mirror), ET525/50 m (emission filter) configuration from Chroma was used for GFP fluorescence acquisition. The microscope was controlled using the NIS Elements software by Nikon and the ND acquisition module.

For the time-lapse observation of cells (strain CJW7323) treated with chloramphenicol, A22, and/or cephalexin, images were taken using a Nikon Ti-E microscope, equipped with a 100 x Plan Apo 1.45NA Ph3 oil objective, a Hamamatsu Orca-Flash4.0 V2 CMOS camera (16-bit, Slow Scan sensor mode), and a Sola solid-state white light source (Lumencor), in a temperature-controlled enclosure (Okolabs). The AT470/40 x excitation filter, combined with a T495LPXR beam-splitter and an ET525/50 m emission filter, was used for GFP. For mCherry visualization, the ET560/40 x excitation filter, combined with a T585lp beam-splitter and a ET630/75 m emission filter, was used. A neutral density filter (32 x) was applied in the excitation path to reduce phototoxicity and photobleaching. Images were taken

every 2.5 min in the brightfield channel (phase contrast) and every 5 min in the fluorescence channels (RplA-GFP and HupA-mCherry).

For the rest of the brightfield and epi-fluorescence wide-field microscopy experiments (strains CJW7323, CJW7651, and CJW7798), snapshots or time-lapse images were taken using a Nikon Ti2-E inverted microscope, equipped with a 100 x Plan Apo 1.45NA Ph3 oil objective, a Photometrics Prime BSI back-illuminated sCMOS camera (2048x2048 pixels sensor with a pixel size of 6.5 μm), and a Lumencor Spectra III LED (Light Emitting Diode) engine, in a temperature-controlled enclosure (Okolabs). The HDR 16-bit sensor mode was used to acquire images. The microscope was controlled using the NIS Elements software by Nikon, and the JOBS or the ND acquisition module were used to acquire snapshots or time-lapse images. The Perfect Focus System (by Nikon) was used to maintain focus in microfluidics experiments. The auto-focus function in NIS Elements was used to locate the optimal z-position in agarose pad experiments (snapshots or time-lapse). For the DAPI, BFP, GFP, or mCherry channels, a polychroic mirror (FF-409/493/596-Di02 by Shemrock) combined with a triple-pass emitter (FF-1-432/523/702-25 by Shemrock) was used. For DAPI, BFP, and GFP imaging, additional emission filters were applied in the optical path (FF01-432-36 by Shemrock, FF01-432-36 by Shemrock, and ET525/50 M by Chroma, respectively). DAPI, BFP, GFP, and mCherry were excited using a 390/22 nm, 390/22 nm, 475/28 nm, and 575/25 nm LED, respectively.

For the microfluidic experiments (strain CJW7323), the excitation light intensity was reduced to 20% for the 1 min interval and 30% for the 3 min interval imaging of the RplA-GFP and HupA-mCherry. An exposure time of 120 ms was used for both markers. In all time-lapse experiments, a neutral density filter (absorptive ND filter, OD:1.3/5% transmission, NE13B by Thorlabs) was also applied to reduce the LED power and minimize phototoxicity and photobleaching. The LED powers (factored by the neutral density filters) were calibrated using a microscope slide power meter with an 18×18 mm sensor size (S170C by Thorlabs). The light intensity was measured for each excitation wavelength (475/28 nm and 575/25 nm) at the end of the objective (without immersion oil) for different LED intensities (% of maximum intensity). From the generated calibration curves, the RplA-GFP excitation light power was estimated to be 524 μW and 629 μW for the 1 min and 3 min fluorescence interval imaging, respectively. HupA-mCherry was excited with 109 μW light power for the 1 min and 191 μW for the 3 min fluorescence interval imaging.

The type N immersion oil (by Nikon) was used in all experiments except for the time-lapse observation of cephalexin, A22, A22 + cephalexin, and chloramphenicol- and cephalexin-treated cells where the type F immersion oil (by Nikon) was used. All imaging (time-lapse and snapshots) was done at 30 °C.

## Analysis software and code availability

Cropping and alignment of the microfluidic channels were performed with MATLAB (https://www.mathworks.com/) using a previously published pipeline from our lab (*Lin and Jacobs-Wagner, 2022*). Supervised classification and curation of the Oufti cell meshes was also implemented in MATLAB (*Campos et al., 2018*; *Govers et al., 2024*; *Gray et al., 2019*). For the T7 experiments, segmentation and tracking of the CJW7798 cells was performed using the Omnipose deep neural network architecture (*Cutler et al., 2022*), using a previously trained model (*Thappeta et al., 2024*) and the SuperSegger MATLAB-based package (*Stylianidou et al., 2016*). The remaining analyses were performed using Python 3.9 (https://www.python.org/), that included the *numpy* (*Harris et al., 2020*), *scipy* (*Virtanen et al., 2020*), *pandas* (*McKinney, 2010*), *scikit-mage* (*van der Walt et al., 2014*), *scikit-learn* (*Pedregosa et al., 2012*), *shapely* (*Gillies et al., 2023*), *statsmodels* (*Seabold and Perktold, 2010*) and *pytorch* (*Paszke et al., 2019*) libraries. The *matplotlib* (*Hunter, 2007*) and *seaborn* (*Waskom, 2021*) libraries were used for plotting. The analysis pipeline and functions (summarized in *Supplementary file 6*) are available in the Jacobs-Wagner lab GitHub repository http://www.github.com/JacobsWagnerLab/published/tree/master/Papagiannakis_2025, copy archived at *JacobsWagnerLab, 2025* and the GitHub repository of Alexandros Papagiannakis https://github.com/alexSysBio/Time_lapse_on_agarose_pad, copy archived at *Papagiannakis, 2025a*; https://github.com/alexSysBio/flowio_to_pandas, copy archived at *Papagiannakis, 2025b*; https://github.com/alexSysBio/Adding_ND2_images_to_python, copy archived at *Papagiannakis, 2025c*; https://github.com/alexSysBio/Cell_medial_axis_definitions, copy archived at *Papagiannakis, 2025d*; https://github.com/alexSysBio/Image_background_subtraction, copy archived at *Papagiannakis, 2025e*.

## Cell segmentation and tracking

For the CJW6768, CJW6769, CJW7020, and CJW7021 strains used in *Figure 2*, *Figure 2—figure supplement 1*, Oufti (*Paintdakhi et al., 2016*) was used to draw cell meshes on the phase contrast snapshots and a MATLAB-based support vector machine model was used to remove the badly segmented cells as previously described (*Campos et al., 2018*; *Govers et al., 2024*; *Gray et al., 2019*).

The Omnipose deep neural network architecture (*Cutler et al., 2022*) with a previously trained model (*Thappeta et al., 2024*) was used to segment the CJW7798 cells during T7 RNA polymerase induction on an agarose pad. The segmented cells were then tracked using SuperSegger (*Stylian-idou et al., 2016*). A custom class (*omnipose_to_python_ghv.py*) was developed to transfer the SuperSegger segmentation and tracking data into Python for post-processing.

Due to the unusual and variable morphology of the A22/cephalexin-treated cells, a custom Python class was developed for their segmentation and tracking. The *Otsu_phase_segmentation_ghv.py* class segments the cells by applying an Otsu threshold (*Otsu, 1979*) on the inverted phase images, followed by binary dilation (*scikit-image.morphology.binary_dilation* Python function) hole filling (*scipy.ndimage.binary_fill_holes* Python function), and binary closing (*scikit-image.morphology.binary_closing Python function*). The segmentation masks were tracked between subsequent time-points, using cell distance as well as cell area constraints, and linked into cell growth trajectories. Cell morphology criteria such as the maximum pixel distance from the medial axis and the medial axis sinuosity were used to remove bad segmentations. The remaining segmentation masks were manually curated.

For the other experiments on agarose pads, a neural network (*Wiktor et al., 2021*; *Zhou et al., 2020*) with a previously designed and trained U-net architecture (*Mäkelä et al., 2024*) was used to segment single cells based on phase contrast snapshots. The generated segmentation masks were further processed by watershed separation, filling the holes within masks, and removing unusually small masks. Finally, a graphical interface was used to manually remove the badly curated cells (less than 5% of the segmented cell population).

In contrast to the Oufti software (*Paintdakhi et al., 2016*) that generated sub-pixel meshes around the cell boundaries, the other applied segmentation algorithms returned pixel-based cell masks. To deal with this discrepancy, the sub-pixel cell meshes from Oufti were converted into pixel-based cell masks by collecting the pixels within the circumscribed single-cell area in Python (*oufti_snapshots_GrayGovers* class, included in the *snapshots_analysis_OUFTI_GrayGovers* Python script). After this conversion, the same Python-based functions were applied for the analysis of the cell fluorescence and morphology statistics regardless of the segmentation method used.

To segment cells growing in the microfluidic device, a custom library of image analysis functions (*mother_machine_segmentation* class, included in the *microfluidics_segmentation_ghv* Python script) was developed in Python. This algorithm was applied to the cropped, aligned, background-subtracted, and inverted phase contrast images that were produced using a previously published pipeline (*Lin and Jacobs-Wagner, 2022*) in MATLAB. Cell segmentation was performed in three steps. First, all the cells within the microfluidics channel, which were brighter than the background in the inverted and background-corrected phase-contrast channel, were segmented using an Otsu intensity threshold (*Otsu, 1979*). This crude thresholding step, which separated the cell (brighter: 1) from the back-ground (darker: 0) pixels, yielded at least one masked label for the entire row of stacked cells in each microfluidic channel. A watershed segmentation was then applied to define the boundaries between individual cells and split the Otsu-based binary mask(s). The watershed algorithm was guided by the number of cells and their relative positions in the microfluidic channel, which were determined using an adaptive filter combined with the local decrease of the inverted phase intensity between the poles of adjacent cells. Finally, a graphical interface was used to curate the cell masks by manually splitting or merging cell labels.

After segmentation, the curated single-cell masks were tracked over time and linked into trajectories from birth to division using the centroid distance and the relative area difference between cells at consecutive timepoints (*mother_machine_tracking* class, included in the *microfluidics_segmentation_ghv* Python script, or the *fluorescence_analysis* class and its depending functions in the *Time_lapse_on_agarose_pads* Python package). Examples of cell segmentation and tracking in microfluidics are shown in *Video 1* and *Figure 1—figure supplement 1A*. Regardless of the image acquisition time

intervals for the fluorescence channels, phase-contrast images were acquired every minute, which allowed for accurate cell tracking.

Fluorescence background correction was different between microfluidic and agarose-pad experiments. In microfluidics, the average background was measured within two 8-pixel (0.528 μm) wide areas, one on the left side and one on the right side of the channel, at least 10 pixels (0.66 μm) away from the channel boundaries from top to bottom. The estimated background was then subtracted along the channel length. The background correction was integrated into the *get_fluorescence_image* function, located in the *microfluidics_analysis_functions_*ghv Python script.

In agarose-pad experiments, an Otsu threshold (*Otsu, 1979*) was used on the inverted phase-contrast image to segment all cells. A binary dilation was then performed on these crude cell masks before estimating the local average fluorescence of the unmasked pixels (pixels outside the dilated cell masks). The locally-averaged background was then used to fill in the cell areas and reconstruct the background of the entire field of view in the absence of cells. The smoothed (Gaussian smoothing) reconstructed background fluorescence was subtracted from each fluorescence image. This background correction pipeline (*back_sub* function) is integrated in both the *unet_snapshots* class and the *oufti_snapshots_GoversGray* class. For the time-lapse imaging of A22-treated CJW7323 cells (*Figure 8*) and CJW7798 cells during induction of T7 RNA polymerase expression (*Figure 7*), the fluorescence background was not subtracted since we did not need to quantify the raw pixel values. The normalized polysome and nucleoid fluorescence normalized by the average whole cell fluorescence is shown instead, marking their relative cellular rearrangements.

## mCherry-μNS particle localization and tracking

For the localization of the diffraction-limited mCherry-μNS particles (*Figure 5*), a set of custom functions was developed in Python and implemented in the *particle_positions_snapshots* class, included in the *snapshots_analysis_UNET_ghv* Python script.

The fluorescent particles were segmented in the background corrected mCherry-μNS fluorescence images using a Laplace of Gaussian (LoG) and an adaptive filter combined. The particle masks that had an area larger than a specified threshold (90 pixels), which usually included two particles from the same or adjacent cells, were further processed by applying a relative fluorescence threshold (90th percentile of masked pixel intensity) to find the local fluorescence maxima and separate the two objects. Additional minimum area and aspect ratio constraints were applied. To accurately estimate the particle position with sub-pixel resolution, a 2D Gaussian function (*Equation 1*) with rotation (*Equation 2*) was fitted (least square method: *scipy.optimize.leastsq*) to an area of 7×7 pixels centered at the centroid of the particle mask:

$$g\left(x, y\right) = A \exp\left(-\frac{1}{2}\left(\left(\frac{x_0 - x_{rot}}{\sigma_x}\right)^2 + \left(\frac{y_0 - y_{rot}}{\sigma_y}\right)^2\right)\right) \tag{1}$$

and

$$\begin{aligned} x_{rot} &= x \cos\left(\theta\right) - y \sin\left(\theta\right) \\ y_{rot} &= x \sin\left(\theta\right) + y \cos\left(\theta\right) \end{aligned} \tag{2}$$

where *x* and *y* are the coordinates in the 2D imaging plane, *A* is the amplitude of the fitted Gaussian, *σ* is the standard deviation in each dimension, and *θ* is the rotation angle in radians, which was used to rotate the Gaussian distribution ($x_{rot}, y_{rot}$) around the particle center or the Gaussian mean ($x_0, y_0$).

## Two-dimensional cell mapping and alignment

Ribosome and nucleoid fluorescence statistics, as well as particle positions within cells, were mapped in relative 2D cellular coordinates from pole to pole and across the cell width. Such mapping allowed us to project the fluorescence or position statistics in 1 or 2D for specific cell length ranges and cell division cycle intervals. For this analysis, it was necessary to draw the medial axis for each of the segmented cells, which was obtained differently in microfluidic and agarose-pad images.

In the microfluidic experiments, where all the cells were stacked in a vertically oriented channel, the medial axis was drawn by fitting a second-degree polynomial to the most distant coordinates from the cell boundaries, excluding the cell caps at the poles where the medial axis was linearly extrapolated.

The length of the cells, which was parallel to the length of the microfluidic channel, was used as the independent variable for the fitting. This medial axis estimation is implemented via the *all_medial_axis* function in the *microfluidics_analysis_functions_ghv* Python script.

However, in the agarose-pad experiments, cells were randomly oriented in the imaging plane. Thus, it was impossible to use one of the coordinates (x or y) as the independent variable as this would bias our medial axis estimation toward the same coordinate for cells that were not diagonally oriented. Instead, we developed an algorithm that scanned through the middle of the cells with a fixed sub-pixel step and directionality constraints to identify nodes at the most distant locations from the cell boundaries. These ordered (from pole to pole) and numbered nodes were used to fit the x and y coordinates of the medial axis separately, using the number of the node as an independent variable and its position coordinates as dependent variables. The degree of the fitted polynomials (*d*) scaled linearly with the total number of nodes (*N*) based on the relation $d = 0.1N - 5$. Similar to the microfluidic experiments, the medial axis was linearly extrapolated at the cell caps. This medial axis estimation is implemented using the *get_medial_axis* function in both the *unet_snapshots* and *oufti_snapshots_GoversGray* classes located in the *snapshots_analysis_UNET_ghv* and *snapshots_analysis_OUFTI_GrayGovers_ghv* Python scripts, respectively. The medial axis estimation function is also provided as a separate function in the *Bivariate_medial_axis_estimation.py* Python script.

The medial axis of each cell, with a resolution of 0.1 pixels, was used to map the cell pixels and particle positions along the cell length and width. The projection of each pixel or particle position on the medial axis, defined as the most proximal node on the central line, was used to determine its cell length coordinate. The distance between the pixel or particle position and its medial axis projection was used to determine its absolute cell width coordinate. The sign of the cross-product $\overrightarrow{PL} \times \overrightarrow{PC}$ (plus or minus), where *L* was the location of the particle, *P* was its projection on the medial axis, and *C* was the center of mass of the cell mask, was used to determine the position of particles or cell pixels on the sagittal plane.

## Cell polarity and ages

In time-lapse experiments, the old and new poles of the cells were determined as follows. The pole of the daughter cell that was closer to the mid-cell position of its predivisional mother cell, where cell constriction occurs, was defined as the new pole. The medial axis coordinates were adjusted based on this polarity. As a result, the relative cell length extended from –1 to 1, with –1 corresponding to the old pole, 1 to the new pole, and zero to the cell center (see *Figure 3—figure supplement 1B and D*). The implementation of this method is included in the *get_polarity* function in the *microfluidics_analysis_functions_ghv* Python script.

Cell lineages and ages relative to the oldest mother cells (first lineage) at the closed end of the microfluidic channel were determined using the orientation of the cell poles. The daughter cells that had the same polarity as the oldest mother cells were assigned the same age and lineage as their mothers. The daughter cells with the opposite polarity, which inherited the new pole of their mother cells, were considered younger and belonged to the second lineage. The daughter cells of mother cells belonging to the second lineage were classified as lineages three and four depending on their polarity. Cells from the third lineage had opposite polarity and those from the fourth lineage had the same polarity as their second lineage mother cells. Finally, mother cells from the third lineage, which inherited the new pole of their second-lineage mother cells, divided to yield the fifth and sixth cell lineages. The fifth lineage had the same polarity as its third-lineage mother and its sister lineage (sixth) had the opposite polarity and inherited the new pole from its third-lineage mother cell. The implementation of this method is included in the *get_ages* function in the *microfluidics_analysis_functions_ghv* Python script. For the analysis of the data presented in *Figure 3*, only the third, fourth, fifth, and sixth lineages were considered to avoid age-related effects at the cell poles (*Chao et al., 2024*; *Coquel et al., 2013*; *Koleva and Hellweger, 2015*; *Lapińska et al., 2019*; *Lindner et al., 2008*; *Proenca et al., 2019*).

## Construction of intensity profiles, 2D cell projections, demographs, and kymographs

With the ribosome and nucleoid fluorescence pixels as well as the μNS particle positions inside the cells having been mapped, the fluorescence and particle statistics from pole to pole (medial axis

projection) and across the cell width (distance from the medial axis) were plotted for different cell division cycle intervals or cell length ranges.

An intensity profile represents the change in a fluorescence statistic along the cell from one pole to the other. Such a fluorescence statistic can be the average fluorescence intensity (as in *Figures 1D, 3G and 5C*, *Figure 3—figure supplement 1B*), which corresponds to the concentration of the reported protein per cell area (sum of pixels divided by the number of pixels). In other analysis (as in *Figure 6C*, *Figure 1—figure supplement 6C*), the protein concentration along the cell length was divided by the whole cell average. This scaled statistic is expressed as a percentage change relative to average concentration (mean %). A value above 100 indicates an increase above the average, whereas a value below 100 corresponds to a decrease below the average concentration. This scaled statistic is insensitive to the RplA and HupA concentration variability between cells, or to the maturation of the GFP or mCherry fluorophores. As a result, the scaled concentrations showcase rearrangements of the tagged proteins and do not imply changes in protein synthesis relative to cell growth.

Intensity profiles were plotted for single cells (*Figure 3—figure supplement 1B*, *Figure 1—figure supplement 6A*) or populations (as in *Figures 1D, 3G, 5C, 6C and E*). In the second instance, the intensity profiles represent the average 1D projection of the fluorescence statistic for a specific cell division cycle interval, cell length range, growth rate range, polysome and nucleoid asymmetry group, or time range (e.g. during antibiotic treatment). To generate a single-cell intensity profile, the medial axis length, and its projected pixels were binned, and the average value of the selected fluorescence statistic was calculated per bin. If the number of bins was higher than the length of the medial axis in pixels, the missing values were filled in by linear interpolation. To avoid artifacts from the cell boundaries, the intensity profile was estimated for a box with a width of six pixels (~0.4 μm) centered on the medial axis (i.e. three pixels on each side of the medial axis). The intensity profile was occasionally smoothed by a moving average of a specified cell-length window, without excluding the bins at the edges which were not smoothed. Alternatively, a univariate spline (*scipy.interpolate.UnivariateSpline*) was fitted to the cell-length-binned data (as in *Figure 5C*). The univariate splines were fitted to the binned fluorescence intensity along the cell length. The number of cell-length bins ($N_{bin}$) linearly scaled with the cell length in μm ($l_{cell}$) using the first-order relationship: $N_{bin} = \frac{100}{4.3} l_{cell}$ (only the whole part of the decimal was considered). Similar to the intensity profile where a fluorescence statistic was plotted along the cell length, the density profiles (histograms of the particle positions along the cell) for the mCherry-μNS (*Figure 5C*) particles along the cell were also plotted. Note that the intensity profiles shown in *Figures 1D, 3G, 5C, 6C and E*, *Figure 1—figure supplement 1D*, and *Figure 2—figure supplement 3D* are averages of many segmented cells or cell division cycles. In these averaged profiles, the variability in the timing of nucleoid segregation across cells attenuates the appearance of polysome accumulation. Furthermore, the point spread function of fluorescent ribosome markers causes an underestimation of the magnitude of the polysome accumulations in epifluorescence micrographs (*Bakshi et al., 2012*).

The intensity profiles were also used to construct demographs from agarose-pad experiments (*Figure 2A*, *Figure 2—figure supplement 1*, and *Figure 1—figure supplement 6B*) where cells were sorted by cell length from the shorter newborn to the longer predivisional cells. Intensity profiles were also used to construct ensemble kymographs from microfluidic experiments where the cells were sorted by their relative position in the cell division cycle (*Figure 1C*, *Figure 2—figure supplement 2B*) or single-cell kymographs from agarose pad time-lapse measurements (*Figure 7D–F*). Specifically, the average intensity profile was estimated for each cell length or cell division cycle bin, and the average intensity profiles were stacked from birth to division (left to right) oriented according to the cell polarity (top to bottom). In the ensemble kymographs, the fluorescence intensities were projected along the relative cell length (cell length %), whereas in the single-cell kymographs the length of the projection linearly scales with the cell length. A Gaussian smoothing (*skimage.filters.gaussian*) was applied to smooth the demographs and ensemble kymographs.

In addition to the intensity or density profiles for a fluorescence statistic or the density of the μNS particles along the medial axis, 2D projections of the ribosome or nucleoid signal distribution (*Figures 1D, 3D, 5B, 6C and E*, *Figure 2—figure supplement 3A*) as well as the μNS particle positions (*Figure 5B*) were also constructed. These 2D intensity or density maps were constructed by binning the cell pixels or particle positions not only by cell length, but also by cell width. The average fluorescence statistic or the particle density was then shown per bin, including data from many single

cells within a specified cell division cycle or cell length range. A Gaussian smoothing (*skimage.filters. gaussian*) was applied to smooth the 2D projections.

## Extraction of population-level statistics of polysomes and nucleoids

To quantify the correlation between polysome accumulation and nucleoid segregation at the population level (*Figure 2*, *Figure 2—figure supplement 1*), the average polysome accumulation and nucleoid depletion at mid-cell were extracted from cell snapshot images. These statistics were obtained by averaging the scaled (divided by the whole cell average concentration) RplA-GFP and HupA-mCherry intensity profiles of all the cells in the population. Therefore, these statistics describe the average 'behavior' of the population under a specific growth condition. We reasoned that if, in a specific nutrient condition, nucleoid segregation happens earlier during the division cycle, then a higher fraction of the population will have segregated nucleoids. If so, the scaled HupA-mCherry concentration should exhibit a stronger depletion at mid-cell on average. This depletion was measured as the relative concentration difference between the HupA-mCherry peaks, which corresponded to the two lobes of the constricting nucleoid, and the trough at mid-cell, which corresponds to the point of nucleoid splitting. Similarly, the average polysome accumulation was measured as the RplA-GFP relative concentration difference between the mid-cell accumulation (peak) and the polysome-depleted regions near the quarter-cell positions, or at the centers of the segregating sister nucleoids (troughs). The same procedure was used to quantify the relative polysome accumulation at mid-cell and the relative nucleoid depletion in the same region between cell division cycles with different growth rates in the same nutrient condition (*Figure 2—figure supplement 2*). However, in the latter case the raw RplA-GFP and HupA-mCherry fluorescence was used (not scaled by the cell mean), because each calculation was based to the average intensity profile from a single cell division cycle and was not affected by the intercellular fluorescence variability.

## Nucleoid segregation cycle tracking

In this work, the nucleoid segregation cycle was defined as the period from the end of a nucleoid splitting event (i.e. when a segregating nucleoid is segmented by local thresholding (custom Python function: *LoG_adaptive_image_filter.py*) as a separate object from its sister nucleoid) until the end of the splitting of the same nucleoid object (*Figure 1—figure supplement 4D*), usually after cell division but sometimes in the same cell division cycle (*Figure 1—figure supplement 4A*).

To track the nucleoid objects through the nucleoid segregation cycle from the mother cells to their daughters, we used the cell polarity, i.e., based on pole identity (*Figure 1—figure supplement 4B*). The nucleoids of daughter cells that had opposite polarity to their mother cells were inherited from the new pole of the mother cells (groups 1 and –1 in *Figure 1—figure supplement 4B–C*). The nucleoids in the daughter cells that had the same polarity as their mother cells were inherited from the old pole of the mother cells (groups 2 and –2 in *Figure 1—figure supplement 4B and C*). During a complete nucleoid segregation cycle, the nucleoid was tracked from the quarter position of the mother cell to the center of its daughter cells (*Figure 1—figure supplement 4C*). This method is implemented in the *track_nucleoids* function, included in the *microfluidics_analysis_functions_ghv* Python script.

To quantify the polysome accumulation and nucleoid splitting during the nucleoid segregation cycle (*Figure 1E and F*), we measured the RplA-GFP and HupA-mCherry concentration within the cell length region of 2.5 pixels adjacent to the center of the tracked nucleoid object (five pixels or 0.33 μm in total).

## Quantification and segmentation of polysomes and nucleoids segmentation in cells treated with cephalexin or cephalexin + A22

The polysome accumulations and nucleoid objects were also segmented in A22 and cephalexin-treated cells using an adaptive filter (custom Python function: *LoG_adaptive_image_filter.py*) on the masked cell fluorescence. The number of the detected polysome accumulations or nucleoid objects corresponds to the number of the segmented labels in the RplA-GFP or HupA-mCherry channel, respectively. Similarly, the average polysome accumulation or nucleoid area corresponds to the average area of all segmented labels per channel. For a fair comparison between cells treated with A22 + cephalexin and cells treated with cephalexin alone, cells were randomly sampled without substitution from

12 cell area bins between 10 and 13 μm² (bin width = 0.25 μm²) such that the compared populations in *Figure 8F* had the same cell area distributions (same number of cells per cell area bin).

## Quantification of the distance between fusing or non-fusing nucleoids in cephalexin- and chloramphenicol-treated cells

The distance between nucleoids corresponds to the cell length difference between the peaks of the longitudinal HupA-mCherry signal projection (*Figure 4—figure supplement 1A and B*). To calculate these distances, the HupA-mCherry longitudinal profile was first smoothed using a fourth-order univariate regression (*scipy.interpolate.UnivariateSpline*). The optimal smoothing factor ($10^3$) was determined using the L-curve method (*Nasehi Tehrani et al., 2012*) where the x-axis corresponds to the value of the smoothing factor and the y-axis to the variance in detected nucleoid number during cell growth. The average variance was considered across the tracked cell trajectories. The smoothing factor at the elbow of the L-curve is the optimal point between overfitting, where the noise in the HupA-mCherry signal is wrongly assigned to nucleoid objects, and underfitting, where two nucleoids in proximity may be wrongly identified as one. After smoothing, the peaks of the HupA-mCherry projection were detected using the *scipy.signal.find_peaks* function, applying a lower threshold of 125 arbitrary units. The number of the detected peaks is equal to the nucleoid number per cell, whereas the distance between the peaks is equal to the distance between the nucleoid centroids. A nucleoid fusion event is indicated by the reduction in nucleoid number between consecutive frames in a cell trajectory.

The tracked cell trajectories were classified into those that fuse their nucleoids and those that do not. The first class includes cell trajectories that contain a nucleoid fusion event and end with less detected nucleoids objects compared to their beginning. The second class includes cell trajectories that do not contain any nucleoid fusion event and end with an equal number or more detected nucleoids compared to their beginning. All classified trajectories start at timepoint 0 min, which corresponds to the first timepoint of the experiment, or the first microscopy image taken after spotting the cells on the chloramphenicol-containing agarose pad. Only trajectories with more than one nucleoid at timepoint 0 min are considered. This first timepoint also corresponds to when the minimum distance between adjacent nucleoids is quantified for each cell trajectory (plotted in *Figure 4—figure supplement 1C*). The nucleoid segregation variability shown in *Figure 4—figure supplement 1B*, is consistent with previous observations (*Spahn et al., 2014*).

## Linear mixed-effects models to determine the relative contribution of polysome accumulation and cell elongation to the migration of the sister nucleoids

To estimate the relative contribution of the polysome accumulation at mid-cell and that of cell elongation to the migration of the separated sister nucleoids (first seen in *Figure 1H*), linear mixed-effects regressions were fitted to the scaled single-cell data shown in *Figure 1—figure supplement 5B* (*statsmodel* package in Python, *statsmodels.formula.api.mixedlm* function). One model was fitted for each relative time interval between the end of nucleoid splitting and cell division, describing the rate of distance increase between the separated sister nucleoids ($\sigma$, response variable) as a function of the polysome concentration increase rate ($\rho$), the rate of cell elongation ($\lambda$), their interaction ($\rho\lambda$), and noise ($\varepsilon$) (*statsmodels.formula.api.ols*) as follows:

$$\sigma = \beta_0 + \beta_1\rho + \beta_2\lambda + \beta_3\left(\rho\lambda\right) + \varepsilon \tag{3}$$

The single-cell data were scaled by subtracting the population mean and then dividing by the standard deviation prior to model fitting for each of the four relative time intervals (in *Figure 1—figure supplement 5B*). The coefficients of the linear regression model (shown in *Figure 1I*) provide a measure of the relative effect of the independent variables on the dependent variable. The cell elongation ($\lambda$) had a statistically significant effect to the response variable ($\sigma$) (Prob(>|Z|)<$10^{-9}$) across all relative time intervals. The polysome accumulation coefficients ($\rho$) had a statistically significant effect to the response variable ($\sigma$) (Prob(>|Z|)<$10^{-14}$) for the first three relative time intervals (0–25%, 25–50% and 50–75%) and a marginally significant one (Prob(>|Z|)=0.02) for the last quartile (75–100%). The interaction term ($\rho\lambda$) had a marginally significant effect (Prob(>|Z|)=0.002) for the first quartile

(0–25%) yet with a very small coefficient of 0.06 and did not present any significance (Prob(>|Z|)>0.05) for the remaining time intervals (25–50%, 50–75% and 75–100%). Hence, it is not shown in *Figure 1I*. The most significant effect on the migration of the sister nucleoids (σ) was presented by the rate of polysome accumulation at mid-cell ( $\rho$ ), 0–25% from the end of nucleoid splitting to cell division, with a coefficient of 0.46 and Prob(>|Z|)<$10^{-84}$.

## Calculation of rates

The rates of the polysome accumulation ($\frac{\Delta RplA_{mid-nuc}\ conc.}{\Delta T}$) or nucleoid depletion ($\frac{\Delta HupA_{mid-nuc}\ conc.}{\Delta T}$) in the middle of the nucleoid (2.5 pixels adjacent to the nucleoid center) (*Figure 1F*), correspond to the slope of a linear regression (Python fitting function: *numpy.polyfit*) fitted to the change of the mid-nucleoid RplA-GFP or HupA-mCherry concentration over time, 40–90% into the nucleoid segregation cycle (as in *Figure 1E*). This nucleoid cycle interval corresponds to the average time from the initiation of nucleoid splitting to just before its completion. Choosing the appropriate range to correlate the two statistics was important since polysomes appeared to accumulate in the middle of the nucleoid before the onset of nucleoid splitting (*Figure 1E*, *Figure 1—figure supplement 6D*).

The rate of RplA-GFP concentration increase at mid-cell ($\frac{\Delta RplA_{mid-cell}\ conc.}{\Delta T}$) was calculated (*Figure 1H*) within a region covering 10% of the total cell length at the cell center. The same results were obtained when the rates were calculated within a cell region covering 2.5 pixels adjacent to the cell center. The rate of nucleoid migration ($\frac{\Delta Distance_{nuc}}{\Delta T}$) corresponds to the slope of a linear regression that describes the natural log-transformed distance increase between the separated sister nucleoids over time (*Figure 1H*). Similarly, the rate of cell elongation ($\frac{\Delta Length_{cell}}{\Delta T}$) was calculated considering the natural log-transformed cell length increase over time (*Figure 1H*). The rate of RplA-GFP concentration increase at mid-cell ($\frac{\Delta RplA_{mid-cell}\ conc.}{\Delta T}$), the rate of nucleoid migration ($\frac{\Delta Distance_{nuc}}{\Delta T}$), and the rate of cell elongation ($\frac{\Delta Length_{cell}}{\Delta T}$) were calculated within four relative time intervals from the end of nucleoid splitting to cell division (*Figure 1G*, *Figure 1—figure supplement 5B*). Only single nucleoid migration intervals with more than four timepoints were considered to ensure a reliable fitting. The small fraction (~8%) of cells that were born with two separately detected nucleoid objects, which indicates that the end of nucleoid splitting occurred in the previous cell division cycle, were removed from the analysis. The cell division cycles with a negative cell elongation rate (<1%) were also excluded from the analysis.

## Reaction-diffusion model

Our theoretical model is an extension of previous work (*Miangolarra et al., 2021*), which showed that steric effects between the DNA and the transcription-translation machinery (including polysomes) are sufficient to drive nucleoid segregation. To simply describe these interactions, we employed the Flory-Huggins theory for regular solutions (*Flory, 1942*; *Huggins, 1941*) and modeled the non-dimensionalized free-energy density $f$ as a function of the local volume fractions of the nucleoid ($n$) and polysomes ($p$):

$$f(n,p) = \frac{p}{v_p}\ln p + \frac{n}{v_n}\ln n + (1-p-n)\ln(1-p-n) + \chi_p p(1-p-n) + \chi_n n(1-p-n) + \chi_{np}np - \kappa_p \nabla p \nabla (1-p-n) - \kappa_n \nabla n \nabla (1-p-n) - \kappa_{np} \nabla n \nabla p. \tag{4}$$

Here, $(1-n-p)$ is the volume fraction of the cytoplasm not occupied by nucleoid or polysomes, whereas $v_n$ and $v_p$ are proportional to the molecular volumes of the elementary translational degrees of freedom (i.e. of the nucleoid element and single polysome, respectively). $\chi_p$, $\chi_n$, and $\chi_{np}$ are the Flory-Huggins interaction parameters, and $\kappa_p$, $\kappa_n$, and $\kappa_{np}$ are the interfacial tensions. We take $\kappa_{ij} = \lambda^2 \chi_{ij}$, with $\lambda$ being a characteristic interface width.

The time evolution of the volume fractions is described by a combination of the Cahn-Hilliard theory (*Cahn and Hilliard, 1958*) and the reaction kinetics of the polysomes. We assume that polysomes are produced inside the nucleoid at a constant rate $k_1$ and degraded uniformly along the cell at a rate $k_{-1}$:

$$\frac{\partial p}{\partial t} = \nabla(M_p \nabla \mu_p) + k_1 n - k_{-1}p, \tag{5}$$

$$\frac{\partial n}{\partial t} = \nabla(M_n \nabla \mu_n), \tag{6}$$

where $\mu_p = \frac{\delta F}{\delta p}$ and $\mu_n = \frac{\delta F}{\delta n}$ are the local chemical potentials of polysomes and nucleoid elements, respectively, with $F = \int f\left[n\left(x\right), p\left(x\right)\right] dx$ representing the total free energy. $M_p = v_p D_p p$ and $M_n = v_n D_n n$ are the mobility coefficients of the polysomes and the nucleoid, respectively, which depend on the diffusion coefficients of the polysomes ($D_p$) and the nucleoid ($D_n$). This recovers the Fick law of diffusion in the noninteracting limit.

Since *E. coli* grows by elongation while maintaining its cylindrical shape and cell width, we reduce the problem to one dimension, with $x$ representing the position along the long axis of the cell. We assume that the cell length grows exponentially such that the length at time $t$ is $L\left(t\right) = L\left(0\right) e^{\gamma t}$, with $\gamma$ being the growth rate and $L\left(0\right)$ the initial cell length at birth. Exponential growth dilutes the existing polysomes with rate $\gamma$. On the other hand, we assume that the nucleoid is not diluted since DNA replication occurs continuously, and the nucleoid length is known to grow in proportion to cell length (*Gray et al., 2019*). Letting $\widetilde{x} = x/L\left(t\right)$ be the relative position along the cell's long axis, the time evolution is given by:

$$\frac{\partial p}{\partial t} = \partial_{\widetilde{x}}\left(\widetilde{M}_p \partial_{\widetilde{x}} \mu_p\right) + k_1 n - \left(k_{-1} + \gamma\right) p, \tag{7}$$

$$\frac{\partial n}{\partial t} = \partial_{\widetilde{x}}\left(\widetilde{M}_n \partial_{\widetilde{x}} \mu_n\right), \tag{8}$$

where $\partial_{\widetilde{x}} = \frac{\partial}{\partial \widetilde{x}} = L\left(t\right)\frac{\partial}{\partial x}$, and $\widetilde{M}_{n,p} = M_{n,p} L\left(t\right)^{-2}$. The chemical potentials are:

$$
\begin{aligned}
\mu_p = v_p^{-1}\ln p - \ln\left(1 - p - n\right) + \chi_p\left(1 - 2p - n\right) + \left(\chi_{np} - \chi_n\right) n \\
- \chi_p \widetilde{\lambda}^2 \partial_{\widetilde{x}}^2\left(2p + n\right) + \left(\chi_{np} - \chi_n\right)\widetilde{\lambda}^2 \partial_{\widetilde{x}}^2 n,
\end{aligned} \tag{9}
$$

$$
\begin{aligned}
\mu_n = v_n^{-1}\ln n - \ln\left(1 - p - n\right) + \chi_n\left(1 - 2n - p\right) + \left(\chi_{np} - \chi_p\right) p \\
- \chi_n \widetilde{\lambda}^2 \partial_{\widetilde{x}}^2\left(2n + p\right) + \left(\chi_{np} - \chi_p\right)\widetilde{\lambda}^2 \partial_{\widetilde{x}}^2 p,
\end{aligned} \tag{10}
$$

where $\widetilde{\lambda} = \lambda/L(t)$.

We take the non-dimensionalized Flory-Huggins interaction parameters to be $\chi_p = 0.2$, $\chi_n = 0.4$, and $\chi_{np} = 1.2$. The $\chi_{np}$ parameter captures the steric repulsion between the nucleoid and the polysomes (*Miangolarra et al., 2021*). The diffusion coefficient of the polysome was set to $D_p = 0.023\,\mu\text{m}^2/\text{s}$, which lies within the same order of magnitude as was previously measured by single-molecule tracking (*Bakshi et al., 2012*). For simplicity, we adopt an average description of all polysomes with an average diffusion coefficient. Considering multiple polysome species does not change the physical picture. To illustrate, we considered an extension of our model, which contains three polysome species, each with a different diffusion coefficient ($D_p = 0.018, 0.023, 0.028\,\mu\text{m}^2/\text{s}$), reflecting that polysomes with more ribosomes will have a lower diffusion coefficient. Simulation of this model reveals that the different polysome species have essentially the same concentration distribution (*Figure 4—figure supplement 4*), suggesting that the average description in our minimal model is sufficient for our purposes.

Different diffusion coefficients were tested for the nucleoid ($D_n = 0.0005, 0.001\text{ or }0.005\,\mu\text{m}^2/\text{s}$) to capture different levels of nucleoid stiffness and hence response time to the local changes in polysome concentration. The polysome degradation rate was set to $k_{-1} = 0.003\text{s}^{-1}$, which corresponds to a half-life of around 5 min (*Bernstein et al., 2004*). We take the polysome production rate to be $k_1 = k_{1,0}\left(1 + \frac{\gamma}{k_{-1}}\right)$, with $k_{1,0} = 0.002\text{s}^{-1}$, to match the observation that the nucleoid length is proportional to cell length (*Gray et al., 2019*). The initial cell length $L\left(0\right)$ matched the cell length at birth, which has been previously shown to increase exponentially with the groOur model can also be extended to consider ectopic pwth rate (*Govers et al., 2024*). Hence, each of the simulated growth rates ($\gamma = 0.25, 0.36, 0.46, 0.57, 0.67, 0.78, 0.88, 0.99, 1.09, 1.2\text{h}^{-1}$) was matched to a cell length at birth (*Figure 4—figure supplement 2*) by fitting a linear regression ($L\left(0\right) = l_0 e^{\gamma/\gamma_0}$, with fitted parameters $l_0 = 2.00\,\mu\text{m}$ and $\gamma_0 = 2.81\text{h}^{-1}$) to the experimental data (*Govers et al., 2024*). Other parameters are chosen as follows: $\lambda = 0.03\,\mu\text{m}$, $v_n = 10$, $v_p = 5$.

The dynamic equations (*Equations 7-8*) were solved numerically in a fixed 1D domain $\widetilde{x} \in \left(0, 1\right)$, with no-flux boundary conditions. Space is discretized into $N = 128$ grid points, and time is discretized to steps of $\Delta t$. Time stepping was implemented with an implicit-explicit scheme as previously

demonstrated (*Mao et al., 2020*; *Mao et al., 2019*). The simulations were initialized from the (symmetric) steady-state solution of the system at the initial cell length $L(0)$ (*Figure 4B*, dashed curves in *Figure 4E*, and *Video 3*), or using the (asymmetric) distributions of polysomes and DNA in the predivisional cell of the former simulation as the initial condition (*Figure 4D*, solid curves in *Figure 4E*), which allowed us to capture cell polarity (new versus old pole). The cell length $L(t)$ was updated and recorded at each time step during the simulation. The nucleoid splitting time (shown in *Figure 4D*) was determined by thresholding the relative decrease of the nucleoid concentration at the cell center. A depletion threshold of 30% was used to mark the event of nucleoid splitting.

The $p$ field in our model describes the distribution of all polysomes, which could include one, two, or multiple ribosomes (in this paper, the term 'polysomes' refers to both monosomes and true polysomes). For simplicity, we adopt an average description of all polysomes with an average diffusion coefficient and interaction parameters, which is sufficient for capturing the fundamental mechanism underlying nucleoid segregation. The model can be extended to include multiple polysome species: $p = \sum_i p_i$, with each species' evolution following *Equation 5* with a different diffusion coefficient. *Figure 4—figure supplement 4* shows an example containing 3 polysome species, with diffusion coefficients $D_p = 0.018, 0.023, 0.028\ \mu\text{m}^2/\text{s}$.

Our model can also be extended to consider ectopic polysome production, which introduces an additional polysome production term:

$$\frac{\partial p}{\partial t} = \nabla \left( M_p \nabla \mu_p \right) + k_1' n - k_{-1} p + k_{\text{ect}}(x)$$

where $k_{\text{ect}}(x)$ describes ectopic polysome production from plasmids, with $x$ being the distance from one pole measured in $\mu$m. $k_1'$ represents a reduced polysome production rate in the nucleoid. *Figure 7—figure supplement 2* shows two examples with details as follows. For ectopic production near the poles, we use $k_{\text{ect}}(x) = 0.8 k_1 \Theta(x) \Theta(0.8 - x)$ and $k_1' = 0.5 k_1$, where $\Theta(x)$ is the Heaviside step function; this restricts ectopic production to within $0.8\mu$m from one pole. For ectopic production between sister nucleoids, we use $k_{\text{ect}}(x) = 0.5 k_1 \Theta\left(0.4 - \left|\frac{L}{2} - x\right|\right)$ and $k_1' = 0.5 k_1$, where $L$ is the cell length; this restricts ectopic production to within $0.4\mu$m from mid-cell. These ectopic production rates are for illustrative purposes only and the qualitative result does not depend sensitively on parameter values.

The code for the model (including scripts to reproduce all simulation figures) is available at https://github.com/qiweiyuu/polysome, copy archived at *Yu, 2025*.

## Polysome and nucleoid analysis based on fitted Gaussian functions

Quantification of polysome and nucleoid asymmetries within cells along the cell division cycle was achieved by fitting Gaussian functions (Python least squares optimization: *scipy.optimize.curve_fit*) to the RplA-GFP or the HupA-mCherry intensity profiles of cells growing in M9gluCAAT in the microfluidic device. Three Gaussian functions were fitted to the polysome signal concentration early in the cell division cycle. Two of them captured the nucleoid-excluded polysomes accumulating at the poles and the third captured the polysome accumulation in the middle of nucleoid (*Figure 3—figure supplement 1A and B*):

$$Poly(l) = A_{old}\exp\left(-\frac{1}{2}\left(\frac{l_{old} - l}{\sigma_{old}}\right)^2\right) + A_{mid}\exp\left(-\frac{1}{2}\left(\frac{l_{mid} - l}{\sigma_{mid}}\right)^2\right) + A_{new}\exp\left(-\frac{1}{2}\left(\frac{l_{new} - l}{\sigma_{new}}\right)^2\right)$$

$$(11)$$

where $A_{old}$, $A_{new}$ and $A_{mid}$ are the amplitudes, $l_{old}$, $l_{new}$ and $l_{mid}$ are the means, and $\sigma_{old}$, $\sigma_{new}$ and $\sigma_{mid}$ are the standard deviations of the respective fitted functions along the relative cell length ($l$).

Two Gaussian functions were fitted to the concentration of constricting nucleoids (*Figure 3—figure supplement 1A and B*):

$$Nuc(l) = A_{old}\exp\left(-\frac{1}{2}\left(\frac{l_{old} - l}{\sigma_{old}}\right)^2\right) + A_{new}\exp\left(-\frac{1}{2}\left(\frac{l_{new} - l}{\sigma_{new}}\right)^2\right)$$

$$(12)$$

All Gaussian functions were fitted over the cell background fluorescence, which corresponds to the baseline fluorescence within the masked cell boundaries and along the relative cell length coordinates from the old pole (–1) via the cell center (0) to the new pole (1). The Gaussian functions were fitted either to single newborn cells (0–2.5% into the cell division cycle) as shown in *Figure 3—figure supplement 1A and B* and as applied in *Figure 3A and E*, or to the less noisy average intensity profile of the segmented cells within the 0–10% range of their cell division cycle, as shown in *Figure 3—figure supplement 1C and D* and used in *Figure 3B and F*. The areas of the Gaussian functions fitted to the polysomes (used in *Figure 3A and B*) were estimated using the function:

$$Area = A\sigma\sqrt{2\pi} \tag{13}$$

where $A$ is the amplitude and $\sigma$ is the standard deviation.

The Gaussian areas were used to quantify the polysome asymmetries between the poles:

$$Poly_{asym} = log_{10}\left(\frac{Area_{new}}{Area_{old}}\right) \tag{14}$$

where $Area_{new}$ and $Area_{old}$ are the areas of the fitted Gaussians to the nucleoid-excluded polysomes accumulating in the new and the old poles, respectively. The statistic in *Equation 14* corresponds to the x-axis in *Figure 3B*. It is a positive value when the new pole contains more polysomes than the old one, and a negative value when the old pole contains more polysomes than the new one.

The nucleoid Gaussian statistics were used to quantify the nucleoid position asymmetry defined as:

$$Nuc_{pos} = \frac{l_{new} + l_{old}}{2}\frac{l_{cell}}{2} \tag{15}$$

and the nucleoid compaction asymmetry was calculated as:

$$Nuc_{comp} = \frac{A_{new}}{A_{old}} \tag{16}$$

where $l_{old}$ and $l_{new}$ are the means of the nucleoid Gaussians in relative cell coordinates (as in *Equation 12*) and $l_{cell}$ is the cell length in μm. The nucleoid position was first estimated in relative cell coordinates from the old pole (–1) via the cell center (0) to the new pole (1) and then multiplied by half the cell length to convert to absolute cell length units. $A_{old}$ and $A_{new}$ are the amplitudes of the fitted nucleoid Gaussians (as in *Equation 12*) in arbitrary fluorescence units. The nucleoid position corresponds to the mid-point between the two Gaussian means (*Equation 15*) and is a positive number when the nucleoid is shifted toward the new pole and a negative number when it is positioned closer to the old pole (y-axis in *Figure 3B*).

The nucleoid compaction asymmetry was calculated as the relative nucleoid Gaussian amplitude (*Equation 16*). It had a positive value greater than 1 when the nucleoid concentration was higher toward the new pole and had a positive value smaller than 1 if the nucleoid was more concentrated toward the old pole (y-axis in *Figure 3F*).

To determine which polysome Gaussian statistics and related asymmetries may contribute to nucleoid compaction asymmetry (*Equation 16*), a linear mixed-effects model was used (*statsmodel* package in Python, *statsmodels.formula.api.mixedlm* function). The polysome Gaussian statistics were used as independent variables and the nucleoid compaction asymmetry as a dependent variable:

$$Nuc_{comp} = \beta_0 + \beta_1 l_{old} + \beta_2 l_{mid} + \beta_3 l_{new} + \beta_4 \sigma_{old} + \beta_5 \sigma_{mid}$$
$$+ \beta_6 \sigma_{new} + \beta_7 A_{old} + \beta_8 A_{mid} + \beta_9 A_{new} + \varepsilon \tag{17}$$

where $\beta_0$ is the constant (intercept), $\beta_n$ (n=1,2...,9) are the coefficients for each of the polysome Gaussian statistics (same as in *Equation 11* and plotted in *Figure 3—figure supplement 2A and B*), and $\varepsilon$ is the error. The entire dataset was treated as a single group. Before fitting the linear mixed-effects model, all data were normalized by subtracting the average and dividing the resulting value by their standard deviation (*Figure 3—figure supplement 2B*). The relative cell length positions of the polysomes at the old pole ($l_{old}$) and in the middle of the nucleoid ($l_{mid}$) were found to significantly influence the asymmetric nucleoid compaction (*Figure 3—figure supplement 2C*). We reasoned that the available spaces between the polysome positions (*Figure 3—figure supplement 2A*) may influence

the compaction of the nucleoid and devised a new statistic that compares the available polysome-free space toward the new pole to that toward the old pole:

$$Poly_{space} = \frac{l_{new} - l_{mid}}{l_{mid} - l_{old}} \tag{18}$$

where $l_{new}$, $l_{mid}$, and $l_{old}$ are the polysome Gaussian means (as in *Equation 11*), which represent the positions of the polysomes at the new pole, mid-cell region, and old pole, respectively, expressed as relative cell coordinates from the old pole (–1) via the cell center (0) to the new pole (1). The fraction above (*Equation 18*) has a value greater than 1 when the polysome-free space is greater toward the new pole and a value smaller than 1 when the polysome-free space is greater toward the old pole (x-axis in *Figure 3F*).

### Linear regressions

Two types of linear regressions were used in this work. In *Figure 2B*, a first-degree polynomial was fitted using the ordinary least squares method and the numpy library in Python (*numpy.polyfit*), always using the *x*-axis as the independent variable and the *y*-axis as the dependent variable. In *Figures 1F and 3B*, a principal component regression was fitted using the scikit-learn library in Python (*sklearn.decomposition.PCA*). A principal component analysis was first applied on the two-dimensional z-transformed data to find the linear regressor that explained most of the variance (the first principal component). This linear regressor was then rescaled to the original 2D plane. The principal component regression eliminates the dependent variable bias associated with a traditional univariate fit. This bias is introduced when calculating the prediction error (sum of squares) only along the dependent variable axis during the least squares optimization. The elimination of this univariate bias is particularly important for the fitted regressions in *Figures 1F and 3B* since their parameters were used to estimate the rate of nucleoid splitting in the absence of polysome accumulation in the middle (y-intercept in *Figure 1F*), the position of the nucleoid with equal polysomes at the poles (y-intercept in *Figure 3B*), or the polar asymmetry required for a centered nucleoid (x-intercept in *Figure 3B*).

### Kernel density estimations

One-dimensional (*Figure 3A and E*, *Figure 1—figure supplement 4C*) or two-dimensional (*Figure 1F and H*, *Figure 1—figure supplement 5A and B*, and *Figure 3—figure supplement 2B*) Gaussian kernel density estimations were fitted using the *scipy.stats.gaussian_kde* method. The bandwidth was determined using Scott's rule (*Scott, 1992*).

### Correlation coefficient calculations

All Spearman correlation coefficients (referred to as Spearman $\rho$) were estimated using the *scipy.stats* Python library and the *spearmanr* function. Only those Spearman correlation coefficients with a p-value below 0.05 are shown.

## Acknowledgements

We are grateful to Dr. A Janakiraman (City College of New York) for the pER12 (pBAD322A-gfp-μNS) plasmid, Dr. K Prather (Massachusetts Institute of Technology) for the MG1655 (DE3) strain, and the members of the Jacobs-Wagner laboratory for fruitful discussions and critical reading of the manuscript. This work was supported in part by the National Institutes of Health (R01 GM082938 to NSW) and by the Dutch Research Council (research program Rubicon Science 2018–1, project number 019.181EN.018 to AP). CJW is an investigator of the Howard Hughes Medical Institute.

## Additional information

### Funding

| Funder | Grant reference number | Author |
|---|---|---|
| Howard Hughes Medical Institute | | Christine Jacobs-Wagner |
| National Institutes of Health | R01 GM082938 | Ned S Wingreen |
| Dutch Research Council | Rubicon Science 2018-1 019.181EN.018 | Alexandros Papagiannakis |

The funders had no role in study design, data collection and interpretation, or the decision to submit the work for publication.

### Author contributions

Alexandros Papagiannakis, Conceptualization, Resources, Data curation, Software, Formal analysis, Funding acquisition, Validation, Investigation, Visualization, Methodology, Writing – original draft, Writing – review and editing; Qiwei Yu, Resources, Software, Investigation, Methodology, Writing – review and editing; Sander K Govers, Investigation, Writing – review and editing; Wei-Hsiang Lin, Resources, Writing – review and editing; Ned S Wingreen, Supervision, Funding acquisition, Writing – review and editing; Christine Jacobs-Wagner, Conceptualization, Supervision, Funding acquisition, Visualization, Writing – original draft, Project administration, Writing – review and editing

### Author ORCIDs

Alexandros Papagiannakis ⬮ https://orcid.org/0000-0002-6363-804X
Qiwei Yu ⬮ https://orcid.org/0000-0003-0610-3484
Wei-Hsiang Lin ⬮ https://orcid.org/0000-0001-8177-5892
Ned S Wingreen ⬮ https://orcid.org/0000-0001-7384-2821
Christine Jacobs-Wagner ⬮ https://orcid.org/0000-0003-0980-5334

Reviewer #1 (Public review): https://doi.org/10.7554/eLife.104276.3.sa1
Reviewer #2 (Public review): https://doi.org/10.7554/eLife.104276.3.sa2
Reviewer #3 (Public review): https://doi.org/10.7554/eLife.104276.3.sa3
Author response https://doi.org/10.7554/eLife.104276.3.sa4

# Additional files

### Supplementary files

Supplementary file 1. Growth medium abbreviation and composition.

Supplementary file 2. *Escherichia coli* strains used in this study.

Supplementary file 3. Plasmids used in this study.

Supplementary file 4. DNA oligonucleotides used in this study.

Supplementary file 5. Chemicals used in this study.

Supplementary file 6. Software used in this study.

MDAR checklist

### Data availability

The microscopy images and data frames used in this study are available on BioStudies and the BioImage Archive (accession # S-BIAD1658). The analysis codes are uploaded on GitHub: https://github.com/JacobsWagnerLab/published/tree/master/Papagiannakis_2025, copy archived at *JacobsWagnerLab, 2025* and https://github.com/alexSysBio/Time_lapse_on_agarose_pad, copy archived at *Papagiannakis, 2025a*; https://github.com/alexSysBio/flowio_to_pandas, copy archived at *Papagiannakis, 2025b*; https://github.com/alexSysBio/Adding_ND2_images_to_python, copy archived at *Papagiannakis, 2025c*; https://github.com/alexSysBio/Cell_medial_axis_definitions, copy

archived at *Papagiannakis, 2025d*; https://github.com/alexSysBio/Image_background_subtraction, copy archived at *Papagiannakis, 2025e* as specified in the Methods and in *Supplementary file 6*. The modeling code is uploaded on GitHub:https://github.com/qiweiyuu/polysome, copy archived at *Yu, 2025*.

The following dataset was generated:

| Author(s) | Year | Dataset title | Dataset URL | Database and Identifier |
|---|---|---|---|---|
| Papagiannakis A, Yu Q, Govers S, Lin WH, Wingreen N, Jacobs-Wagner C | 2025 | Nonequilibrium polysome dynamics promote chromosome segregation and its coupling to cell growth in *Escherichia coli* | https://www.ebi.ac.uk/biostudies/bioimages/studies/S-BIAD1658 | BioImage Archive, S-BIAD1658 |

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
