## [Editor Report · eLife Assessment]

This **important** study presents **compelling** observational data supporting a role for transcription and polysome accumulation in the separation of newly replicated bacterial chromosomes. Through a comprehensive and rigorous comparative analysis of the spatiotemporal dynamics of ribosomal accumulation, nucleoid segregation, and cell division, the authors develop a model that nucleoid segregation rates are determined at least in part by the accumulation of ribosomes in the center of the cell, exerting a steric force to drive nucleoid segregation prior to cell division. This model circumvents the need to invoke as yet unidentified active mechanisms (e.g. an equivalent to a eukaryotic spindle) as drivers of bacterial chromosome segregation and intrinsically couples this vital step in the cell cycle to cell growth.

---

## [Referee Report · Reviewer #1 (Public review)]

Summary:

The paper by Papagiannakis et al is an elegant, mostly observational work detailing observations that polysome accumulation appears to drive nucleoid splitting and segregation. Overall I think this is an insightful work with solid observations.

Strengths:

The strengths of this paper are the careful and rigorous observational work that leads to their hypothesis. They find the accumulation of polysomes correlates with nucleoid splitting, and that the nucleoid segregation occurring right after splitting correlates with polysome segregation. These correlations are also backed up by other observations:

(1) Faster polysome accumulation and DNA segregation at faster growth rates.

(2) Polysome distribution negatively correlating with DNA positioning near asymmetric nucleoids.

(3) Polysomes form in regions inaccessible to similarly sized particles.

These above points are observational, I have no comments on these observations leading to their hypothesis.

Comments on revisions:

The authors have satisfied all of my concerns.

---

## [Referee Report · Reviewer #2 (Public review)]

Summary:

The authors perform a remarkably comprehensive, rigorous, and extensive investigation into the spatiotemporal dynamics between ribosomal accumulation, nucleoid segregation, and cell division. Using detailed experimental characterization and rigorous physical models, they offer a compelling argument that nucleoid segregation rates are determined at least in part by the accumulation of ribosomes in the center of the cell, exerting a steric force to drive nucleoid segregation prior to cell division. This evolutionarily ingenious mechanism means cells can rely on ribosomal biogenesis as the sole determinant for the growth rate and cell division rate, avoiding the need for two separate 'sensors,' which would require careful coupling.

Strengths:

In terms of strengths; the paper is very well written, the data are of extremely high quality, and the work is of fundamental importance to the field of cell growth and division. This is an important and innovative discovery enabled through the combination of rigorous experimental work and innovative conceptual, statistical, and physical modeling.

Weaknesses:

The authors have reasonably addressed by minor weaknesses raised in the first round of reviews, and I see no other weaknesses at this point worth raising.

---

## [Referee Report · Reviewer #3 (Public review)]

Summary:

Papagiannakis et al. present a detailed study exploring the relationship between DNA/polysome phase separation and nucleoid segregation in *Escherichia coli*. Using a combination of experiments and modelling, the authors aim to link physical principles with biological processes to better understand nucleoid organisation and segregation during cell growth.

Strengths:

The authors have a conducted a large number of experiments under different growth conditions and physiological perturbations (using antibiotics) to analyse the biophysical factors underlying the spatial organisation of nucleoids within growing *E. coli* cells. A simple model of ribosome-nucleoid segregation has been developed to explain the observations and tested with cleverly designed perturbation experiments.

The model and explanation presented in the original version have been strengthened with additional results and consideration of new factors. In particular, the radial attachment of the nucleoid, supported by previous studies and the A22 treatment data in this study, provides a plausible mechanism that prevents ribosomes from diffusing between and around the nucleoid lobes through the radial shells surrounding the nucleoid. The revised version of the paper incorporates this effect, resulting in model predictions that align well with the drug treatment outcomes and the observed mid-cell accumulation and confinement of ribosomes.

Furthermore, experiments involving plasmid-based gene expression, designed to redirect transcription away from chromosomal loci, offer compelling validation of the model's predictions. Overall, this is a robust and insightful study that will be of significant value to the quantitative microbiology community.

---

## [Author Response]

The following is the authors’ response to the original reviews

**Public Reviews:**

**Reviewer #1 (Public review):**
Summary:This paper is an elegant, mostly observational work, detailing observations that polysome accumulation appears to drive nucleoid splitting and segregation. Overall I think this is an insightful work with solid observations.

Thank you for your appreciation and positive comments. In our view, an appealing aspect of this proposed biophysical mechanism for nucleoid segregation is its self-organizing nature and its ability to intrinsically couple nucleoid segregation to biomass growth, regardless of nutrient conditions.

Strengths:The strengths of this paper are the careful and rigorous observational work that leads to their hypothesis. They find the accumulation of polysomes correlates with nucleoid splitting, and that the nucleoid segregation occurring right after splitting correlates with polysome segregation. These correlations are also backed up by other observations:(1) Faster polysome accumulation and DNA segregation at faster growth rates.(2) Polysome distribution negatively correlating with DNA positioning near asymmetric nucleoids.(3) Polysomes form in regions inaccessible to similarly sized particles.These above points are observational, I have no comments on these observations leading to their hypothesis.

Thank you!

Weaknesses:It is hard to state weaknesses in any of the observational findings, and furthermore, their two tests of causality, while not being completely definitive, are likely the best one could do to examine this interesting phenomenon.

It is indeed difficult to prove causality in a definitive manner when the proposed coupling mechanism between nucleoid segregation and gene expression is self-organizing, i.e., does not involve a dedicated regulatory molecule (e.g., a protein, RNA, metabolite) that we could have eliminated through genetic engineering to establish causality. We are grateful to the reviewer for recognizing that our two causality tests are the best that can be done in this context.

Points to consider / address:Notably, demonstrating causality here is very difficult (given the coupling between transcription, growth, and many other processes) but an important part of the paper. They do two experiments toward demonstrating causality that help bolster - but not prove - their hypothesis. These experiments have minor caveats, my first two points.(1) First, "Blocking transcription (with rifampicin) should instantly reduce the rate of polysome production to zero, causing an immediate arrest of nucleoid segregation". Here they show that adding rifampicin does indeed lead to polysome loss and an immediate halting of segregation - data that does fit their model. This is not definitive proof of causation, as rifampicin also (a) stops cell growth, and (b) stops the translation of secreted proteins. Neither of these two possibilities is ruled out fully.

That’s correct; cell growth also stops when gene expression is inhibited, which is consistent with our model in which gene expression within the nucleoid promotes nucleoid segregation and biomass growth (i.e., cell growth), inherently coupling these two processes. This said, we understand the reviewer’s point: the rifampicin experiment doesn’t exclude the possibility that protein secretion and cell growth drive nucleoid segregation. We are assuming that the reviewer is envisioning an alternative model in which sister nucleoids would move apart because they would be attached to the membrane through coupled transcription-translation-protein secretion (transertion) and the membrane would expand between the separating nucleoids, similar to the model proposed by Jacob et al in 1963 (doi:10.1101/SQB.1963.028.01.048). There are several observations arguing against cell elongation/transertion acting a predominant mechanism of nucleoid segregation.

(1) For this alternative mechanism to work, membrane growth must be localized at the middle of the splitting nucleoids (i.e., midcell position for slow growth and ¼ and ¾ cell positions for fast growth) to create a directional motion. To our knowledge, there is no evidence of such localized membrane incorporation. Furthermore, even if membrane growth was localized at the right places, the fluidity of the cytoplasmic membrane (PMID: 6996724, 20159151, 24735432, 27705775) would be problematic. To circumvent the membrane fluidity issue, one could potentially evoke an additional connection to the rigid peptidoglycan, but then again, peptidoglycan growth would have to be localized at the middle of the splitting nucleoid. However, peptidoglycan growth is dispersed early in the cell division cycle when the nucleoid splitting happens in fast growing cells and only appears to be zonal after the onset of cell constriction (PMID: 35705811, 36097171, 2656655).

(2) Even if we ignore the aforementioned caveats, Paul Wiggins’s group ruled out the cell elongation/transertion model by showing that the rate of cell elongation is slower than the rate of chromosome segregation (PMID: 23775792). In our revised manuscript, we clarify this point and provide confirmatory data showing that the cell elongation rate is indeed slower than the nucleoid segregation rate (Figure 1H and Figure 1 - figure supplement 5A), indicating that it cannot be the main driver.

(3) The asymmetries in nucleoid compaction that we described in our paper are predicted by our model. We do not see how they could be explained by cell growth or protein secretion.

(4) We also show that polysome accumulation at ectopic sites (outside the nucleoid) results in correlated nucleoid dynamics, consistent with our proposed mechanism. It is not clear to us how such nucleoid dynamics could be explained by cell growth or protein secretion (transertion).

(1a) As rifampicin also stops all translation, it also stops translational insertion of membrane proteins, which in many old models has been put forward as a possible driver of nucleoid segregation, and perhaps independent of growth. This should at last be mentioned in the discussion, or if there are past experiments that rule this out it would be great to note them.

It is not clear to us how the attachment of the DNA to the cytoplasmic membrane could alone create a directional force to move the sister nucleoids. We agree that old models have proposed a role for cell elongation (providing the force) and transertion (providing the membrane tether). Please see our response above for the evidence (from the literature and our work) against it. This was mentioned in the Introduction and Results section, but we agree that this was not well explained. We have now put emphasis on the related experimental data (Figure 1H, Figure 1 – figure supplement 5A) and revised the text (lines 199 - 210) to clarify these points.

(1b) They address at great length in the discussion the possibility that growth may play a role in nucleoid segregation. However, this is testable - by stopping surface growth with antibiotics. Cells should still accumulate polysomes for some time, it would be easy to see if nucleoids are still segregated, and to what extent, thereby possibly decoupling growth and polysome production. If successful, this or similar experiments would further validate their model.

We reviewed the literature and could not find a drug that stops cell growth without stopping gene expression. Any drug that affects the integrity or potential of the membrane depletes cells of ATP; without ATP, gene expression is inhibited. However, our experiment in which we drive polysome accumulation at ectopic sites decouples polysome accumulation from cell growth. In this experiment, by redirecting most of chromosome gene expression to a single plasmid-encoded gene, we reduce the rate of cell growth but still create a large accumulation of polysomes at an ectopic location. This ectopic polysome accumulation is sufficient to affect nucleoid dynamics in a correlated fashion. In the revised manuscript, we have clarified this point and added model simulations (Figure 7 – figure supplement 2) to show that our experimental observations are predicted by our model.

(2) In the second experiment, they express excess TagBFP2 to delocalize polysomes from midcell. Here they again see the anticorrelation of the nucleoid and the polysomes, and in some cells, it appears similar to normal (polysomes separating the nucleoid) whereas in others the nucleoid has not separated. The one concern about this data - and the differences between the "separated" and "non-separated" nuclei - is that the over-expression of TagBFP2 has a huge impact on growth, which may also have an indirect effect on DNA replication and termination in some of these cells. Could the authors demonstrate these cells contain 2 fully replicated DNA molecules that are able to segregate?

We have included new flow cytometry data of fluorescently labeled DNA to show that DNA replication is not impacted.

(3) What is not clearly stated and is needed in this paper is to explain how polysomes do (or could) "exert force" in this system to segregate the nucleoid: what a "compaction force" is by definition, and what mechanisms causes this to arise (what causes the "force") as the "compaction force" arises from new polysomes being added into the gaps between them caused by thermal motions.They state, "polysomes exert an effective force", and they note their model requires "steric effects (repulsion) between DNA and polysomes" for the polysomes to segregate, which makes sense. But this makes it unclear to the reader what is giving the force. As written, it is unclear if (a) these repulsions alone are making the force, or (b) is it the accumulation of new polysomes in the center by adding more "repulsive" material, the force causes the nucleoids to move. If polysomes are concentrated more between nucleoids, and the polysome concentration does not increase, the DNA will not be driven apart (as in the first case) However, in the second case (which seems to be their model), the addition of new material (new polysomes) into a sterically crowded space is not exerting force - it is filling in the gaps between the molecules in that region, space that needs to arise somehow (like via Brownian motion). In other words, if the polysome region is crowded with polysomes, space must be made between these polysomes for new polysomes to be inserted, and this space must be made by thermal (or ATP-driven) fluctuations of the molecules. Thus, if polysome accumulation drives the DNA segregation, it is not "exerting force", but rather the addition of new polysomes is iteratively rectifying gaps being made by Brownian motion.

We apologize for the understandable confusion. In our picture, the polysomes and DNA (conceptually considered as small plectonemic segments) basically behave as dissolved particles. If these particles were noninteracting, they would simply mix. However, both polysomes and DNA segments are large enough to interact sterically. So as density increases, steric avoidance implies a reduced conformational entropy and thus a higher free energy per particle. We argue (based on Miangolarra et al. 2021 PMID: 34675077 and Xiang et al. 2021 PMID: 34186018) that the demixing of polysomes and DNA segments occurs because DNA segments pack better with each other than they do with polysomes. This raises the free energy cost associated with DNA-polysome interactions compared to DNA-DNA interactions. We model this effect by introducing a term in the free energy χ_np, which refers to as a repulsion between DNA and polysomes, though as explained above it arises from entropic effects. At realistic cellular densities of DNA and polysomes, this repulsive interaction is strong enough to cause the DNA and polysomes to phase separate.

This same density-dependent free energy that causes phase separation can also give rise to forces, just in the way that a higher pressure on one side of a wall can give rise to a net force on the wall. Indeed, the “compaction force” we refer to is fundamentally an osmotic pressure difference. At some stages during nucleoid segregation, the region of the cell between nucleoids has a higher polysome concentration, and therefore a higher osmotic pressure, than the regions near the poles. This results in a net poleward force on the sister nucleoids that drives their migration toward the poles. This migration continues until the osmotic pressure equilibrates. Therefore, both phase separation (due to the steric repulsion described above) and nonequilibrium polysome production and degradation (which creates the initial accumulation of polysomes around midcell) are essential ingredients for nucleoid segregation.

This has been clarified in the revised text, with the support of additional simulation results showing how the asymmetry in polysome distribution causes a compaction force (Figure 4A).

The authors use polysome accumulation and phase separation to describe what is driving nucleoid segregation. Both terms are accurate, but it might help the less physically inclined reader to have one term, or have what each of these means explicitly defined at the start. I say this most especially in terms of "phase separation", as the currently huge momentum toward liquid-liquid interactions in biology causes the phrase "phase separation" to often evoke a number of wider (and less defined) phenomena and ideas that may not apply here. Thus, a simple clear definition at the start might help some readers.

In our case, phase separation means that the DNA-polysome steric repulsion is strong enough to drive their demixing, which creates a compact nucleoid. As mentioned in a previous point, this effect is captured in the free energy by the χ_np term, which is an effective repulsion between DNA and polysomes, though it arises from entropic effects.

In the revised manuscript, we now illustrate this with our theoretical model by initializing a cell with a diffuse nucleoid and low polysome concentration. For the sake of simplicity, we assume that the cell does not elongate. We observe that the DNA-polysome steric repulsion is sufficient to compact the nucleoid and place it at mid-cell (new Figure 4A).

(4) Line 478. "Altogether, these results support the notion that ectopic polysome accumulation drives nucleoid dynamics". Is this right? Should it not read "results support the notion that ectopic polysome accumulation inhibits/redirects nucleoid dynamics"?

We think that the ectopic polysome accumulation drives nucleoid dynamics. In our theoretical model, we can introduce polysome production at fixed sources to mimic the experiments where ectopic polysome production is achieved by high plasmid expression. The model is able to recapitulate the two main phenotypes observed in experiments (Figure 7). These new simulation results have been added to the revised manuscript (Figure 7 – figure supplement 2).

(5) It would be helpful to clarify what happens as the RplA-GFP signal decreases at midcell in Figure 1- is the signal then increasing in the less "dense" parts of the cell? That is, (a) are the polysomes at midcell redistributing throughout the cell? (b) is the total concentration of polysomes in the entire cell increasing over time?

It is a redistribution—the RplA-GFP signal remains constant in concentration from cell birth to division (Figure 1 – Figure Supplement 1E). This is now clarified in the revised text.

(6) Line 154. "Cell constriction contributed to the apparent depletion of ribosomal signal from the mid-cell region at the end of the cell division cycle (Figure 1B-C and Movie S1)" - It would be helpful if when cell constriction began and ended was indicated in Figures 1B and C.

Good idea. We have added markers in Figure 1C to indicate the average start of cell constriction. This relative time from birth to division was estimated as described in the new Figure 1 – figure supplement 2. We have also indicated that cell birth and division correspond to the first and last images/timepoint in Figure 1B and C, respectively. The two-imensional average cell projections presented in Figure 3D also indicate the average timing of cell constriction, consistent with our analysis in Figure 1 – figure supplement 2.

(7) In Figure 7 they demonstrate that radial confinement is needed for longitudinal nucleoid segregation. It should be noted (and cited) that past experiments of Bacillus l-forms in microfluidic channels showed a clear requirement role for rod shape (and a given width) in the positing and the spacing of the nucleoids.Wu et al, Nature Communications, 2020. "Geometric principles underlying the proliferation of a model cell system" https://dx.doi.org/10.1038/s41467-020-17988-7

Good point! We have revised the text to mention this work. Thank you.

(8) "The correlated variability in polysome and nucleoid patterning across cells suggests that the size of the polysome-depleted spaces helps determine where the chromosomal DNA is most concentrated along the cell length. This patterning is likely reinforced through the displacement of the polysomes away from the DNA dense region"It should be noted this likely functions not just in one direction (polysomes dictating DNA location), but also in the reverse - as the footprint of compacted DNA should also exclude (and thus affect) the location of polysomes

We agree that the effects could go both ways at this early stage of the story. We have revised the text accordingly.

(9) Line 159. Rifampicin is a transcription inhibitor that causes polysome depletion over time. This indicates that all ribosomal enrichments consist of polysomes and therefore will be referred to as polysome accumulations hereafter". Here and throughout this paper they use the term polysome, but cells also have monosomes (and 2 somes, etc). Rifampicin stops the assembly of all of these, and thus the loss of localization could occur from both. Thus, is it accurate to state that all transcription events occur in polysomes? Or are they grouping all of the n-somes into one group?

In the original discussion, we noted that our term “polysomes” also includes monosomes for simplicity, but we agree that the term should have been defined much earlier. The manuscript has been revised accordingly. Furthermore, in the revised manuscript, we have included additional simulation results with three different diffusion coefficients that reflect different polysome sizes to show that different polysome species with less or more ribosomes give similar results (Figure 4 – figure supplement 4). This shows that the average polysome description in our model is sufficient.

Thank you for the valuable comments and suggestions!

**Reviewer #2 (Public review):**
Summary:The authors perform a remarkably comprehensive, rigorous, and extensive investigation into the spatiotemporal dynamics between ribosomal accumulation, nucleoid segregation, and cell division. Using detailed experimental characterization and rigorous physical models, they offer a compelling argument that nucleoid segregation rates are determined at least in part by the accumulation of ribosomes in the center of the cell, exerting a steric force to drive nucleoid segregation prior to cell division. This evolutionarily ingenious mechanism means cells can rely on ribosomal biogenesis as the sole determinant for the growth rate and cell division rate, avoiding the need for two separate 'sensors,' which would require careful coupling.

Terrific summary! Thank you for your positive assessment.

Strengths:In terms of strengths; the paper is very well written, the data are of extremely high quality, and the work is of fundamental importance to the field of cell growth and division. This is an important and innovative discovery enabled through a combination of rigorous experimental work and innovative conceptual, statistical, and physical modeling.

Thank you!

Weaknesses:In terms of weaknesses, I have three specific thoughts.Firstly, my biggest question (and this may or may not be a bona fide weakness) is how unambiguously the authors can be sure their ribosomal labeling is reporting on polysomes, specifically. My reading of the work is that the loss of spatial density upon rifampicin treatment is used to infer that spatial density corresponds to polysomes, yet this feels like a relatively indirect way to get at this question, given rifampicin targets RNA polymerase and not translation. It would be good if a more direct way to confirm polysome dependence were possible.

The heterogeneity of ribosome distribution inside *E. coli* cells has been attributed to polysomes by many labs (PMID: 25056965, 38678067, 22624875, 31150626, 34186018, 10675340). The attribution is also consistent with single-molecule tracking experiments showing that slow-moving ribosomes (polysomes) are excluded by the nucleoid whereas fast-diffusing ribosomes (free ribosomal subunits) are distributed throughout the cytoplasm (PMID: 25056965, 22624875). These points are now mentioned in the revised manuscript.

Second, the authors invoke a phase separation model to explain the data, yet it is unclear whether there is any particular evidence supporting such a model, whether they can exclude simpler models of entanglement/local diffusion (and/or perhaps this is what is meant by phase separation?) and it's not clear if claiming phase separation offers any additional insight/predictive power/utility. I am OK with this being proposed as a hypothesis/idea/working model, and I agree the model is consistent with the data, BUT I also feel other models are consistent with the data. I also very much do not think that this specific aspect of the paper has any bearing on the paper's impact and importance.

We appreciate the reviewer’s comment, but the output of our reaction-diffusion model is a bona fide phase separation (spinodal decomposition). So, we feel that we need to use the term when reporting the modeling results. Inside the cell, the situation is more complicated. As the reviewer points out, there are likely entanglements (not considered in our model) and other important factors (please see our discussion on the model limitations). This said, we have revised our text to clarify our terms and proposed mechanism.

Finally, the writing and the figures are of extremely high quality, but the sheer volume of data here is potentially overwhelming. I wonder if there is any way for the authors to consider stripping down the text/figures to streamline things a bit? I also think it would be useful to include visually consistent schematics of the question/hypothesis/idea each of the figures is addressing to help keep readers on the same page as to what is going on in each figure. Again, there was no figure or section I felt was particularly unclear, but the sheer volume of text/data made reading this quite the mental endurance sport! I am completely guilty of this myself, so I don't think I have any super strong suggestions for how to fix this, but just something to consider.

We agree that there is a lot to digest. We could not come up with great ideas for visuals others than the schematics we already provide. However, we have revised the text to clarify our points and added a simulation result (Figure 4A) to help explain biophysical concepts.

**Reviewer #3 (Public review):**
Summary:Papagiannakis et al. present a detailed study exploring the relationship between DNA/polysome phase separation and nucleoid segregation in *Escherichia coli*. Using a combination of experiments and modelling, the authors aim to link physical principles with biological processes to better understand nucleoid organisation and segregation during cell growth.Strengths:The authors have conducted a large number of experiments under different growth conditions and physiological perturbations (using antibiotics) to analyse the biophysical factors underlying the spatial organisation of nucleoids within growing *E. coli* cells. A simple model of ribosome-nucleoid segregation has been developed to explain the observations.Weaknesses:While the study addresses an important topic, several aspects of the modelling, assumptions, and claims warrant further consideration.

Thank you for your feedback. Please see below for a response to each concern.

Major Concerns:Oversimplification of Modelling Assumptions:The model simplifies nucleoid organisation by focusing on the axial (long-axis) dimension of the cell while neglecting the radial dimension (cell width). While this approach simplifies the model, it fails to explain key experimental observations, such as:(1) Inconsistencies with Experimental Evidence:The simplified model presented in this study predicts that translation-inhibiting drugs like chloramphenicol would maintain separated nucleoids due to increased polysome fractions. However, experimental evidence shows the opposite-separated nucleoids condense into a single lobe post-treatment (Bakshi et al 2014), indicating limitations in the model's assumptions/predictions. For the nucleoids to coalesce into a single lobe, polysomes must cross the nucleoid zones via the radial shells around the nucleoid lobes.

We do not think that the results from chloramphenicol-treated cells are inconsistent with our model. Our proposed mechanism predicts that nucleoids will condense in the presence of chloramphenicol, consistent with experiments. It also predicts that nucleoids that were still relatively close at the time of chloramphenicol treatment could fuse if they eventually touched through diffusion (thermal fluctuation) to reduce their interaction with the polysomes and minimize their conformational energy. Fusion is, however, not expected for well-separated nucleoids since their diffusion is slow in the crowded cytoplasm. This is consistent with our experimental observations: In the presence of a growth-inhibitory concentration of chloramphenicol (70 μg/mL), nucleoids in relatively close proximity can fuse, but well-separated nucleoids condense and do not fuse. Since the growth rate inhibition is not immediate upon chloramphenicol treatment, many cells with well-separated condensed nucleoids divide during the first hour. As a result, the non-fusion phenotype is more obvious in non-dividing cells, achieved by pre-treating cells with the cell division inhibitor cephalexin (50μg/mL). In these polyploid elongated cells, well-separated nucleoids condensed but did not fuse, not even after an hour in the presence of chloramphenicol. We have revised the manuscript to add these data (illustrative images + a quantitative analysis) in Figure 4 – figure supplement 1.

(2) The peripheral localisation of nucleoids observed after A22 treatment in this study and others (e.g., Japaridze et al., 2020; Wu et al., 2019), which conflicts with the model's assumptions and predictions. The assumption of radial confinement would predict nucleoids to fill up the volume or ribosomes to go near the cell wall, not the nucleoid, as seen in the data.

The reviewer makes a good point that DNA attachment to the membrane through transertion could contribute to the nucleoid being peripherally localized in A22 cells. We have revised the text to add this point. However, we do not think that this contradicts the proposed nucleoid segregation mechanism described in our model. On the contrary, by attaching the nucleoid to the cytoplasmic membrane along the cell width, transertion might help reduce the diffusion and thus exchange of polysomes across nucleoids. We have revised the text to discuss transertion over radial confinement.

(3) The radial compaction of the nucleoid upon rifampicin or chloramphenicol treatment, as reported by Bakshi et al. (2014) and Spahn et al. (2023), also contradicts the model's predictions. This is not expected if the nucleoid is already radially confined.

We originally evoked radial confinement to explain the observation that polysome accumulations do not equilibrate between DNA-free regions. We agree that transertion is an alternative explanation. Thank you for bringing it to our attention. However, please note that this does not contradict the model. In our view, it actually supports the 1D model by providing a reasonable explanation for the slow exchange of polysomes across DNA-free regions. The attachment of the nucleoid to the membrane along the cell width may act as diffusion barrier. We have revised the text and the title of the manuscript accordingly.

(4) Radial Distribution of Nucleoid and Ribosomal Shell:The study does not account for well-documented features such as the membrane attachment of chromosomes and the ribosomal shell surrounding the nucleoid, observed in super-resolution studies (Bakshi et al., 2012; Sanamrad et al., 2014). These features are critical for understanding nucleoid dynamics, particularly under conditions of transcription-translation coupling or drug-induced detachment. Work by Yongren et al. (2014) has also shown that the radial organisation of the nucleoid is highly sensitive to growth and the multifork nature of DNA replication in bacteria.

We have revised the manuscript to discuss the membrane attachment. Please see the previous response.

The omission of organisation in the radial dimension and the entropic effects it entails, such as ribosome localisation near the membrane and nucleoid centralisation in expanded cells, undermines the model's explanatory power and predictive ability. Some observations have been previously explained by the membrane attachment of nucleoids (a hypothesis proposed by Rabinovitch et al., 2003, and supported by experiments from Bakshi et al., 2014, and recent super-resolution measurements by Spahn et al.).

We agree—we have revised the text to discuss membrane attachment in the radial dimension. See previous responses.

Ignoring the radial dimension and membrane attachment of nucleoid (which might coordinate cell growth with nucleoid expansion and segregation) presents a simplistic but potentially misleading picture of the underlying factors.

Please see above.

This reviewer suggests that the authors consider an alternative mechanism, supported by strong experimental evidence, as a potential explanation for the observed phenomena:Nucleoids may transiently attach to the cell membrane, possibly through transertion, allowing for coordinated increases in nucleoid volume and length alongside cell growth and DNA replication. Polysomes likely occupy cellular spaces devoid of the nucleoid, contributing to nucleoid compaction due to mutual exclusion effects. After the nucleoids separate following ter separation, axial expansion of the cell membrane could lead to their spatial separation.

This “membrane attachment/cell elongation” model is reminiscent to the hypothesis proposed by Jacob et al in 1963 (doi:10.1101/SQB.1963.028.01.048). There are several lines of evidence arguing against it as the major driver of nucleoid segregation:

(Below is a slightly modified version of our response to a comment from Reviewer 1—see page 3)

(1) For this alternative model to work, axial membrane expansion (i.e., cell elongation) would have to be localized at the middle of the splitting nucleoids (i.e., midcell position for slow growth and ¼ and ¾ cell positions for fast growth) to create a directional motion. To our knowledge, there is no evidence of such localized membrane incorporation. Furthermore, even if membrane growth was localized at the right places, the fluidity of the cytoplasmic membrane (PMID: 6996724, 20159151, 24735432, 27705775) would be problematic. To go around this fluidity issue, one could potentially evoke a potential connection to the rigid peptidoglycan, but then again, peptidoglycan growth would have to be localized at the middle of the splitting nucleoid to “push” the sister nucleoid apart from each other. However, peptidoglycan growth is dispersed prior to cell constriction (PMID: 35705811, 36097171, 2656655).

(2) Even if we ignore the aforementioned caveats, Paul Wiggins’s group ruled out the cell elongation/transertion model by showing that the rate of cell elongation is slower than the rate of chromosome segregation (PMID: 23775792). In the revised manuscript, we confirm that the cell elongation rate is indeed overall slower than the nucleoid segregation rate (see Figure 1 - figure supplement 5A where the subtraction of the cell elongation rate to the nucleoid segregation rate at the single-cell level leads to positive values).

(3) Furthermore, our correlation analysis comparing the rate of nucleoid segregation to the rate of either cell elongation or polysome accumulation argues that polysome accumulation plays a larger role than cell elongation in nucleoid segregation. These data were already shown in the original manuscript (Figure 1I and Figure 1 – figure supplement 5B) but were not highlighted in this context. We have revised the text to clarify this point.

(4) The membrane attachment/cell elongation model does not explain the nucleoid asymmetries described in our paper (Figure 3), whereas they can be recapitulated by our model.

(5) The cell elongation/transertion model cannot predict the aberrant nucleoid dynamics observed when chromosomal expression is largely redirected to plasmid expression (Figure 7). In the revised manuscript, we have added simulation results showing that these nucleoid dynamics are predicted by our model (Figure 7 – figure supplement 2).

Based on these arguments, we do not believe that a mechanism based on membrane attachment and cell elongation is the major driver of nucleoid segregations. However, we do believe that it may play a complementary role (see “Nucleoid segregation likely involves multiple factors” in the Discussion). We have revised the text to clarify our thoughts and mention the potential role of transertion.

Incorporating this perspective into the discussion or future iterations of the model may provide a more comprehensive framework that aligns with the experimental observations in this study and previous work.

As noted above, we have revised the text to mention transertion.

Simplification of Ribosome States:Combining monomeric and translating ribosomes into a single 'polysome' category may overlook spatial variations in these states, particularly during ribosome accumulation at the mid-cell. Without validating uniform mRNA distribution or conducting experimental controls such as FRAP or single-molecule measurements to estimate the proportions of ribosome states based on diffusion, this assumption remains speculative.

Indeed, for simplicity, we adopt an average description of all polysomes with an average diffusion coefficient and interaction parameters, which is sufficient for capturing the fundamental mechanism underlying nucleoid segregation. To illustrate that considering multiple polysome species does not change the physical picture, we have considered an extension of our model, which contains three polysome species, each with a different diffusion coefficient (*DP* = 0.018, 0.023, or 0.028 μm^2^/s), reflecting that polysomes with more ribosomes will have a lower diffusion coefficient. Simulation of this model reveals that the different polysome species have essentially the same concentration distribution, suggesting that the average description in our minimal model is sufficient for our purposes. We present these new simulation results in Figure 4 – figure supplement 4 of the revised manuscript.

**Recommendations for the authors:**

**Reviewer #1 (Recommendations for the authors):**
(1) Does the polysome density correlate with the origins? If the majority of ribosomal genes are expressed near the origins,

This is indeed an interesting point that we mention in the discussion. The fact that the chromosomal origin is surrounded by highly expressed genes (PMID: 30904377) and is located near the middle of the nucleoid prior to DNA replication (PMID: 15960977, 27332118, 34385314, 37980336) can only help the model that we propose by increasing the polysome density at the mid-nucleoid position.

(2) Red lines in 3C are hard to resolve - can the authors make them darker?

Absolutely. Sorry about that.

**Reviewer #2 (Recommendations for the authors):**
The authors use rifampicin treatment as a mechanism to trigger polysome disassembly and show this leads to homogenous RplA distribution. This is a really important experiment as it is used to link RplA localization to polysomes, and tp argue that RplA density is reporting on polysomes. Given rifampicin inhibits RNA polymerase, and given the only reference of the three linking rifampicin to polysome disassembly is the 1971 Blundell and Wild ref, it would perhaps be useful to more conclusively show that polysome depletion (as opposed to inhibition of mRNA synthesis, which is upstream of polysome assembly) by using an alternative compound more commonly linked to polysome disassembly (e.g., puromycin) and show timelapse loss of density as a function of treatment time. This is not a required experiment, but given the idea that RplA density reports on polysomes is central to the authors' interpretation, it feels like this would be a thing worth being certain of. An alternative model is that ribosomes undergo self-assembly into local storage depots when not being used, but those depots are not translationally active/lack polysomes. I don't know if I think this is likely, but I'm not convinced the rifampicin treatment + waiting for a relatively long period of time unambiguously excludes other possible mechanisms given the large scale remodeling of the intracellular environment upon mRNA inhibition. I 100% buy the relationship between ribosomal distribution and nucleoid segregation (and the ectopic expression experiments are amazing in this regard), so my own pause for thought here is "do we know those ribosomes are in polysomes in the ribosome-dense regions". I'm not sure the answer to this question has any bearing on the impact and importance of this work (in my mind, it doesn't, but perhaps there's a reason it does?). The way to unambiguously show this would really be to do CryoET and show polysomes in the dense ribosomal regions, but I would never suggest the authors do that here (that's an entire other paper!).

We agree that mRNAs play a role, as mRNAs are major components of polysomes and most mRNAs are expected to be in the form of polysomes (i.e., in complex with ribosomes). In addition, as mentioned above, the enrichments of ribosome distribution are known to be associated with polysomes (PMID: 25056965, 38678067, 22624875, 31150626, 34186018, 10675340). The attribution is consistent with single-molecule tracking experiments showing that slow-moving ribosomes (polysomes) are excluded by the nucleoid whereas fast-diffusing ribosomes (free ribosomal subunits) are distributed throughout the cytoplasm (PMID: 25056965, 22624875). This is also consistent with cryo-ET results that we actually published (see Figure S5, PMID: 34186018). We have added this information to the revised manuscript. Thank you for alerting us of this oversight.

On line 320 the authors state "Our single-cell studies provided experimental support that phase separation between polysomes and DNA contributes to nucleoid segregation." - this comes pretty out of left field? I didn't see any discussion of this hypothesis leading up to this sentence, nor is there evidence I can see that necessitates phase separation as a mechanistic explanation unless we are simply using phase separation to mean cellular regions with distinct cellular properties (which I would advise against). If the authors really want to pursue this model I think much more support needs to be provided here, including (1) defining what the different phases are, (2) providing explicit description of what the attractive/repulsive determinants of these different phases could be/are, and (3) ruling out a model where the behavior observed is driven by a combination of DNA / polysome entanglement + steric exclusion; if this is actually the model, then being much more explicit about this being a locally arrested percolation phenomenon would be essential. Overall, however, I would probably dissuade the authors from pursuing the specific underlying physics of what drives the effects they're seeing in a Results section, solely because I think ruling in/out a model unambiguously is very difficult. Instead, this would be a useful topic for a Discussion, especially couched under a "our data are consistent with..." if they cannot exclude other models (which I think is unreasonably difficult to do).

Thank you for your advice. We have revised the text to more carefully choose our words and define our terms.

Minor comments:The results in "Cell elongation may also contribute to sister nucleoid migration near the end of the division cycle" are really interesting, but this section is one big paragraph, and I might encourage the authors to divide this paragraph up to help the reader parse this complex (and fascinating) set of results!

We have revised this section to hopefully make it more accessible.

**Reviewer #3 (Recommendations for the authors):**
Technical Controls:The authors should conduct a photobleaching control to confirm that the perceived 'higher' brightness of new ribosomes at the mid-cell position is not an artefact caused by older ribosomes being photobleached during the imaging process. Comparing results at various imaging frequencies and intensities is necessary to address this issue.

The ribosome localization data across 30 nutrient conditions (Figure 2, Figure 1 – figure supplement 6, Figure 2 – Figure supplement 1, Figure 2 – Figure supplement 3 and Figure 5) are from snapshot images, which do not have any photobleaching issue. They confirm the mid-cell accumulation seen by time-lapse microscopy. We have revised the text to clarify this point.

Novelty of Experimental Measurements:While the scale of the study is unprecedented, claims of novelty (e.g., line 142) regarding ribosome-nucleoid segregation tracking are overstated. Similar observations have been made previously (e.g., Bakshi et al., 2012; Bakshi et al., 2014; Chai et al., 2014).

Our apologies. The text in line 142 oversimplified our rationale. This has been corrected in the revised manuscript.